# Iso-orientation bias of layer 2/3 connections unifies spontaneous, visually and optogenetically driven V1 dynamics

Tibor Rózsa [1,2], Rémy Cagnol [1,2] & Ján Antolík [1] ✉

Functionally specific long-range lateral connectivity in layer 2/3 of the adult primary visual cortex (V1) supports the integration of visual information across visual space and shapes spontaneous, visual and optogenetically driven V1 activity. However, a comprehensive understanding of how these diverse cortical regimes emerge from this underlying cortical circuitry remains elusive. Here we address this gap by showing how the same model assuming moderately iso-orientation biased long-range cortical connectivity architecture explains diverse phenomena, including (i) range of visually driven phenomena, (ii) modular spontaneous activity, (iii) the propagation of spontaneous cortical waves, and (iv) neural responses to patterned optogenetic stimulation. The model offers testable predictions, including presence of slower and iso-tropic spontaneous wave propagation in layer 4 and non-monotonicity of optogenetically driven cortical response to increasingly larger disk of illumination. We thus offer a holistic framework for studying how cortical circuitry governs information integration across multiple operating regimes.

A fundamental function of sensory systems is integrating information across sensory space. In vision, this is often studied by sensory stimulation and analysis of lateral interactions between spatially offset cortical sites[1–3]. However, such sensory-driven approaches often conflate the complexities of feed-forward processing with recurrent lateral interactions. The primary visual cortex (V1) exhibits complex dynamics even without visual input, either during spontaneous activity, or in response to artificial external stimulation, offering an alternative means to study cortical integration without feed-forward confounds. Several phenomena highlight the interconnectedness of visually and non-visually driven dynamics:

Spontaneous travelling waves (STW): Waves of activity propagating across V1[4–6] at speeds consistent with unmyelinated lateral axons[7], with propagation biases aligned with cortical functional organisation[8], and influencing behaviour[7,9].

Modular spontaneous activity: Spontaneously emerging activity patterns in V1 are correlated with functional maps[10], linking resting-state dynamics with visual coding.

Optogenetically evoked patterns of activity: Optogenetic perturbations of ongoing V1 activity bridge the resting and visually evoked dynamics[11–14], offering a way to study cortical interactions without engaging retino-cortical pathways. For example, optogenetic stimulation aligned with an orientation map is more effective at eliciting a response than un-aligned stimulation.

Ultimately, cortical dynamics across these different driving regimes (spontaneous, visually driven, and optogenetically driven) rely on a shared neural substrate. Yet, a systematic understanding of how this common substrate supports the diverse phenomena involved in visual integration, and under different cortical regimes, remains elusive.

Bridging this gap requires a comprehensive approach capable of simultaneously addressing anatomy, network dynamics, and function, which computational models are uniquely suited for. While offering valuable insights, previous computational studies fall short of unifying the full range of dynamical regimes and levels of integration. Recent phenomenological models have shown how modular activity arises spontaneously or through optogenetic stimulation in young

[1]Faculty of Mathematics and Physics, Charles University, Prague, Czech Republic. [2]These authors contributed equally: Tibor Rózsa, Rémy Cagnol. ✉e-mail: antolik@ksvi.mff.cuni.cz

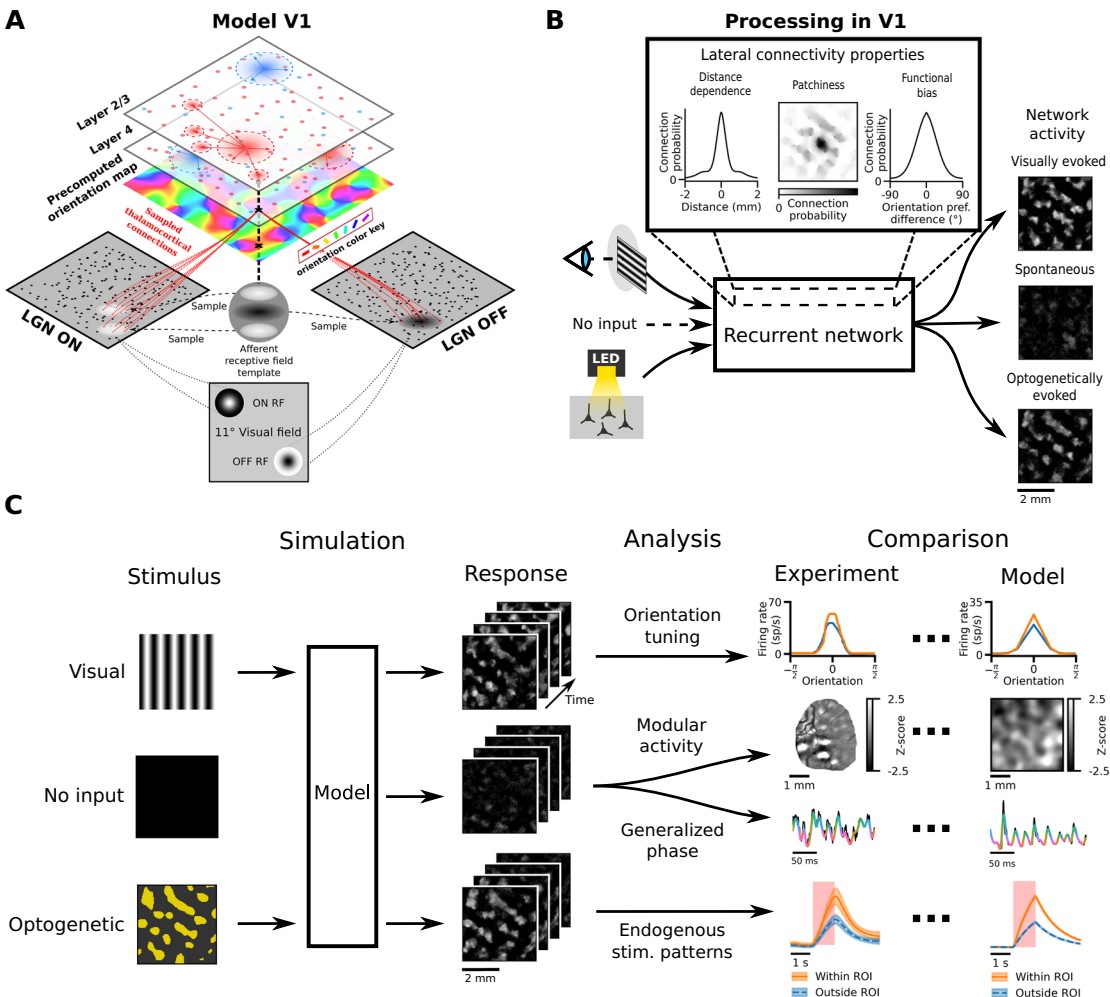

**Fig. 1 | Common neural substrate underlies a range of visually evoked, optogenetically evoked, and spontaneous phenomena. A** Schematic of the V1 model: thalamocortical connections to cortical Layer 4 from ON and OFF LGN cells are sampled based on a precomputed orientation preference map. All cortical neurons make local connections, while only Layer 2/3 neurons make long-range intra-layer horizontal connections. See "Methods" for the functionally specific rules of connectivity. **B** Hypothesis: the recurrent distance-dependent, patchy, orientation-biased horizontal connectivity in Layer 2/3 underpins spontaneous, visually and

optogenetically evoked cortical responses. **C** Computational approach: the simulated responses of neurons of a single parameterisation of the V1 model under multiple stimulation paradigms (no stimulus, visual or optogenetic) are recorded and analysed according to a range of experimentally measured criteria and compared to experimental results. Source data are provided as Source Data files. Illustrative calcium imaging frame adapted from Mulholland et al.[14], *Nature Communications*, CC BY 4.0; line curves adapted from ref. 79 under the GNU licence or scraped from refs. 12,80. Table 1 shows all experimental data sources.

animals[10,14], while others investigated travelling waves[15,16], optogenetic stimulation[17] or statistics of spontaneous activity[18] in adult V1. But all these models did not consider visually specific processing or anatomy. The model of Davis et al.[9] included structured connectivity and demonstrated functionally biased spontaneous activity, but did not link to visually or optogenetically driven processing. Finally, Cai et al.[19] demonstrated modular spontaneous activity in a single-layer V1 model, but did not address spontaneous wave propagation or external stimulation.

To explain how V1's shared neural substrate supports spatial visual integration across diverse dynamical regimes, we developed a unified model of generic adult columnar V1 (Fig. 1A) that accurately simulates visual processing, STW, modular spontaneous activity, and functionally specific optogenetic modulation (Fig. 1C). By comparing to data from multiple species (see data sources in Table 1), we demonstrate that the model replicates (i) distributions of wavelength and propagation speed in spontaneous waves observed in marmoset, along with biases toward iso-oriented propagation seen in macaque and cat V1, (ii) modular spontaneous activity correlated with orientation maps in both excitatory and inhibitory neurons in ferret V1, (iii)

impulse-response properties to both full-field homogeneous and patterned optogenetic stimulation in ferret V1, and (iv) a broad range of different visually driven phenomena. The model yields testable predictions, including slower, functionally independent spontaneous wave propagation in layer 4 and non-monotonic response of neurons in the surround of a disk of optogenetic stimulation that is increasing in diameter. Analysis of the model's parameters reveals that functional specific lateral interactions in layer 2/3 are critical for emergence of these spontaneous, visually driven and optogenetically driven phenomena (Fig. 1C), unifying an unprecedented range of previously fragmented experimental findings[4,8,10,12–15,20]. This study advances understanding of how these diverse processes are interlinked, moving towards a holistic understanding of sensory processing in the adult brain.

## Results

Here we present a generic model of adult columnar V1 derived from our previous work[21] (see "Methods" for full list of changes), which we will compare to data from multiple species (macaque, cat, ferret and marmoset). It considers the first two stages of V1 processing—the

**Table 1 | Sources for experimental data presented in Figs. 1–5. Source data are provided as Source Data files**

| Figure | Panel | Data description | Source |
|---|---|---|---|
| Fig. 1 | C | Orientation tuning curve | Alitto & Usrey[80] Fig. 3A |
| | | Spontaneous activity frame | Mulholland et al.[14]. Supplementary Fig. S4a |
| | | Generalized Phase Analysis time course | Lyle Muller[79], permissions obtained under the GNU licence. |
| | | Fullfield optogenetic stimulation response | Mulholland et al.[12] Fig. 5K |
| Fig. 2 | K | Propagation speed distribution | Davis et al. 2020[7] Extended Data 2d "Monkey T" |
| Fig. 3 | H | E. → Or. similarity | Smith et al.[10]. Supplementary Fig. 5g |
| | | All other metrics | Mulholland et al.[20]. Figs. 4C, E, G, J and 5F |
| | I | Correlation histograms | Kenet et al.[29] Fig. 2 |
| | JKL | Correlation histograms, predicted instantaneous firing rate | Tsodyks et al.[31] Fig. 3D–F |
| Fig. 4 | B | Optogenetic stimulation calc. imaging response | Mulholland et al.[14] Fig. 2C |
| | E | Spontaneous-opto correlation map similarity | Mulholland et al.[14] Supplementary Fig. S10c |
| | H | Endogenous stimulation pattern responses | Mulholland et al.[12] Fig. 5K |
| | L | Visual vs. visual+opto stimulation responses | Chernov et al. Figure 4L |
| Fig. 5 | C | Spontaneous activity calcium imaging signal | Kettlewell et al.[34] Fig. 1A |
| | | Same as above, deconvolved | Kettlewell et al.[34] Fig. 1B |

cortical layers 4 and 2/3, each populated by excitatory and inhibitory cells modelled as adaptive exponential integrate-and-fire neurons (Fig. 1A). Layer 4 receives input from a phenomenological model of the retino-thalamic pathway, with feed-forward receptive field templates globally organised according to pre-computed retinotopic and orientation preference maps (see "Methods"). The cortical model circuit follows multiple connectivity rules: the probability of connectivity falls off with distance, intra-layer 4 connectivity follows push-pull organisation, and the long-range layer 2/3 connections exhibit a moderate iso-orientation bias[22–24]; (Fig. 1B). The model was calibrated using anatomical and physiological data from cat and macaque, but we demonstrate here that it also explains activity dynamics in marmoset and ferret V1. As with its predecessor[21,25], the new model offers quantitatively accurate account of a broad range of visually evoked phenomena, including orientation tuning of spikes, membrane potential and conductances, distribution of modulation ratios, differences between flashed vs. continuous stimulation, and cross-trial spiking precision modulation by stimulus (see Supplementary Figs. S1–5).

### Spontaneous travelling waves exhibit iso-oriented bias of propagation dependent on functionally biased connectivity in layer 2/3

Voltage-sensitive dye and local field potential (LFP) studies have shown that cortical activity can propagate as travelling waves on both macroscopic and mesoscopic scales under various conditions[4–6]. To determine if the model's spontaneous activity exhibits wave-like patterns, we generated a synthetic LFP signal from the neurons' input current[26], which we then convolved with a 2D Gaussian (SD = 250 μm) to account for the spatial pooling of LFPs in electrode recordings[27] (see "Methods"). The signal was then wide-band filtered from 5 to 100 Hz. As illustrated in Fig. 2A, the synthetic LFP signal propagates across the model cortical surface as a STW.

We next wished to study the relationship between spontaneous wave propagation and the intrinsic functional maps identified by Nauhaus et al. in cats and macaques[8]. They used spike-triggered LFP (stLFP), which enables investigating how spiking activity at one reference recording location affects the LFP activity across all other recording sites. They found that the delay of the peak of the stLFP increased and that the amplitude of the peak decreased with distance from the reference electrode. We repeated their analysis using the spike trains of 207 model neurons as reference to compute the stLFPs (see "Methods").

As in the experimental data, we observe that the delay of the peak of stLFP increases with distance between recording sites

(Fig. 2B), and that its amplitude decreases exponentially based on distance (Fig. 2C), confirming the validity of the stLFP method implementation in our model. The mean propagation speed of the stLFP signal is 0.52 m/s, within one standard deviation from the value of 0.31 m/s reported in cat V1 by Nauhaus et al. (Fig. 2D). For each reference electrode, Nauhaus et al. used an exponential function to fit the relationship between the stLFP amplitude and the distance to the reference electrode, and then computed the stLFP residuals by subtracting the value predicted by the exponential fit from the stLFP amplitude at each electrode. We applied the same fitting procedure (Fig. 2C) and computed stLFP residuals for all but two reference neurons for which the fitting performances were poor (see "Methods"). Crucially, Nauhaus et al. observed significantly higher amplitudes of stLFP residuals at recording sites with orientation preference congruent to the reference electrode than for sites with orthogonal preference, indicating functionally specific wave propagation. We find that, for a given reference neuron, iso-oriented recording sites has significantly higher stLFP residuals than recording sites in orthogonal domains (Iso-oriented: $0.0124 \pm 0.0008$, ortho-oriented: $-0.0129 \pm 0.0008$, $p = 4.74 \times 10^{-31}$, $n = 205$, two-sided Wilcoxon signed-rank test) (Fig. 2E). This is inline with the observations of Nauhaus et al., although the magnitude of the iso-orientation bias they report is higher. Overall, these findings indicate that cortical STW propagate preferentially towards iso-oriented domains in the model as observed experimentally, confirming that we successfully account for functionally specific spontaneous V1 dynamics. Furthermore, we performed the same analysis after removing the functional bias in the long-range excitatory connectivity in the model layer 2/3. After this manipulation, the stLFP residuals are no longer higher for iso-oriented recording sites than for orthogonal recording sites (Iso-oriented: $-0.0020 \pm 0.0008$, ortho-oriented: $-0.0001 \pm 0.0008$, $p = 0.07$, $n = 201$, two-sided Wilcoxon signed-rank test) (Fig. 2E), demonstrating that functionally specific connectivity is critical for the iso-oriented bias of STW propagation.

Next, Davis et al. offered a detailed characterization of the spontaneous wave properties in marmoset middle temporal area (MT) using the Generalized Phase analysis[7,15]. We applied the Generalized Phase analysis to compute the instantaneous phases of the wideband filtered signal (Fig. 2F), yielding the distribution of wavelengths and propagation speeds of the synthetic LFP (see "Methods"), which we compared to Davis et al. marmoset data. Prior to convolution, the wavelengths of our synthetic LFP signal are significantly higher than in the spatially shuffled signal, demonstrating that the signal is indeed spatially structured (Fig. 2J). After applying

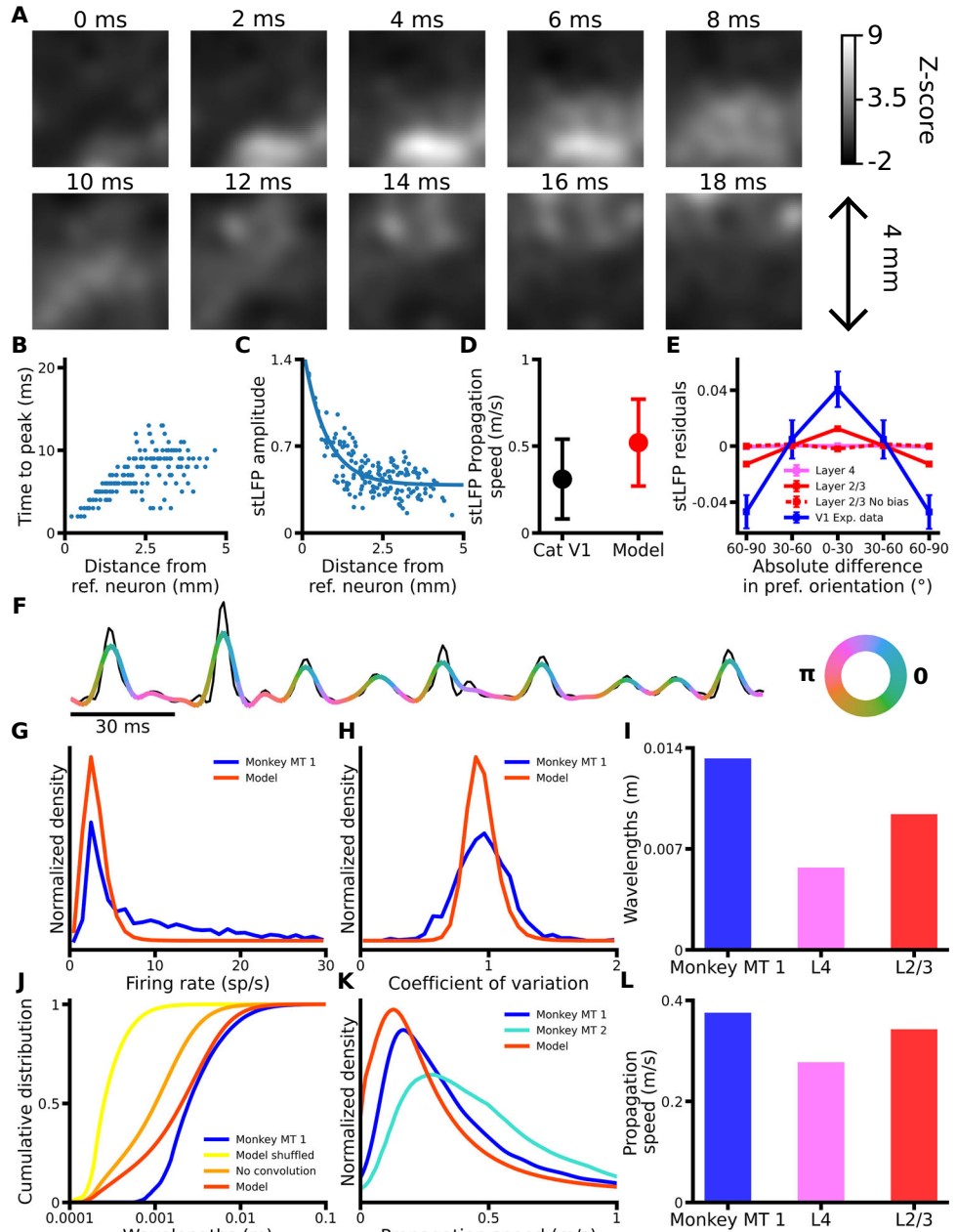

**Fig. 2 | Spontaneous travelling waves in the V1 model. A** Example LFP signal of the model Layer 2/3 spontaneous activity showing propagation of a STW. See Supplementary Movie 1 for the corresponding movie. **B** Spike-triggered LFP (stLFP) peak times for each spatial location plotted against their distance from one example reference neuron from layer 2/3. **C** stLFP amplitude for each spatial location plotted against their distance from one example reference neuron from layer 2/3. Solid blue line: exponential fit. **D** stLFP propagation speeds in cat V1 (black) from Nauhaus et al.[8], and for the model. Error bars represent the standard deviations. **E** Mean stLFP residuals for layers 4 (pink, *n* = 207) and 2/3 (red, solid: default configuration, *n* = 205; dashed: no functional bias in horizontal connections *n* = 201; blue: experimental data from Nauhaus et al.[8]) across reference neurons and recording sites, binned by orientation preference differences. Error bars: Standard error of the mean. **F** Synthetic LFP signal (black) and corresponding output of the

generalised phase analysis (color-coded with the instantaneous phase), for one recording site. **G** Distribution of the neurons' firing rates in model Layer 2/3 (red) and analogous experimental data in marmoset MT (Davis et al.[15]; blue). **H** Same as (**G**), for the coefficient of variation of the inter-spike-interval. **I** The average wavelengths for model layers 4 (pink) and 2/3 (red) and for the experimental data of Davis et al.[15], computed with the Generalized phase analysis. **J** Cumulative distributions of the wavelengths of the simulated LFP signal in model Layer 2/3 before (orange) and after (red) its spatial convolution with a Gaussian. Yellow: Distribution for the spatially shuffled signal; blue: distribution for recordings in marmoset MT by Davis et al.[15]. **K** Propagation speeds distribution of the LFP signal in the model (red) and in 2 marmosets (turquoise: data from Davis et al.[7]; blue: Davis et al.[15]). **L** Same as (**I**), for propagation speeds. Source data are provided as Source Data files; Table 1 shows experimental data sources.

the Gaussian convolution to account for spatial pooling, the distribution of the wavelengths in the signal closely matches experimental data in marmoset MT at high and intermediate wavelengths. However, we observe a greater proportion of low wavelengths, likely due to the limited simulated cortical area and the absence of extra-areal input, which increases global network synchronisation. The

distribution of propagation speeds also lies within the inter-subject variability identified by Davis et al.[7,15] (Fig. 2K). The peaks of both the distribution of firing rates and coefficients of variation (CV) match well the experimental data from marmoset MT by Davis et al. (Fig. 2G, H), ensuring that our network is in a realistic regime of activity similar to that of animal visual cortex.

Due to inaccessibility to imaging techniques, STWs have not been yet analysed in layer 4 of V1. We hypothesised that due to the absence of functionally specific long-range lateral connectivity in layer 4[28], the propagation of STWs in layer 4 would lack orientation bias. Indeed, we find that the stLFP residuals for recording sites of orientation preference congruent to the reference neurons' were not significantly different to stLFP at orthogonally oriented sites (Iso-oriented: $0.0001 \pm 0.0003$, ortho-oriented: $-0.0005 \pm 0.0003$, $p = 0.23$, $n = 207$, two-sided Wilcoxon signed-rank test) (Fig. 2E). Interestingly, we also find that propagation speeds and wavelengths in model layer 4 are lower than in layer 2/3 (Fig. 2I, L). We attribute this to shorter-range connectivity in layer 4, which requires activity to make multiple synaptic jumps to propagate across the same distance. Polysynaptic transmission of activity has previously been hypothesized to slow down traveling waves propagation[6].

### Functional specificity of cortical connections dictate correlation structure of spontaneous activity

Activity of both inhibitory and excitatory neurons is correlated over multiple millimetre distances, with correlation patterns of excitatory activity resembling orientation preference maps in young and adult carnivore/primate V1[10,20,29,30]. To test the presence of this phenomenon in the model we have first replicated the analysis of Smith et al.[10], who recorded from the V1 of young ferrets: first, we accounted for the spatiotemporal blurring of the underlying neural signal due to calcium imaging (Fig. 3A; "Methods"; Fig. S6) and subsequently calculated correlation maps (CMs) for each cortical position (Fig. 3C), by correlating the activity time course at the given "seed" cortical position to the activity time courses of all other cortical positions.

We observed long-range correlations across the model cortex, with excitatory CMs with seeds at iso-oriented positions in the orientation map being near-identical (Fig. 3C), in line with ferret data[10]. The model also replicates the finding that excitatory CMs match inhibitory CMs (Fig. 3D). When we quantify their similarity (see "Methods"), it is consistently high regardless of seed position (Fig. 3E, mean similarity = $0.457 \pm 0.002$, range $(-1,1))$[20]. Additionally, excitatory CMs exhibit high similarity (see "Methods") to the orientation preference map, regardless of seed position (Fig. 3F (left), mean similarity = $0.744 \pm 0.001$, range $(0,1))$. Note that this similarity metric is different from the aforementioned CM to CM similarity due to the periodicity of orientation preference[10]. Crucially, the orientation map similarity diminishes when we remove the orientation bias of Layer 2/3 long-range horizontal connections (Figs. 3E (right), 4F), further underlining the importance of the functional biased lateral connectivity in shaping spontaneous cortical dynamics.

Next, replicating the findings of Kenet et al.[29] in cat V1, we reconstructed the orientation preference map from excitatory spontaneous activity by fitting a circular Self-Organizing Map (SOM) to excitatory CMs (Fig. 3G), demonstrating that the orientation preference encoding can be fully recovered from the correlations of the model spontaneous activity.

Finally, we quantitatively assess how well the model captures the spatial properties of in-vivo spontaneous activity using several key metrics of spontaneous V1 activity[10,20] (Fig. 3H), overall finding a good match with young ferret data (see "Methods" for a detailed description of all metrics). The model reproduces the degree of similarity of excitatory and inhibitory CMs, long-range correlations and correlation wavelength for both excitatory and inhibitory activity, while the inhibitory dimensionality and local correlation eccentricity lie only slightly out of experimental data ranges. Our model does exhibit somewhat higher similarity between spontaneous activity and orientation preference maps than observed in young ferrets, which could be due to the absence of measurement noise in the model. Alternatively, this discrepancy could be explained by the young age of the imaged ferrets, whereby the orientation representation becomes more refined as

the animal matures (see Fig. 5C in ref. 10), reflecting the stronger alignment between orientation maps and spontaneous activity we observe in the model of adult V1.

To validate our model also against data from adult animals, we assess how well the model reproduces the level of correlation between the orientation preference map and spontaneous activity measured in adult cat (Kenet et al.[29], Tsodyks et al.[31]) using alternative metrics. The distribution of correlation coefficients between spontaneous activity frames and single-condition orientation maps (activity maps in response to gratings of a single orientation; see "Methods") matches that of Kenet et al.[29], including the control distribution computed using horizontally flipped single-condition maps (Fig. 3I). Next, the model matches the correlation distributions of spontaneous activity frames to the Preferred Cortical State (PCS) of neurons (Fig. 3J), a proxy for single-condition orientation maps (Tsodyks et al.[31], see "Methods"). Additionally, the subset of the above distribution consisting of only spontaneous frames during which the neuron fired an action potential (Fig. 3K), exhibits a significant positive bias ($p < 10^{-16}$, $n = 367169$, 1-sided one-sample T-test; mean = $0.1007 \pm 0.0003$ for model, $0.1591 \pm 0.0045$ for experiment). Thus, the predicted firing rate (Fig. 3L, ratio of histograms K, J) increases with stronger correlations to the orientation map, in line with cat data.

### Iso-orientation biased layer 2/3 connectivity boosts response to optogenetic stimulation congruent with orientation maps

We next investigated the V1 model's response to optogenetic stimulation from the cortical surface (Fig. 4A). To do so, we used a brain-machine-interface (BMI) model we recently developed[32], that incorporates an LED array, light propagation through neural tissue, and channelrhodopsin dynamics (see "Methods"; Fig. S8). The BMI model determines the current injected into each V1 neuron through the channelrhodopsin channels in response to arbitrary light stimulation patterns. Subsequently, the network's response to the optogenetic input can be simulated.

First, we replicated findings from a study that utilised an analogous optogenetic stimulation setup to modulate young ferret V1 with homogeneous full-field stimulation[14]. We calibrated the intensity of optogenetic stimulation such that the time course of a synthetic analog of the calcium imaging signal derived from the model's activity (see "Methods") aligned with its biological counterpart (ref. 14; Fig. 4B). Experimentally, responses to full-field optogenetic stimulation vary from trial to trial but CMs derived from such optogenetically evoked activity closely resemble those derived from spontaneous activity[14]. Consistent with these results, the model full-field optogenetic responses were variable (Supplementary Fig. S9) and the derived CMs closely matched spontaneous CMs (Fig. 4C), irrespective of seed position (Fig. 4D). The mean similarity of spontaneous and opto-response CMs was within the range of experimentally measured values (Fig. 4E). As with spontaneous activity, the full-field opto-response CMs showed high similarity to the orientation map, but removing the orientation bias of long-range connections from the model decreased the similarity to chance level, demonstrating that the functional biases in later-connectivity mediate the link between orientation maps and optogenetically evoked activity (Fig. 4F; see "Methods").

Next, we replicated Mulholland et al.'s[12] experiments that used a thresholded CM seeded at an arbitrary position (endogenous stimulation pattern—see "Methods") as optogenetic stimulation pattern (Fig. 4G, bottom left). Since CMs are highly correlated with the orientation map, they predominantly stimulate cortical positions within a narrow band of orientation preferences (Fig. 4G, top left). Following Mulholland et al.[13], we also created surrogate stimulation patterns that match the spatial statistics of endogenous patterns, but are unaligned with the spontaneous CMs (see "Methods") to serve as controls. Finally, we define the region-of-interest (ROI) for the given

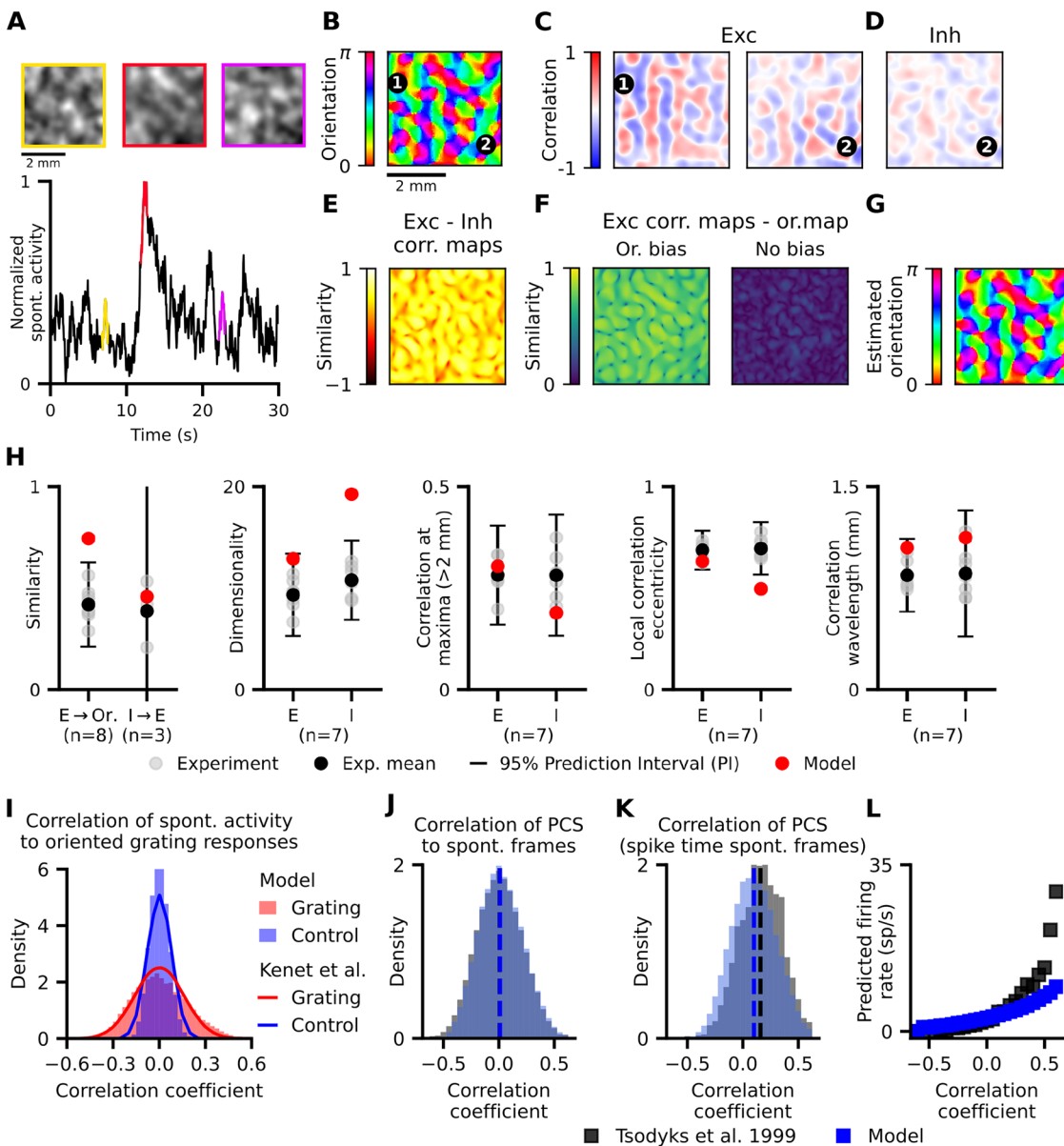

**Fig. 3 | The structure of correlations in spontaneous activity is related to orientation maps. A** Example spontaneous activity events in the simulated visual cortex calcium imaging recordings. The colour border indicates the dominant orientation preference of activated neurons. **B** Orientation map with two iso-oriented points[1,2] and **C** correlation maps (CMs) of excitatory activity with seeds at those points. The CMs with iso-oriented seeds are similar (example CMs similarity = 0.73). **D** CMs of inhibitory activity are similar to CMs from excitatory activity (example CMs similarity = 0.57). **E** Similarity of excitatory and inhibitory CMs is high, regardless of seed position. **F** (left) Similarity between CMs at each seed point and the orientation map. Similarity is consistently high regardless of seed position. (right) Removing the orientation bias of lateral connections decreases similarity. **G** A circular SOM fitted to spontaneous excitatory CMs predicts the structure of the orientation map, as seen in Kenet et al.[29]. **H** For a wide range of metrics recorded in the ferret cortex[10,20], excitatory and inhibitory activity in the model lies within

experimentally recorded values. "n", "E", "I", "Or." denote number of animals, excitatory activity, inhibitory activity and orientation preference map, respectively. **I** Validation against Kenet et al.[29]: The distribution of correlation coefficients between model spontaneous activity frames and single-condition orientation preference map is consistent with experimental data, including correlations to control (horizontally flipped) maps. **J–L** Validation against Tsodyks et al.[31]: **J** The distribution of correlations between model spontaneous activity frames and Preferred Cortical States (PCS) (see "Methods") closely matches experimental data. **K** The distribution of correlations to PCS for spontaneous frames at which the monitored neuron fired an action potential exhibits a significant positive bias, consistent with experimental data. **L** The predicted instantaneous firing rate increases with PCS correlation for both model and experiment. Source data are provided as Source Data files; Table 1 shows experimental data sources.

optogenetic stimulus as the area of cortex beneath the stimulation pattern (Fig. 4G; "Methods").

Model responses to stimulation with endogenous patterns matched the maximum-normalised population response time course of young ferret[12], both within and outside the stimulation ROIs (Fig. 4H, left). This result suggests that our model can account for the spatio-temporal dynamics of V1 responses to patterned optogenetic stimulation. Consistent with the results of Mulholland et al.[13], we found that

endogenous patterns evoked responses in the model that are stronger and more correlated to the stimulation patterns than those evoked by surrogate stimuli (Fig. 4HI). We hypothesised that this is because aligned stimulation amplifies network responses via functionally specific recurrent connectivity, while unaligned stimulation fails to effectively engage this circuitry. This recurrent amplification thus acts as a "soft winner-take-all" mechanism across the orientation domain, amplifying the dominant orientations while inhibiting the rest, leading

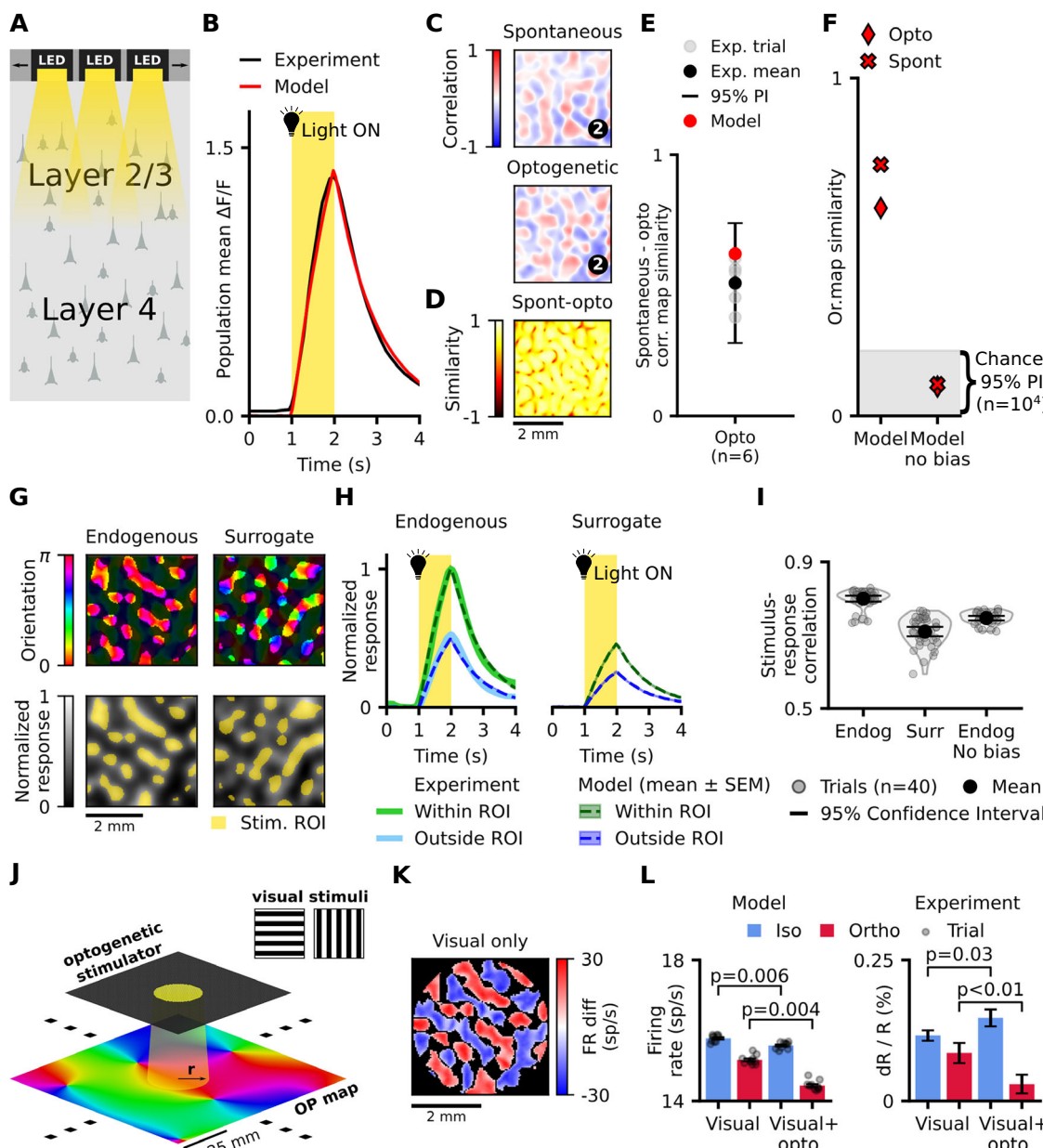

**Fig. 4 | Full-field and patterned stimulation evokes structured model responses. A** Schematic of optogenetic stimulation of model layer 2/3 neurons using a 400 × 400 LED array. **B** Mean simulated population response to full-field optogenetic cortical stimulation matches experimental data[14]. **C** CMs calculated from spontaneous activity (top) and response to optogenetic stimulation (bottom) at the same seed position are similar (example CMs similarity = 0.64). **D** Spontaneous and optogenetic response CMs are highly correlated to each other, regardless of seed position. **E** The mean correlation between spontaneous activity and optogenetic responses matches experimental (*n* = 6 animals) results. **F** The similarity of both spontaneous and optogenetic correlation maps to the orientation map decreases to chance level (*n* = 10⁴ spatial positions) when we remove the orientation bias of long-range connections. **G** (left) Endogenous stimulation patterns based on CMs and (right) statistically matched surrogate patterns uncorrelated with CMs. **H** The model matches the experimentally measured[12] maximum-normalised mean population response to endogenous stimulation both within and outside the stimulation ROI (left). Stimulating with surrogate patterns evokes a weaker activation

compared to the endogenous stimulation response (right). **I** Correlation between the stimulation pattern and evoked network activity pattern is higher for endogenous stimuli than for surrogate stimuli, in line with ferret experiments[13]. Removing the orientation bias from long-range connections decreases the endogenous stimulus-response correlation towards the level of surrogate stimulus responses. **J** Schematic of Chernov et al[33]. experiment: simultaneous visual stimulation with full-field square gratings and optogenetic stimulation of an orientation preference column (circle, *r* = 100 μm), **K** Spatial map of regions with a significant preference for visual stimuli iso-oriented (iso-regions, blue) or ortho-oriented (ortho-regions, red) relative to the optogenetically stimulated cortical column, within 2 mm from the stimulation site. **L** (left) Optogenetic stimulation significantly increases response in iso-regions (blue) and decreases response in ortho-regions (right). Intrinsic imaging data from Chernov et al.[33] show a qualitatively similar effect. Source data are provided as Source Data files; Table 1 shows experimental data sources.

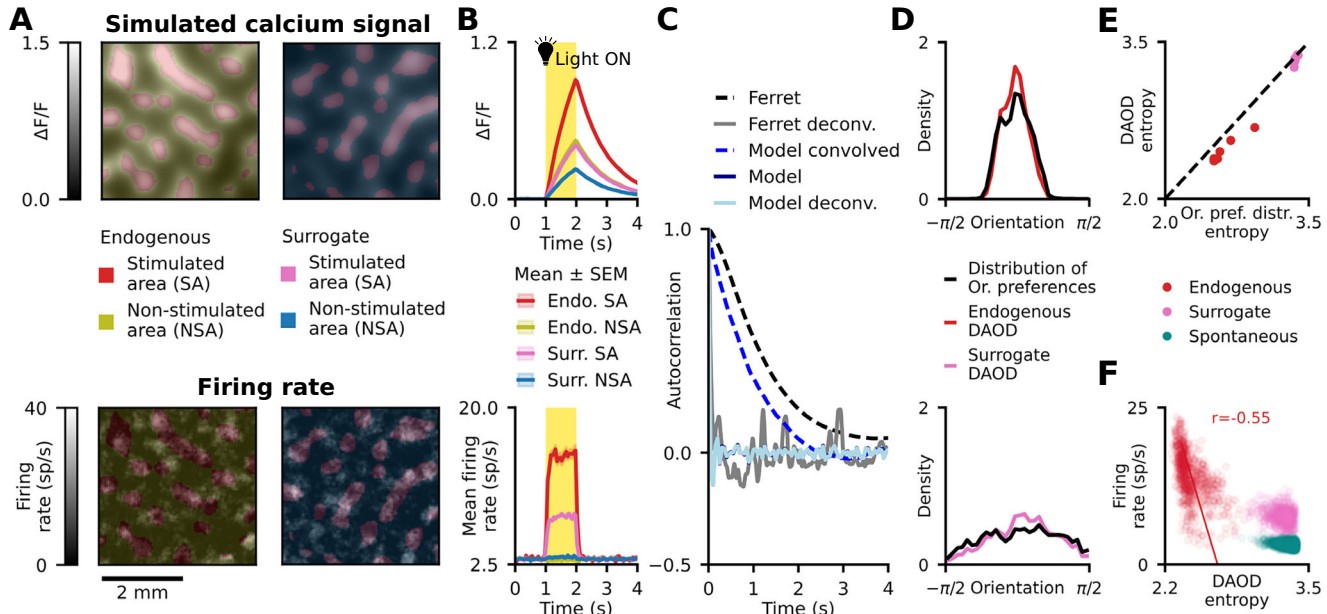

**Fig. 5 | Sharper orientation tuning of optogenetic stimulation responses compared to stimulation patterns. A** Example stimulation patterns (endogenous and surrogate) and corresponding responses at the end of the 1 s stimulation period shown for raw spiking data (bottom) and simulated calcium imaging signal (top). **B** Trial mean of stimulation responses within and outside the stimulated areas, showing the spatiotemporal smoothing effect of the calcium imaging readout. **C** The temporal autocorrelation functions of ferret calcium imaging spontaneous activity signal[34] (black dashed, see "Methods") and model simulated calcium imaging spontaneous activity signal (blue dashed) are of similar timescale. In addition, temporally deconvolved ferret calcium imaging activity (gray) using the Prior Frame Subtraction (PFS) technique is of a similar timescale to model simulated calcium imaging signal, where we omitted convolving with the temporal kernel (dark blue). The autocorrelation functions of raw spiking activity and PFS-deconvolved calcium imaging signal (light blue) are close to identical. **D** The mean

DAOD over the stimulation period and trials is more sharply oriented for both endogenous (red) and surrogate (pink) stimuli, than the distribution of orientation preferences pooled from the orientation map in the stimulated area. The plot shows the DAOD for a single example endogenous/surrogate pattern, see all patterns on Fig. S10. **E** Entropy of mean DAOD is consistently lower than the distribution of orientation preferences entropy across all stimulation patterns, showing that the activity due to stimulation is more sharply oriented than the stimulated area itself. **F** Entropy of DAOD is closely linked with firing rate—activity snapshots with higher activity exhibit sharper activations for both endogenous/surrogate stimulus responses and spontaneous activity (Pearson correlation $r_{endogenous} = -0.55$, $r_{surrogate} = -0.20$, $r_{spontaneous} = -0.27$). The points represent DAOD histograms pooled across all stimuli, trials and stimulation time points. Source data are provided as Source Data files; Table 1 shows experimental data sources.

to stronger, more correlated responses. In support of our hypothesis, removing the orientation bias of layer 2/3 lateral connections reduced the stimulus-response correlation for endogenous stimuli to a level close to that evoked by surrogate stimuli (Fig. 4I, endogenous: $0.7996 \pm 0.004$, surrogate: $0.7097 \pm 0.006$, endogenous no bias: $0.7465 \pm 0.003$; $p_{endo-surr} = 2 \times 10^{-18}$, $p_{endo-endo\_no\_bias} = 5 \times 10^{-16}$, $p_{surr-endo\_no\_bias} = 3 \times 10^{-6}$, Welch's two sample $T$-test: two-sided, assuming unequal variances, $n = 40$ for all three populations).

To confirm that our model captures the optogenetically driven regime also in adult animals, we replicated the combined visual and optogenetic stimulation experiment of Chernov et al.[33], who recorded intrinsic imaging responses in adult macaques. Following Chernov et al., we optogenetically stimulated the horizontal orientation preference column in a circle of 100 μm radius, while simultaneously visually presenting full-field drifting square gratings either iso- or ortho-oriented relative to the stimulated column (Fig. 4J, see "Methods"). We also presented the same gratings without optogenetic stimulation, as the baseline condition. We then identified cortical positions significantly (two-tailed paired $t$-test for dependent samples, $n = 10$ trials, $P < 0.05$) preferring either the orientation of the optogenetically targeted column (iso-regions) or the orthogonal orientation (ortho-regions) (Fig. 4K). We observed that the response in iso-regions increased significantly (two-sided Wilcoxon signed-rank test, $p < 0.006$, $n = 10$) during optogenetic stimulation compared to the visual-only condition (Fig. 4L, left), while the response of ortho-regions decreased ($p < 0.004$, $n = 10$). Chernov et al.[33] observed an analogous iso-activity increase and ortho-decrease of the intrinsic imaging signal when stimulating the vertical orientation column (Fig. 4L, right).

## Orientation specific connectivity is necessary for sharpening of V1 response to optogenetic stimulation

Having access to the underlying spiking responses of all neurons within the $5 \times 5\,mm^2$ section of modelled cortex, we wished to gain deeper insight into how optogenetic stimulation engages the functionally specific cortical network.

First, we noticed that the spatiotemporal blurring effect of the calcium imaging readout underestimates the precision of neural responses in relation to stimulation, both spatially and temporally (Fig. 5A, B). Notably, this temporal smoothing effect of calcium imaging has been recently demonstrated by Kettlewell et al.[34], who deconvolved the ferret calcium imaging spontaneous activity recordings using Prior Frame Subtraction (see "Methods"), revealing faster temporal dynamics. Our model spike data matches the timescale of the example deconvolved signal provided in their article[34], with the first peak of the temporal autocorrelation function occurring at ~226 ms in ferret and ~200 ms in our model (Fig. 5C). Given this alignment, we opted to perform all subsequent analyses directly on model spike responses.

We have shown that model responses to endogenous (orientation preference map correlated) optogenetic stimuli are stronger than to equivalent surrogate stimuli. We hypothesise this is due to orientation-biased recurrent connectivity selectively boosting the activity of the dominantly active orientation, resulting in a sharper orientation tuning profile than would be expected from the underlying orientation preference map. To quantify how the population activity is distributed across the orientation domain during patterned stimulation, we introduced the measure of Distribution of Activity across the

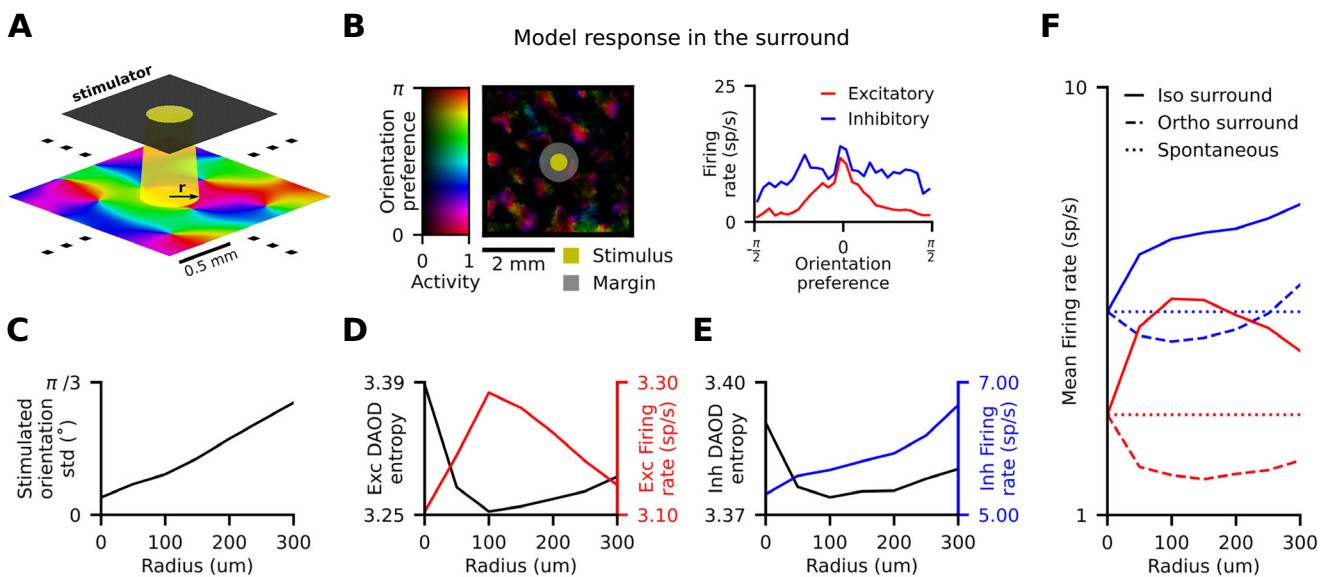

**Fig. 6 | Central stimulation produces functionally specific nonlinear effects in the cortical surround. A** Localised optogenetic stimulation—we stimulate in a small circle with radii increasing from 50 to 300 μm in the middle of the simulated cortical region, targeting a cortical orientation column. **B** (left) Stimulation responses beyond the stimulated area. The central stimulated orientation column has horizontal orientation preference (red colour code). The activity in the surrounding cortex is biased towards iso-oriented areas (red). (right) DAOD histograms for excitatory and inhibitory neurons in the non-stimulated cortex for the depicted frame (with entropies 2.9, 3.3). Activity of the excitatory population is much more sharply tuned than that of the inhibitory population. **C** The range of

orientation preferences under stimulation ROIs increases with the increasing stimulation radius. **D** Excitatory firing rate rises to a maximum and subsequently begins falling, while **E** inhibitory firing rate in the surround scales monotonically with increasing radius. Both excitatory and inhibitory DAOD entropy inversely follows excitatory activation. **F** At the stimulation radius maximally exciting the surround, in iso-oriented regions both excitatory (red, solid) and inhibitory (blue, solid) neurons show increased firing rate compared to spontaneous, while in ortho-oriented regions both excitatory (red, dashed) and inhibitory (blue, dashed) neurons exhibit decreased firing rates. Source data are provided as Source Data files.

Orientation Domain (DAOD). For each frame of activity, we binned cortical positions by their orientation preference, summed the activity within each orientation bin, and normalised the resulting distribution to a unit-integral density function. For each of the 8 endogenous and 8 surrogate stimulation patterns, we calculated DAOD within the optogenetically stimulated regions and averaged it across all frames of the response and 5 trials (see Fig. 5D for example patterns, Supplementary Fig. S10 for all patterns).

To quantify the sharpening of DAODs over the distribution of orientation preferences in stimulated regions, we compared the Shannon entropies of these two distributions (Fig. 5E), with lower entropies indicating sharper tuning in the orientation domain. For all of the 16 tested stimulation patterns (8 endogenous, 8 surrogate), DAOD entropy was lower than that of the distribution of stimulated orientations. The entropy difference between the DAOD and stimulated orientation distribution was significant compared to spatially shuffled controls ($p < 10^{-17}$, two-tailed $Z$-test, $n = 1000$; for all stimulation patterns, see "Methods"). This result supports our hypothesis that the network amplifies the response of the dominant orientation preference subpopulation, resulting in a more sharply oriented response than the orientation distribution of the stimulation pattern, suggesting soft-winner-takes-all dynamics often hypothesised in cortical networks[35].

Finally, we found an inverse relationship between DAOD entropy and firing rate across both spontaneous activity and responses to endogenous/surrogate stimulation—the higher the firing rate, the lower the DAOD entropy (Fig. 5F). This effect was strongest for endogenous stimulation (Pearson correlation $r = -0.55$), but still present with surrogate stimulation ($r = -0.20$) and spontaneous activity ($r = -0.27$). This relationship is also consistent with soft-winner-takes-all dynamics, which would predict recurrent boosting of any spontaneous or externally induced patterns of activity aligned with the network's functional organisation due to the iso-orientation biased connectivity.

## Complex nonlinear interactions across the cortical space underlie population responses to optogenetic stimulation in V1

Experimental and modelling results discussed in the previous section suggest that stimulation patterns aligned with V1 functional organisation are more effective at driving cortical activity than equivalent unaligned patterns, which we hypothesise is underpinned by the functionally specific connectivity in layer 2/3. To understand these network phenomena more mechanistically, we performed simulations focusing on how lateral centre-surround interactions emanating from a single column shape the response to optogenetic stimulation. To investigate this, we stimulated the model in a circular area of variable diameter centred on a central orientation domain (Fig. 6A), and analysed the resulting activations in the surrounding cortical region. We defined the not-directly-stimulated surround as the cortex beyond the largest tested circular stimulation area (300 μm) plus an extra 100 μm margin to exclude neurons potentially directly activated by scattered light (Fig. 6B, left). Crucially, we kept the total luminance input to the cortex constant by rescaling the stimulation intensity for each radius accordingly.

In line with iso-orientation biased long-range lateral connectivity, we found that activated neurons in the surround of the directly stimulated cortex are more likely to have similar orientation preference to that of the stimulated column (Fig. 6B, left). As we increase the radius of illumination, neurons from an increasingly broader range of orientation domains are directly stimulated (Fig. 6C). Yet, the activity and the overall orientation selectivity of the excitatory response in the surround is non-monotonic (Fig. 6D), in contrast with the monotonic increase of inhibitory activity (Fig. 6E). Initially, increasing stimulation radius raises the mean activation of neurons in the surround (Fig. 6D, red line), while decreasing their DAOD entropy (Fig. 6D, black line). This mirrors the relationship between firing rates and selectivity observed in patterned stimulation (Fig. 5F). This initial rise in selectivity is driven by the recurrent

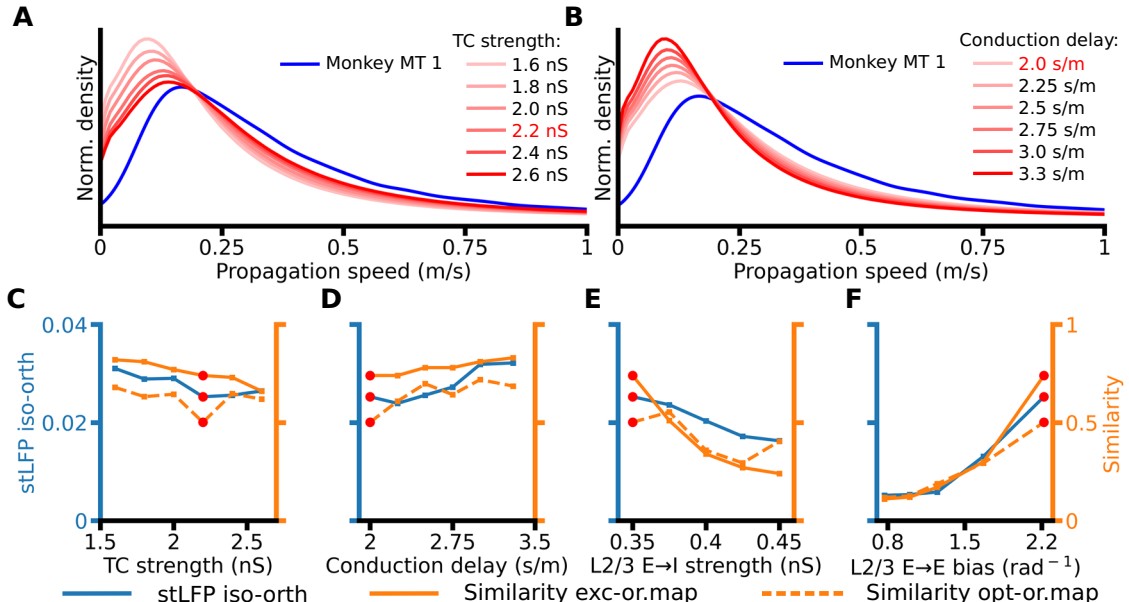

**Fig. 7 | Impact of the parameters on model properties. A** The distribution of propagation speeds for six parametrizations varying in TC strength. Red font in legends indicates the parameter values used in the base model. **B** Same, for six parametrizations varying in intra-laminar distance-dependent conduction delays. **C** For six parametrizations varying in TC strength: (blue) stLFP residuals difference between recording sites with orientation preference congruent to the reference electrode and sites with orthogonal preference; (orange) similarity between the excitatory CMs and the orientation map (solid), and between the uniform full-field optogenetically evoked CMs and the orientation map (dashed). The red circles indicate the parameter values used in the base model. **D** Same, for six parametrizations varying in intra-laminar distance-dependent conduction delays. **E** Same, for five parametrizations varying in L2/3 E→I strength. **F** Same, for five parametrizations varying in L2/3 E→E bias. Source data are provided as Source Data files.

boosting of the dominantly stimulated orientation mediated by the iso-oriented long-range connectivity, and consequent inhibition of the non-dominant orientation domains, in analogous manner to the previous results on patterned optogenetic stimulation (Fig. 4I). Crucially, as the stimulation radius, and hence the range of stimulated orientations increases, both the mean activity level and DAOD in the surround concomitantly flip their trends. This flip cannot be explained by the total input coming from the directly stimulated central area, where the total spiking output of excitatory cells increases monotonically (Fig. S11D, E). Instead, this non-monotonic effect is due to the expanding range of directly stimulated orientations, which gradually diminishes the aforementioned recurrent iso-orientation boosting, leading to reduced activity and less sharp orientation tuning in the surround (Fig. 6D). The precise location of this inflection point is the result of a complex interplay between the spatial scale of the orientation map, the spatial extents of long-range excitatory and short-range inhibitory connections, and balance of excitation and inhibition in the network. Nevertheless, the existence of the inflection point at around 100 μm radius, corresponding to ~0.11 of the model hypercolumn distance, is an important prediction of the model.

We next wished to understand the difference between how the stimulated centre engages populations of iso-oriented vs. ortho-oriented neurons in the surround, which receive substantially different input from the centre. We found that in the iso-oriented regions, inhibitory activity rose continuously, while mean excitatory activity mirrored the non-monotonic profile of the pooled analysis (Fig. 6F, full lines). In contrast, in the ortho-oriented subpopulations, both excitatory and inhibitory responses followed the non-monotonic U-shaped curve (Fig. 6F, interrupted lines). This demonstrates the complexity of intra-cortical interactions that can demonstrate qualitatively different behavior, depending on the relative functional properties between the stimulated and recorded neurons.

## Afferent vs. lateral interactions ratio, conduction delays, and functional connectivity bias impact cortical waves and modular spontaneous activity

To gain understanding on how key components of the model impact the expression of investigated model phenomena, we systematically manipulated model parameters (see "Methods"), while monitoring the various measures introduced earlier. Here we pinpoint the most interesting findings. More exhaustive results of this parameter search can be found in Figs. S12 and S13. Increasing the strength of the afferent thalamo-cortical connections (TC strength) increases the spontaneous activity of the network (Fig. S12A) and the mean propagation speeds of spontaneous waves (Fig. 7A). We hypothesise this is due to the increase of the resting voltage level, which raises the likelihood of fast monosynaptic propagation of the signals over long distances. Because we found that the overall model activity has a strong impact on its dynamics (Fig. 5, S12AEIM), in the following, we dissociate the impact of activity level from the impact of the remaining parameters. We do so by adjusting TC strength for each investigated parameter value combination to match the overall activity level of the base model (see "Methods", Table 2) (Fig. S12B–D).

As expected, the propagation speed of the cortical activity decreases with increasing intra-laminar distance-dependent conduction delays, highlighting its association with horizontal connections (Fig. 7B). Next, in Fig. 7C–F we show that the orientation bias of travelling waves concomitantly changes with both the spontaneous and opto-evoked CMs similarity to the orientation map as we change model parameters, indicating a close association between these phenomena and the lateral connectivity substrate. Specifically, we find that decreasing the relative impact of the lateral vs. feed-forward connectivity (through increasing TC strength) is negatively associated with these three metrics (Fig. 7C). They are also dependent on the E:I balance as increasing the strength of the layer 2/3 excitatory-to-inhibitory connections (L2/3 E → I strength) also induces a decrease in propagation bias and both similarity measures (Fig. 7E). Finally, all

**Table 2 | Pairing of values of TC strength with other parameters modifications**

| Delays | | L2/3 E → E bias | | L2/3 E → I strength | | Cortical density | |
|---|---|---|---|---|---|---|---|
| Delay (s/m) | TC Strength (nS) | Bias (rad⁻¹) | TC Strength (nS) | Strength (nS) | TC Strength (nS) | Density (cells mm⁻²) | TC Strength (nS) |
| **0.2** | **2.2** | 0.77 | 2.4 | **0.35** | **2.2** | 500 | 0.7 |
| 0.225 | 2.1 | 1 | 2.38 | 0.375 | 2.5 | 1500 | 1.6 |
| 0.25 | 2 | 1.25 | 2.36 | 0.4 | 2.7 | 3000 | 2 |
| 0.275 | 1.9 | 1.67 | 2.3 | 0.425 | 2.9 | **5000** | **2.2** |
| 0.3 | 1.75 | **2.22** | **2.2** | 0.45 | 3.05 | 7000 | 2.3 |
| 0.333 | 1.6 | | | | | 10000 | 2.3 |

Emphasis is placed on parameters values used for the baseline version of the model. Source data are provided as Source Data files.

three measures sharply increase as the orientation bias of long-range horizontal connections increases (L2/3 E → E bias, Fig. 7F), underlining the importance of the functional selectivity of the long-range connections for the coexistence of STWs, modular spontaneous activity and optogenetically evoked responses.

## Discussion

In this study, we presented a unified computational model of adult V1 grounded in biologically established mechanisms and anatomical data, that explains an unprecedented range of phenomena across multiple dynamical regimes, including visually driven processing, external optogenetic stimulation and spontaneous emerging activity patterns. The model for the first time reproduces: (i) iso-orientation biased propagation of spontaneous waves, (ii) similarity between excitatory and inhibitory CMs, (iii) modular activity response to uniform full-field optogenetic stimulation that is correlated with modular spontaneous activity, (iv) response ratio inside/outside of optogenetically stimulated area for endogenous stimuli, and (v) the iso-facilitation and ortho-suppression during simultaneous visual and single-column optogenetic stimulation.

Our findings further the hypothesis that functionally specific lateral interactions in cortical layer 2/3 are key to the emergence of modular spontaneous activity, functionally specific propagation of spontaneous cortical waves and functionally specific responses to patterned optogenetic stimulation, bridging previously fragmented experimental observations into a coherent understanding of adult V1 dynamics.

This hypothesis is first corroborated by the finding that preferential propagation of STW into iso-oriented regions is abolished by removing the orientation bias of the long-range lateral connections in the model layer 2/3 (Fig. 2E). Next, spontaneous activity in V1[10] and other cortical areas[36] reflects the underlying functional organisation of the cortical circuitry. This manifests in modular events of higher activity, which are correlated with the feature maps encoded across the cortical surface, such as orientation preference or ocular dominance maps in V1[30]. In the macaque, these events are more frequently aligned with ocular dominance maps[30], while in the ferret, spontaneous activity events aligned with orientation preference are more frequent[10]. In our model, lacking binocular processing, we demonstrate the encoding of orientation maps in spontaneous activity, and show that the long-range iso-orientation biased lateral connectivity in layer 2/3 is necessary for the emergence of this phenomenon (Fig. 3F). Finally, the similarity of the orientation map to both CMs derived from spontaneous activity and from activity evoked by full-field optogenetic stimulation in ferrets[14] suggests that spontaneous and optogenetically evoked activities are governed by the same underlying substrate. Indeed, we find that the iso-orientation bias of the layer 2/3 long-range connections is necessary for the emergence of modular activity in both stimulation conditions (Fig. 4F).

Diving deeper into mechanisms of cortical dynamics, our analysis of the spatial scale of optogenetic stimulation revealed a non-linear relationship between the radius of optogenetic stimulation and cortical response (Fig. 6D–F). We propose that this non-linear relationship is a consequence of the network operating within an inhibitory stabilised regime[37,38]. We have shown that our model operates in this regime under both visual[10] and optogenetic stimulation (Fig. S14), where increased input can paradoxically reduce the activity of both excitatory and inhibitory populations[37,38]. Note that excitatory neurons in regions iso-oriented with the column at the centre of the opto-stimulated area receive stronger long-range input from this column due to the iso-orientation bias of the long-range E-to-E connectivity. In contrast, in ortho-oriented regions, it is the inhibitory neurons that receive relatively stronger input, explaining the simultaneous reduction in the activity of both excitatory and inhibitory neurons in the ortho-oriented populations at the smaller stimulation radii, which effectively drive a narrow range of orientations in the centre. These findings demonstrate that in cortical networks with heterogeneous, functionally specific connectivity, multiple functionally differentiated parts of the network can simultaneously operate in different regimes, leading to highly complex dynamical effects. This highlights the necessity to supplement simplified phenomenological models with simulations taking into account the full complexity of cortical interactions.

The critical assumption of this study is the existence of moderately iso-orientation biased long-range connectivity in layer 2/3 of V1. The strongest evidence for functionally biased intra-areal connectivity in V1 comes from mouse studies using a combination of optogenetics and patch-clamp recordings, which demonstrate a higher likelihood of connections between neurons with similar receptive fields[39,40]. In mammals with columnar organisation of orientation preference, early studies combining intrinsic imaging and anatomical labelling provided evidence for long-range iso-orientation biased connectivity in both cat[22,23] and tree-shrew[41]. More recently, an optogenetics-based study confirmed these findings in ferrets, showing that the average orientation preference of synaptic inputs to layer 2/3 neurons aligns with their orientation preference[24]. Moreover, while highlighting the heterogeneity of long-range connection targets, Martin et al.[42]. show a clear bias toward synapses forming on iso-oriented targets in cat V1 neurons (see Fig. 3, top-right histogram with SI peak above 0.5, indicating an iso-orientation bias). Overall, there is strong evidence across multiple species for a moderate iso-orientation bias in long-range layer 2/3 connectivity.

Several other models aimed at studying some of the individual phenomena we reproduce in our model. STW have been generated in small-scale spiking neural networks with local distant-dependent recurrent connectivity[43,44], but these waves are too dense in comparison to the sparse traveling waves observed in vivo[4]. In a simple model with ring architecture, Muller et al. have shown that the oscillatory activity of the model could propagate laterally only when the length of the model was within a specific range[5]. Davis et al. designed a topologically organised spiking neural network which also exhibited spontaneous traveling waves with realistic wavelengths and propagation speeds properties[15]. Importantly, they showed that the presence of the topologically-organized recurrent connectivity is essential for

the emergence of these properties. In a follow up study, they present an iteration of the model with functionally biased long-range connectivity[9]. They demonstrate that this connectivity scheme is needed to reproduce activity phase patterns associated with functionally biased spiking activity, which they observed in vivo. Their work is complementary to ours as they focus on the functional bias of modulation of spiking activity induced by phase patterns of waves, whereas we focus on the functional bias of the propagation of the traveling waves, with both studies emphasizing the importance of functionally biased connectivity. Unlike our study, however, Davis et al. do not investigate iso-orientation bias of activity propagation, do not consider retino-cortical pathway and hence visual driven regime, and do not investigate optogenetic stimulation.

Next, spontaneous structured activity was modeled by Cai et al.[19] in a single-layer spiking V1 model. Just as our study, Cai et al. assumed iso-orientation biased excitatory long-range lateral connectivity and demonstrated spontaneous events that were dominated by a narrow range of orientations, but did not provide direct quantitative comparison of these spontaneous correlations to experimental data. Interestingly, spontaneous structured activity is present already in young animals before eye-opening[10], before long-range correlations are fully formed[45]. Smith et al. offered a possible resolution to this conundrum by demonstrating that a model with only short range mexican-hat-like interactions can explain long-range correlations in spontaneous activity[10], but their model did not account for afferent processing, and hence could not demonstrate the direct correlation of the spontaneous activity with orientation maps. It is important to emphasize that even if during development long-range connections might not be necessary for inducing long-range correlations, this does not imply this is also the case in the adult cortex. For example, the cortex is much more driven by the afferent pathway in adult animals than in the young, in which visual input is ineffective in evoking reliable neuronal responses[10,46]. This is further corroborated by modeling in Smith et al.[10] that shows that the long-range correlations emerge only if the model dynamics are strongly dominated by lateral interactions, i.e., when the afferent input is about 1/100th of the lateral interaction strength. Under the weaker lateral interactions in the adult condition, the long-range connections might be necessary for the emergence of the long-range correlations in spontaneous activity, as our model suggests.

As with any model, we identify several key limitations, some of which are primary candidates for future work. With the increasing range of phenomena we wish to explain within one model, we face the challenge that data from no animal preparation offer full coverage of phenomena of interest. Thus, our model comparisons involve data from multiple species. While this approach encourages a focus on principles common across mammals with columnar V1, rather than species-specific details, it inevitably limits precise quantitative fits across all measures due to inter-species differences. Nonetheless, our quantitative comparisons reveal consistently good matches across diverse measures, with only occasional moderate deviations, confirming the model operates within a biologically plausible regime. Similarly, due to the lack of suitable data from adult animals, some of the model validations were against studies that pool data from young ferrets recorded on days around eye-opening[10,12–14,20]. While the spontaneous structured activity largely stabilises at eye-opening (see e.g., Figs. 4D, E and 5C, SF2F, SF9BDFH of Smith et al.[10]), justifying the comparison to our adult model, this could explain some of the moderate quantitative discrepancies between our adult model and the ferret data. Next, due to the absence of feedback from higher visual areas, which has recently been shown to influence spontaneous activity[47], we cannot rule out that some of the studied phenomena could not be, at least partially, mediated by extra-areal interactions. The present model also does not consider layers 5 and 6, which, with their extensive lateral arborisation, likely also

contribute to the spatial integration of information in V1. Apical dendrites of layer 5 cells extend up to the supragranular layers and participate in the generation of the LFP signal in these layers. Their absence in the model therefore, impacts the magnitude of the LFP signal and might contribute to the discrepancy between the magnitude of the stLFP residuals of our model and the ones observed in Nauhaus et al. data[8]. Next, although the model also provides opportunity to study the interactions between spontaneous and visually evoked activity and their implications for perception[7,9,21], we have not explored these yet in the present manuscript. We have also presented parameter exploration analysis of a number of key empirically identified model parameters, however given the complexity of the model there is opportunity to broaden the scope to many more parameters, albeit at exceptional computational cost given the size of the model and the sheer range of experimental protocols against which we test it. Similarly, deeper investigation of which model components contribute to which of the studied phenomena would be of great interest. Finally, while we made first steps in that direction in Figs. 5 and 6, the present model offers an unparalleled opportunity for deeper analysis of the dynamics of V1 activity, their relationship to the underlying neural circuits, and the relationship between different cortical driving regimes, that are yet to be exploited.

## Methods
### V1 Model
The V1 model is a direct iteration of our previous modelling effort[21]. It is composed of two layers corresponding to 5 × 5 mm patches of layers 4 and layers 2/3 of parafoveal V1. The new model has higher density, with each layer containing 125,000 excitatory neurons and 31 250 inhibitory neurons. Layer 4 gets its input from a simplified model of LGN covering 6° × 6° of visual field and containing 14 400 neurons (see "Methods": "Retina & Lateral Geniculate Nucleus (LGN)"). Layer 2/3 receives its feedforward input from Layer 4. In each layer, the connectivity is dominated by recurrent connections. The connections in Layer 4 are only local, while the excitatory connections in Layer 2/3 are both local and long-range.

We offer detailed justification of modelling choices in our previous study[21]. Here, we will provide justification only for new or modified parameters (see "Methods": "Modifications from the previous model version").

### Neuron modelling
We simulate cortical neurons with adaptive exponential integrate-and-fire models[48]:

$$\tau_m \frac{dV}{dt} = -(V - E_L) + \Delta_T \exp\left(\frac{V - V_T}{\Delta_T}\right) \\ + R_m g_{exc}(E_{exc} - V) + R_m g_{inh}(E_{inh} - V) - w \tag{1}$$

Every time the membrane potential of a neuron crosses a threshold of −40 mV, a spike is generated and the membrane potential of the neuron is reset to a value $V_r = -70$ mV for a refractory period of 2 ms and 0.5 ms for excitatory and inhibitory neurons respectively. $\tau_m$ is the time constant of the membrane and is equal to 8 ms for excitatory neurons and to 9 ms for inhibitory neurons. The leak potential $E_L$ and the spike initiation threshold $V_T$ are set respectively to −80 mV and −57 mV for excitatory neurons, and to −78 mV and −58 mV for inhibitory neurons. The threshold slope factor $\Delta_T$ of the exponential function is equal to 0.8 mV. The membrane resistance $R_m$ is equal to 250 MΩ for excitatory neurons and to 300 MΩ for inhibitory neurons. The excitatory and inhibitory reversal potentials $E_{exc}$ and $E_{inh}$ are respectively set to 0 mV and −80 mV. The adaptation current $w$ is further detailed in Table S2.

**Table 3 | Parameters of the hyperbolic distributions used to generate the local connectivity of the model**

| | $\alpha$ (µm$^{-1}$) | | | | $\theta$ (µm) | | | |
|---|---|---|---|---|---|---|---|---|
| | **L4 Exc** | **L4 Inh** | **L2/3 Exc** | **L2/3 Inh** | **L4 Exc** | **L4 Inh** | **L2/3 Exc** | **L2/3 Inh** |
| L4 Exc | 0.0139 | 0.0148 | 0.0174 | 0.0197 | 207.7 | 191.8 | 154.4 | 131.5 |
| L4 Inh | 0.0126 | 0.0119 | | | 237.5 | 256.4 | | |
| L2/3 Inh | | | 0.0149 | 0.0150 | | | 189.5 | 188.64 |

Rows refer to pre-synaptic populations, and columns to post-synaptic populations. Source data are provided as Source Data files.

## Retina & lateral geniculate nucleus (LGN)

ON and OFF-center LGN neurons are modelled as leaky integrate-and-fire cells. They are driven by injected current computed by convolving stimuli in the model's visual field with a 3D spatiotemporal receptive field kernel based on LGN measurements by Cai et al.[49]. This convolution is subsequently scaled by contrast and luminance gain to match experimental observations[50,51] of contrast and luminance saturation, using separate Naka-Rushton functions $\alpha_c r/(1.0 + \beta_c r)$ and $\alpha_l r/(1.0 + \beta_l r)$, respectively, where $r$ denotes the convolution result, $\alpha_c$, $\alpha_l$ are the gain and $\beta_c$, $\beta_l$ are the saturation parameters. The baseline firing rate of LGN neurons is achieved by injecting Gaussian noise current into the neurons, such that the baseline firing rates of both ON and OFF LGN neurons match experimentally observed values[51]. Due to the relatively small region of visual space our model covers, we do not model the systematic changes in RF parameters with foveal eccentricity and thus assume that all ON and OFF LGN neurons have identical parameters.

## Thalamo-cortical connections

Every neuron in Layer 4 receives an input from the LGN. The spatial pattern of the thalamo-cortical connectivity is based on a template that is the sum of a Gabor distribution and of a 2 dimensional Gaussian.

$$g(x, y, \lambda, \theta, \psi, \sigma, \gamma) = \exp\left(\frac{x'^2 + y'^2 \gamma^2}{2\sigma^2}\right)(G + \cos(2\pi x'\lambda + \psi))$$
$$x' = x\cos\theta + y\sin\theta$$
$$y' = -x\sin\theta + y\cos\theta$$

(2)

Where G represents the relative weight of the Gaussian, and is set at 0.085. In all sheets neurons are populated at random physical positions that correspond to retinotopic positions assuming a magnification factor of 1 between LGN and cortex, corresponding to retinotopic eccentricity parafoveally of roughly 3°. The coordinates x and y represent the retinotopic position of the LGN neurons relative to the retinotopic center of the layer 4 cell. The phase preference $\psi$, is generated randomly from an uniform distribution with support $[0, 2\pi)$. The orientation of the Gabor distribution $\theta$ is determined by the position of the neuron on a pre-computed orientation map (Fig. 1A). We set the aspect ratio $\gamma$ of the Gabor distribution to 2.5, the spatial frequency $\lambda$ to 0.8 cycles per degree, and the size $\sigma$ to 0.17°. To generate connections from LGN cells to a given layer 4 neuron, we take the absolute value of the positive (ON cells) or negative (OFF cells) part of the template, renormalize it to form distribution, and sample from it the respective connections to ON or OFF cells.

Individual layer 4 excitatory and inhibitory cells receive connections from the LGN, whose number is drawn from uniform distributions of respective support of [90, 190] and [112, 168]. A limitation of the model is that we do not model the dominance of the OFF pathway and systematic variations of the ON-OFF pattern of RF following the orientation maps revealed in recent electrophysiological experiments[52,53].

## Cortico-cortical connectivity

Excitatory neurons in layer 4 receive 1000 synapses, whereas layer 2/3 neurons receive 2300 synapses. Inhibitory neurons receive 40% less connections than their excitatory counterparts. Cortical neurons receive excitatory and inhibitory connections following a ratio of 4:1. Layer 2/3 neurons receive 22% of their excitatory input through the feedforward connection originating in Layer 4. Furthermore, layer 4 neurons receive a direct feedback connection from Layer 2/3 corresponding to 20% of their total excitatory inputs, while the remaining excitatory synapses are formed by intra-laminar connections. Although there is little evidence of such direct connections from supragranular layers to layer 4, an important feedback projection from supragranular layers reaches layer 4 via infragranular layers. Because we believe that it is important to close this cortico-cortical loop and as infragranular layers are absent in the model, we have therefore decided to create this direct projection from layer 2/3 to layer 4 in the model. Removing this feedback projection doesn't significantly impact the results presented in this study (Fig. S15).

Recurrent connectivity in Layer 4, inhibitory connectivity in Layer 2/3, and excitatory feedforward connectivity from layer 4 to layer 2/3 are only local and are distance-dependent. For each of these types of projections, the probability distribution of connections between pairs of neurons is given by the following equation:

$$\text{pdf} = p_{dist} p_{func}$$

(3)

Where $p_{func}$ and $p_{dist}$ are respectively the functional and spatial components of the probability of connection. For these connections, $p_{dist}$ follows a hyperbolic distribution:

$$p_{dist}(d) = \exp\left(-\alpha\sqrt{\theta^2 + d^2}\right)$$

(4)

where $d$ is the lateral distance between the two neurons, and $\alpha$ (µm$^{-1}$) and $\theta$ (µm) are the parameters shown in Table 3 which were obtained by fitting connection probability data (see ref. 21 for details) from Stepanyants et al.[54].

For recurrent connectivity in layer 4, the functionally specific component is modelled as a push-pull scheme, with excitatory connections being more likely to occur between neurons with correlated afferent receptive field templates (see "Methods": "Thalamo-cortical connections"), and with inhibitory connections biased towards neurons with anti-correlated receptive fields. A correlation c is computed between their receptive fields, here defined solely based on their afferent inputs from the LGN. This functional component is then the following:

$$p_{func\_l4}(c) = \frac{1}{\sigma\sqrt{2\pi}} e^{\frac{-(c-\mu)^2}{2\sigma^2}}$$

(5)

With $\sigma = 1.3$, and $\mu = 1$ or $-1$, respectively, when the pre-synaptic neuron is excitatory or inhibitory.

For inhibitory connections in layer 2/3 there is no functional component and $p_{func\_L23\_inh}$ is therefore equal to 1 in Eq. (3). Feedforward projections from layer 4 to layer 2/3 are biased for connecting neurons with similar orientation preferences. Their functional

component is:

$$p_{func\_I4 \rightarrow I23}(\Delta o) = \frac{1}{\sigma\sqrt{2\pi}} e^{\frac{\Delta o^2}{2\sigma^2}} \tag{6}$$

Where $\Delta o$ is the difference of orientation preference between the pre-synaptic and post-synaptic cells based on the pre-computed orientation map and with a standard deviation $\sigma$ of 1.3 radians. The feedback connections from layer 2/3 to layer 4 are modelled in the same way as in Eq. (3). Their functional component has the same formula as Eq. (6) with $\sigma$ is equal to 1.3 and 3 radians when targeting respectively excitatory and inhibitory neurons. Their spatial component is Gaussian:

$$p_{dist\_I23 \rightarrow I4}(d) = \exp\left(-\frac{d^2}{2\sigma^2}\right) \tag{7}$$

with $\sigma = 100\,\mu m$. Finally, recurrent excitatory connections in Layer 2/3 consist of both local connections and long-range horizontal projection, the latter being orientation biased (see Supplementary Fig. S16). We model these connections with two spatial components as in Buzás et al.[23]:

$$pdf_{I23} = p_{loc\_I23} + \alpha p_{lr\_I23} p_{funct\_I23} \tag{8}$$

where $p_{loc\_I23}$ and $p_{lr\_I23}$ represent the local and long-range spatial component, and follow the same formula as in Eq. (7) with $\sigma$ respectively equal to $270\,\mu m$ and $1000\,\mu m$. The functional factor $p_{funct\_I23}$ has the same formula as in Eq. (6) with $\sigma$ equal to 0.45 and to 1.3 radians for connections targeting respectively excitatory and inhibitory cells. $\alpha$ defines the relative weight of the long-range component and is equal to 4.

### Synapses

All synapses of the model are modelled with an exponential decay according to the following formula:

$$g_{syn}(t) = \bar{g}_{syn} e^{\frac{-(t - t^*)}{\tau_{syn}}} \tag{9}$$

Where $t^*$ is the spike time and $\tau_{syn}$ is the time constant of the exponential decay and is equal to 1.5 ms for excitatory synapses and to 4.2 ms for inhibitory synapses. $\underline{g}_{syn}$ represents the weight of the synapses and differs according to the type of connection as shown in Table S1C

Every synapse of the model also exhibits short-term plasticity through the Tsodyks-Markram model[55], with parameters shown in Table S3. The delays of the intra-cortical connections of the model are implemented as the sum of two components, one constant and one distance-dependent. The propagation constant of the distance-dependent component is equal to 0.2 s/m for all intra-cortical connections. The constant delays are 1.4 ms and 0.5 ms for excitatory connections targeting respectively excitatory and inhibitory neurons, whereas inhibitory connections have constant delays equal to 1 ms. The delays of thalamo-cortical connections are drawn randomly from an interval of [1.4, 2.4] ms.

### Modifications from the previous model version

The following changes from the previous model version[21] were made: The density was increased from 1500 neurons to 5000 neurons per square millimeters to reduce the downsampling of the number of neurons in the model relative to biological densities. This density roughly represents a one fifth downscaling relative to the neuronal density in cat V1[56], which was necessary for our computational simulations to run in a tractable amount of time and memory. Previous

studies have shown that both properties of waves[15] and correlation of activity[57] are negatively affected by downscaling the neuronal density. We find that downscaling indeed affects the properties we investigated, but that this effect saturates for a downscaling by a factor 5 or less, showing that the density used in this study represents a good compromise between fidelity and computational efficiency (Fig. S17).

To match the activity regime of the experimental data of Davis et al.[15] (see Fig. 2) and to compensate the reduction in spontaneous activity caused by the higher density, the strength of the thalamo-cortical connections was increased from 1.2 to 2.2 nS and the spike initiation threshold was decreased from −58 to −56 mV for inhibitory neurons. Furthermore, to compensate the excess of evoked activity that these changes created, we increased the strength of the inhibitory synapses from 1 to 2 nS as well as reduced their short-term plasticity recovery time constant in L4 from 70 to 30 ms, such that it now matches the other recurrent cortical connections, hence reducing the number of model parameters. Additionally, we have increased the strength of the recurrent and feedback excitatory-to-inhibitory connections targeting layer 4 from 0.22 to 0.35 nS so that it matches the strengths of their equivalent in layer 2/3 further simplifying the model. Unfortunately, the effective synaptic strength has not been accurately determined experimentally at such a layer-to-layer and neuron type to neuron type level of granularity. We therefore tuned these parameters manually to achieve desired dynamical properties of the cortical network, but such that they remain within physiologically realistic ranges.

We have also decreased the standard deviation of the orientation bias of the layer 2/3 excitatory-to-excitatory connections long-range component from 1.3 radians to 0.45 radians, therefore making them more tuned to orientation in comparison to the previous model iteration. We found this increased bias was necessary to quantitatively accurately capture the functional propagation bias of STW and the alignment of the modular spontaneous activity on the network underlying functional organisation (see Fig. 7F). Unfortunately, the exact strength of the orientation bias was never experimentally quantified due to the complexity of designing such experiments, but has been qualitatively demonstrated in several studies[23,58]. We have also decreased the conduction delays from 3.3 to 2 s/m, taking it closer to the mean value of 1.67 s/m but further away from the median value of 3.3 s/m observed in macaques that we used previously[59].

### Data analysis

Quantitative values in this article are expressed as mean ± SEM.

### Coefficient of variation of the inter-spike-interval

For each neuron, we computed the CV of its inter-spike-interval (ISI) as:

$$CV_{ISI}^2 = \frac{Var[ISI]}{\langle ISI \rangle^2} \tag{10}$$

### LFP signal

To synthesise LFP signal from our point neuron population of L2/3 excitatory neurons, we used a proxy method that approximates an LFP signal as the sum of the absolute values of the excitatory and inhibitory currents to estimate the participation of single neurons in the LFP signal of the population[26].

$$LFP(t) = \sum_i^n |I_{exc}^{(i)}(t)| + \sum_i^n |I_{inh}^{(i)}(t)| \tag{11}$$

$I_{exc}$ and $I_{inh}$ represent, respectively, the input excitatory and inhibitory currents. We computed the LFP signal in a $100 \times 100$ grid with 50 μm spacing representing the $5 \times 5$ mm area of the model, with each portion of the grid corresponding to the signal generated

independently by a pool of $n = 12.5$ neurons in average. We then normalised the LFP signal to transform it into a z-score.

To simulate the spatial reach of the LFP, we convolved our signal with a 2D Gaussian with a standard deviation of 250 μm based on the previous modeling study of Davis et al.[15]. A modeling study from Lindén et al. has estimated that without any input correlations the spatial reach of the LFPs is around 200 μm, corresponding to roughly half the gaussian kernel size used in this study, but found that this value increases with the level of input correlations[60]. For the low level of pairwise correlations in our model of 0.013 (Fig. S1C), Lindén et al. find that the spatial reach of LFP starts increasing by a few hundreds of micrometers. Therefore, the choice of 250 μm as the standard deviation of the Gaussian kernel is a plausible estimate in our context.

Ultimately, the current understanding allows only for rough choice of the kernel width. We therefore verified the impact of the kernel width on the key metrics of the spontaneous waves in V1 for which the synthetic LFP is used, that we present in Fig. S18. We find that halving the kernel width moderately but significantly reduces the estimated wavelength and speed of the spontaneous waves, while the magnitude of the iso-orientation propagation bias increases. Doubling the kernel width has the opposite effect on all metrics.

## Wave properties analysis

We applied the Generalised Phase (GP) analysis[7,15] on the computed three dimensional LFP signal of the $4 \times 4$ mm central portion of the model. We first applied a band-pass Butterworth filter (5–100 Hz, 4th order, forward-backward) on the computed LFP signal. Our results are however not significantly affected if using a narrower band of frequency in the filtering (Fig. S19). We then computed the analytic signal after applying an Hilbert transform, based on fast Fourier transform, on the wideband filtered LFP signal. The phase $\phi$ of the signal was computed as the argument of the analytic signal. The instantaneous frequency of the signal was calculated as the time derivative of its phase:

$$f_{\text{inst}} = \frac{\partial \phi}{\partial t} \tag{12}$$

As high frequencies in the signal induce sequences of $n_t$ points with negative frequencies, we found these data points with negative instantaneous frequencies and performed a cubic interpolation along the temporal dimension on the phase of these $n_t$ as well as on the phase of the next $2n_t$ time points.

The wavelengths of the signal were computed as the inverse of the magnitude of the spatial gradient of the phase of the signal at each time and space points:

$$\lambda_{\text{LFP}} = \frac{1}{\left| \nabla_{x,y} \phi \right|} \tag{13}$$

We then computed the propagation speed of the signal as the product of the wavelengths and of the instantaneous frequency of the signal:

$$\upsilon_{\text{prop}} = \lambda_{\text{LFP}} f_{\text{inst}} \tag{14}$$

## Spike-triggered LFP

We computed the spike-triggered LFP (stLFP) based on the spiking activity of 207 reference excitatory neurons in the Layer 2/3 of the model following the procedure of Nauhaus et al.[8]. Every reference neuron was located in the central $4 \times 4$ mm portion of the model. For each reference neuron, we then computed the stLFP as the spike-triggered average of the LFP signal over a $13 \times 13$ sub-grid of the original LFP grid, keeping a cell size of $50 \times 50$ μm but with now 400 μm

of spacing. We then fitted the relationship between stLFP peak amplitude and the distance between the neuron and the corresponding electrode with an exponential (see Fig. 2C). We discarded from further analysis 2 neurons for which the explained variance of the fitting was lower than 0.5. We then computed the stLFP residuals by subtracting the expected stLFP amplitude (determined by the exponential fit) at each electrode from the actually measured amplitude. We assigned an orientation preference to each recording site based on the pre-computed orientation map of the model. Next, for each reference neuron, we binned the residual stLFP of the recording site based on the difference between their assigned orientation preference and the orientation preference of the neuron. We then averaged the stLFP residuals in each bin across recording sites and seed neurons to compute the orientation bias of the stLFP residuals.

## Simulated calcium imaging signal

To facilitate comparisons with calcium imaging experimental data, we constructed a synthetic calcium imaging signal from our model recordings (see Supplementary Fig. S6). We recorded model spikes on a $4 \times 4$ mm grid with 40 μm spacing and binned them into 50 ms time bins, forming the basis of our neural signal. We next convolved the neural signal at each spatial position with a temporal kernel[61] that characterises the exponential decay of calcium fluorescence following spikes ($\tau = 1$ s):

$$K(t) = e^{-\frac{t}{\tau}} \tag{15}$$

Next, to model light absorption and scattering of the fluorescence signals in the cortex, we convolved the signal with a spatial kernel generated using a beam-spread function model[62]. Fluorescence was modelled as isotropic emission from a disk of 20 μm diameter (representing the neuron) at a depth corresponding to the superficial region of cortical Layer 2/3 (150 μm). Due to the lack of available scattering data for the ferret cortex, we based our coefficients on experimental measurements in mouse cortex[62]. Finally, we applied a spatial Gaussian bandpass filter to our signal ($\sigma_{\text{low}} = 26$ μm, $\sigma_{\text{high}} = 200$ μm), to match the experimental calcium image processing pipeline of ref. 10.

For simplicity, we omitted the screening for large-scale events used by Smith et al.[10], as it had minimal impact on results (see Supplementary Fig. S7).

## Correlation maps (CMs) and CM similarity

We constructed CMs from the synthetic calcium imaging signal by calculating pairwise correlations across time between all pairs of spatial positions following Smith et al.[10]. This process resulted in a correlation map for each spatial position, effectively capturing the spatial organisation of spontaneous cortical activity.

Next, to quantify the similarity of each correlation map (CM) to the orientation map, we computed the magnitude of the pairwise correlation coefficient between the CM and the real and imaginary components of the orientation map (Smith et al.[10]). Specifically, for a selected "seed" cortical position s, the similarity Sim(s) was calculated as follows:

$$\text{Sim}(s) = \sqrt{\text{corr}_x(\Re(O(x)), C(s,x))^2 + \text{corr}_x(\Im(O(x)), C(s,x))^2} \tag{16}$$

Here, Sim($s$) represents the similarity at position $s$, C($s,x$) is the correlation value at spatial position $x$ of the CM with seed $s$, and O($x$) is vector-summed orientation map at spatial position $x$.

We generated the chance level orientation map similarity distribution for our neural signal by calculating the synthetic calcium imaging signal for uniform random noise (with equivalent space and time dimensions as our spontaneous activity recording) and

calculating the correlation map similarity for each spatial position in this recording.

Finally, we calculated the similarity between two distinct CMs with the same seed position as the Pearson's correlation coefficient between the two maps. To prevent local correlations from inflating the similarity between the networks, we excluded all pixels within a radius of 400 μm around the seed position in both maps. We repeated this process for each seed position, replicating the analysis of Mulholland et al.[20].

### Determining orientation preference with Self-Organizing Maps

To reconstruct the orientation preference map of the model cortex, we used a circular Self-Organizing Map (SOM) with 40 nodes, analogously to Kenet et al.[29]. We supplied the set of all CMs to the SOM as input features, and ran the algorithm for 1000 iterations using the publicly available pip *som-pbc* package. We subsequently assigned an orientation to each node in the SOM with a step of 4.5° (180°/40), and estimated the orientation preference of each spatial position by standard vectorial summation.

Since this algorithm only assigns orientations in a circular topology, but cannot recover the correct phase and direction of the assignment relative to the orientation map, we evaluated the mean circular distance between the estimated orientations and the true orientation map for both directions and 1000 phase shifts from 0 to π. We subsequently selected the phase and direction with the smallest circular distance.

### Dimensionality of spontaneous activity, correlation wavelength, local correlation eccentricity

We calculated all 3 metrics following Mulholland et al.[20]. We estimated the **dimensionality** (also known as participation ratio[63-65]) of spontaneous activity patterns as follows:

$$d_{\text{eff}} = \frac{\left(\sum_{i=1}^{N}\lambda_i\right)^2}{\sum_{i=1}^{N}\lambda_i^2} \tag{17}$$

Here, $\lambda_i$ are the eigenvalues of the covariance matrix for the pixels in each frame. Similarly to Mulholland et al.[20], we randomly sub-sampled spontaneous activity frames ($n = 30$ frames, 100 simulations), and took the median of the distribution. We omitted the event detection step used by Mulholland et al., as it had minimal impact on our results (see Supplementary Fig. S7).

To estimate the **wavelength of CMs**, we centered and averaged the CMs in a 1500 μm radius across all seed points. To obtain the average correlation as a function of distance, we then calculated the radial mean with respect to the center point. The wavelength of the CMs was then defined as the distance to the first local maximum, after the maximum at 0.

To estimate **local correlation eccentricity**, describing the shape of the local correlation pattern around a seed point, we fit an ellipse (with axes $\zeta_1$, $\zeta_2$) to the contour line at correlation 0.7 around the seed point. The eccentricity of the ellipse was subsequently calculated as:

$$\epsilon = \frac{\sqrt{\zeta_1^2 - \zeta_2^2}}{\zeta_1} \tag{18}$$

Here, $\epsilon = 0$ denotes a circle, and increasing values (up to 1), reflect the increasing elongation of the ellipse.

### Prediction interval

To determine the range of values in Figs. 3H and 4E into which new experimental observations would likely fall, we calculated the 95% prediction interval from the data of each metric. The prediction interval for data generated from a Gaussian distribution with unknown mean and variance is given as follows[66]:

$$\text{PI} = \mu \pm T_a s_n \frac{\sqrt{n+1}}{\sqrt{n}} \tag{19}$$

where $\mu$ denotes sample mean, $s_n$ sample standard deviation, $n$ the number of points, $T_a$ the $100((1-p)/2)$th percentile of Student's t-distribution with $n-1$ degrees of freedom and $p = 0.95$ the confidence level.

### Single-condition orientation preference map

To construct single-condition orientation preference maps following Kenet et al.[29] we visually stimulated the model cortex with high-contrast full-field sinusoidal gratings of 8 orientations over 10 trials. We then generated frames of model cortical activity by binning model spikes on a $4 \times 4$ mm grid with 40 μm spacing, into 50 ms time bins, as substitutes for the Voltage-Sensitive Dye Imaging (VSDI) frames recorded by Kenet et al. The single-condition orientation preference maps for each orientation were then computed as the mean over trials of the 50 ms frame from 100 to 150 ms after stimulus onset, matching the analysis window of Kenet et al. (mean over 5 frames, 100–150 ms after stimulus onset).

Following Kenet et al., we generated control maps by horizontally flipping the 8 single-condition maps.

The full-field sinusoidal gratings were of 2 s duration, 100% contrast with 50 cd/m² background luminance, 0.8 cyc/° spatial frequency, 2 Hz temporal frequency. We used 10 trials per orientation (compared to 33 in Kenet et al.), due to the lack of measurement noise in our model.

### Preferred cortical state

In the study of Tsodyks et al.[31], the PCS of each neuron was calculated by averaging VSDI frames at time points corresponding to the neuron's spike times while presenting drifting grating stimuli. In our analysis, we substituted the VSDI signal frames with model spikes on a $4 \times 4$ mm grid with 40 μm spacing, binned into 50 ms time bins. Spontaneous activity frames were computed using the same binning procedure and correlated with the PCS of all Layer 2/3 excitatory neurons using Pearson's correlation coefficient.

### Timescale of spontaneous activity

We estimated the temporal scale of spontaneous activity in both model and ferret by calculating temporal autocorrelation functions of population activity. For model data, we averaged 120 s of raw spiking and simulated calcium imaging signal across space to arrive at population activity.

For ferret data, we extracted the time courses of calcium imaging and deconvolved calcium imaging population activity from Kettlewell et al.[34]. Fig. 1a, c, and calculated the temporal autocorrelation functions from the extracted traces. As the available deconvolved time course spanned only 4 s, the autocorrelation features on longer timescales (>0.5 s) might be less reliable. As such, when comparing the timescales of simulated and ferret spontaneous activity, we used the position of the first autocorrelation peak as our metric.

To verify that our non-convolved model data was indeed equivalent to deconvolved calcium imaging traces, we used the Prior Frame Subtraction method described by Kettlewell et al.[34], where each pixel value in each time point was calculated as follows:

$$y_{p,t} = r_{p,t} - \gamma r_{p,t-1} \tag{20}$$

Here, $r_{p,t}$ denotes the calcium imaging signal value at pixel $p$ and time $t$, $y_{p,t}$ denotes the deconvolved value, and $\gamma = 0.95$ denotes the calcium imaging temporal kernel decay rate over 50 ms (frame duration).

## Distribution of activity across the orientation domain (DAOD)

To calculate Distribution of Activity across the Orientation Domain (DAOD) histograms for each frame of activity during the 1-s of simulated optogenetic stimulation we assumed orientation preference binned into 30 intervals (ranging from 0 to π). We added the activity at each cortical position into one of these bins based on the orientation preference at that cortical position, and normalised the binned activity to an integral of 1 to form a distribution of activities across the orientation domain. In some cases (Fig. 5D–F), we only included in the analysis positions within a given stimulation pattern's ROI.

To determine if the Shannon entropy difference between the distribution of sampled orientations and the corresponding DAOD is statistically significant, we tested the null hypothesis that the activations from which the DAOD is constructed do not depend on spatial position and thus orientation preference. To do so, we created a population ($n = 1000$) of control DAOD distributions by spatially shuffling the activations from which the DAOD is calculated. Next, we calculated the absolute difference between the entropy of each control DAOD and the entropy of the distribution of sampled orientations, forming a control distribution. Finally, we used a two-tailed $Z$-test to determine if the absolute entropy difference between the distribution of sampled orientations and the corresponding DAOD is significantly different from the control distribution.

## Parameterizations used for studying the impact of model parameters

We modified the following five parameters of the model to explore how they influence the phenomena we investigate in this study: (i) the strength of the afferent thalamo-cortical connections (TC strength), (ii) the cortico-cortical connections distance-dependent conduction delays, (iii) the orientation bias of the layer 2/3 intra-laminar excitatory-to-excitatory connections (L2/3 E → E bias) computed as the inverse of the standard deviation used to derive the functional component in the layer 2/3 excitatory-to-inhibitory connectivity (see "Methods": "Cortico-cortical connectivity"), (iv) the strength of the layer 2/3 intra-laminar excitatory-to-inhibitory connections (L2/3 E → I strength), and (v) the cortical density of neurons. As parameters (ii–v) impacted the overall activity of the model, we paired these modifications with a change in TC strength to match the mean activity in the model for each model parameterization:

## Optogenetic stimulation

To simulate the effects of optogenetic stimulation on the V1 model, we utilised a model of BMI setup we developed recently[32]. Figure S8 summarises the five-tier structure of this optogenetic stimulator construct:

1. Stimulation patterns
2. 2D stimulator array of identical circular light sources at the cortical surface
3. Light propagation in the cortical tissue
4. Illumination-dependent ChrimsonR channelrhodopsin dynamics in transfected cells
5. Large-scale model of the primary visual cortex.

Since the model of the primary visual cortex is detailed in "Methods": "V1 Model", we will only expand on steps 1–4 in this section.

## Stimulation patterns

We define the region-of-interest (ROI) as the cortical area located beneath each stimulation pattern, with uniform stimulation intensity across active pixels in each pattern. We present 3 types of stimuli to the model cortex:

1. Full-field stimulation—ROI covers the entire array.

2. Endogenous patterns—ROI corresponds to regions in the correlation map (CM) surrounding its local maxima, resulting in orientation-specific stimulation (see Fig. 4, pattern construction details below).
3. Surrogate patterns—equivalent stimulation patterns to endogenous patterns but uncorrelated with the orientation map.

With the goal of matching the stimulation protocols of Mulholland et al.[12,13] as closely as possible, we create the patterned stimuli as follows:

1. CM selection: We select CMs with seeds at positions with 8 equidistant orientation preferences (0, 22.5, 45, 67.5, 90, 112.5, 135, 157.5°)—this ensures a variety of different stimulation patterns all following intracortical structure.
2. Endogenous ROIs: We define the ROIs as the 25% highest correlation value pixels in each map. This is in line with the stimulation protocols of Mulholland et al.[12], who manually selected stimulation ROIs around CM maxima, resulting in approximately 25% of the cortical surface being stimulated (24.27% based on scraped data from Mulholland et al.[12] Fig. 5I).
3. Surrogate ROIs: we rotate endogenous patterns by 90°, resulting in stimuli uncorrelated with the orientation map (mean orientation map similarity = 0.1 as opposed to 0.54 for endogenous stimuli).

To replicate the relative intensity difference used by Mulholland et al.[12,13] between full-field (10 mW) and patterned (7.6 mW) stimulation, we set the intensity of patterned stimulation to 76% of that used for full-field stimulation.

## Stimulator array

We stimulate our in-silico cortical surface with a $4 \times 4$ mm array of 10 μm diameter circular light sources, centred on the cortex. We only stimulate layer 2/3 as the light intensity falloff from the surface is exponential (see Fig. S20), resulting in the amount of light reaching layer 4 to be negligible given the assumed illumination intensities. The light intensity at the cortical surface across the array is $3.0 \times 10^{14}$ photons/s/cm².

## Light propagation in cortical tissue

We simulate the propagation of light in the cortical tissue using the simulations of the "Human Brain Grey Matter" model implemented in the LightTools software, assuming emitted light at 590 nm. The scattering and absorption properties of brain tissue are calculated using the Henyey-Greenstein model[67]. Based on the work of Jacques et al.[68] we set the two key parameters of the model, $g$ (anisotropy factor) and $MFP$ (mean free path) to 0.87 and 0.07 mm, respectively. The simulation results were tabulated into a 2D lookup table of light flux values at depth $d$ and radial distance $r$ relative to the centre of the light source to speed up simulations.

## Channelrhodopsin model

To simulate the dynamics of the ChrimsonR channelrhodopsin in response to 590 nm light stimulation, as used in the experiments of Mulholland et al.[12–14], we utilize the model of Sabatier et al.[69]. The electro-chemical behaviour of the ChrimsonR protein is modelled using a 5-state Markov kinetic model[70], representing the conformations the protein can adopt. Transitions between these states, which the real system can chemically switch between, each have their own time constant and can be either thermally or photo-induced. Thermal transition time constants are fixed, while photo-transition time constants vary with light intensity. These transition time constants describe the linear differential equations governing the evolution of the proportion of channels in each state. The total conductance of all channels in a single neuron is then derived from the sum of

conductances of open channels. We use model parameters fitted to 590 nm light stimulation experiments by Sabatier et al.[69] on ChrimsonR-expressing HEK293 cells.

### Combined optogenetic & visual stimulation

For the comparison of visual-only and visual+optogenetic stimulation protocols, we presented full-field square gratings, matching the visual stimulus parameters reported by Chernov et al.[33]: 4 s duration, high contrast (100%, 50 cd/m$^2$ mean luminance), 4 cyc/s temporal frequency and 1 cyc/° spatial frequency. As noise levels when recording from our model cortex were much smaller than with in vivo intrinsic imaging, we used $n = 10$ trials, compared to $n = 40$ of Chernov et al. For the visual + optogenetic condition, we used the same visual stimuli and additionally optogenetically stimulated the orientation column at the center of our model cortex for the first 600 ms of stimulus presentation within a circular region ($r = 100$ μm), matching Chernov et al. Trials were separated by 4 s of blank stimuli, and were presented in succession.

Intrinsic imaging signals reflect slow vascular responses, peaking several seconds after neural activation depending on stimulus duration, and returning to baseline 4–6 s after stimulus cessation[71]. As such, the 4 s visual + opto intrinsic signal response analysed by Chernov et al. likely contained the response to optogenetic stimulation in the first 600 ms of the stimulus across its entire time course. In contrast, spiking neural responses in our model (Fig. 5B) and in vivo (Jun & Cardin[72] Fig. 2C) return to baseline activity within tens of milliseconds after optogenetic stimulus cessation. As such, we only used the first 600 ms of each recording (the durations of optogenetic stimulation) for all further visual and visual + opto analyses.

### Reporting summary

Further information on research design is available in the Nature Portfolio Reporting Summary linked to this article.

## Data availability

All simulation data used in this study can be freely generated by running the virtual experiments in the GitHub repository CSNG-MFF/mozaik-models[73]. Source data are provided with this paper. Additionally, we provide a Zenodo archive from which a large portion of our results (Figs. 2–6) can be generated[74]. Source data are provided with this paper.

## Code availability

The model, experiments, and analyses were implemented through the Mozaik framework[75], freely accessible at CSNG-MFF/mozaik[76] (version 0.4.1). To run the model one first needs to install Mozaik and its dependencies as indicated in the Mozaik Github repository, and then to download the implementation of model at CSNG-MFF/mozaik-models/Rozsa_Cagnol2024[73] (version 1.1) and to follow the instructions to run the experiments presented in this study. We used NEST simulator (version 3.4) as the underlying simulation engine[77]. We also made our model results and virtual experiment specifications available via a dedicated data store (http://v1model.arkheia.org) implemented in the Arkheia framework[78].

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

## Acknowledgements

Funding for this work was provided by: Charles University Primus program No. PRIMUS/20/MED/006, J.A. ERDF-Project Brain dynamics, CZ.02.01.01/00/2 2_008/0004643, J.A. European Commission, 861423, J.A.

## Author contributions

J.A. conceived of and supervised the project. T.R. and R.C. ran computational simulations, along with analyses and visualisations. The three authors jointly wrote, reviewed, and edited the paper.

## Competing interests

The authors declare no competing interests.
