## [Transparent Peer Review file · Nature Communications]

Iso-orientation bias of layer 2/3 connections unifies spontaneous, visually and optogenetically driven V1 dynamics

Corresponding Author: Dr Jan Antolik

Version 0:

Reviewer comments:

Reviewer #1

(Remarks to the Author)

This manuscript investigates the predictions of computational model of primary visual cortex (V1), focusing on experimental data regarding three phenomena that are all related to the intrinsic (rather than feedforward) V1 circuitry. Namely, spontaneous travelling waves, modular spontaneous activity and optogenetically evoke activity patterns. It can quantitatively explain a good selection of data on these three phenomena, although sometimes a comparison between predictions and data is lacking. A previous variation of the model was recently published, demonstrating it could also account for a variety of visual-driven phenomena, and these results are briefly also summarized here.

The intrinsic circuitry of cortical areas such as V1 play an important role shaping neural activity. Yet there is a lack of consensus on the function and computations performed by such circuitry. A strength of this study is that it attempts to explain a range of different experimental phenomena with a single model. This type of integrative approach is important to consolidate the proliferation of many different models, each explaining just a single phenomenon. At the same time, the model does not propose any new mechanism. Nor does it explain any previously unexplained experimental phenomena. Rather its novelty is its ability to explain a broad range of phenomena with a single model.

A second strength of the model is it that it carefully chooses and calibrates parameter choices based on a variety of different experiment data.

My main concern is related to the modelling of the optogenetically evoked activity patterns. Here the model predictions are compared to data from developmental studies of ferrets prior to eye opening at around postnatal day 31 (refs 12 and 14). The problem is that a critical component of the model is the incorporation of long-range lateral connectivity in layer 2/3 (L2/3) of cortex, that is patchy and biased to columns of similar orientation. While there is good evidence for this type of connectivity some days to weeks after eye-opening V1 in a variety of species, the evidence is that this type of connectivity is poorly developed prior to eye-opening. See: Jeremy C. Durack, Lawrence C. Katz, Development of Horizontal Projections in Layer 2/3 of Ferret Visual Cortex, Cerebral Cortex, Volume 6, Issue 2, March 1996, Pages 178–183. Therefore, while the key assumption of orientation-biased, long-range lateral connectivity in layer 2/3 is appropriate for adult V1, its not valid for the immature cortex relevant to experimental studies (ref 12 and 14) it is claiming to explain.

This issue (of the lack of patchy long-range connectivity) was addressed in ref 14, in which they showed that a computational model with only local connectivity that was suitably structured, could none-the-less explain the modular activity patterns evoked by optogenetic stimulation. In contrast, in the current manuscript, when the orientation-biased long-range connection are removed, the model is unable to explain such modular activity patterns (Fig. 4G and 4I).

This is related to a second shortcoming of the manuscript, which is that there is a lack of comparison with other models that explained each of these three phenomena in isolation, including the models in refs 4, 5, 10, 14 (and Muller L, Reynaud A, Chavane F, Destexhe A. The stimulus-evoked population response in visual cortex of awake monkey is a propagating wave. Nat Commun. 2014; 5:3675). While the models are cited, there is no comparison of the whether the models involve similar or quite different mechanisms and assumptions.

At the same time there is a lack of discussion about how the integrative model considered in the manuscript can account for diverse phenomena. Particularly, how does the model account for two different accounts of spontaneous activity in visual cortex: (1) cortical waves as refs 7 & 18, which are dynamic and changing over 10 to 100 ms, (2) modular spontaneous activity, as in ref 10, that changes very little over 100s of ms (e.g. Supp Fig. 2c in ref 10)?

A final concern is that the study ultimately does not clearly describe a functional/ computational role for the long-range orientation-biased connectivity, that is key component of the model. This is despite this being an implied aim of the study:

“Yet, a systematic understanding of how this common substrate supports the diverse computations involved in visual integration, and under different cortical regimes, remains elusive” (line 47)

Specific comments:

102: With respect to the emergence of complex cells in L2/3, what is the mechanism by which complex achieve the spatial phase insensitivity in L2/3? Please provide evidence.

94: Does the feedforward organisation include distinct ON and OFF regions (Jin, J. Z., Weng, C., Yeh, C. I., Gordon, J. A., Ruthazer, E. S., Stryker, M. P., ... & Alonso, J. M. (2008). On and off domains of geniculate afferents in cat primary visual cortex. *Nature neuroscience*, 11(1), 88-94.)?

94: The smaller text in Fig 1A is hard to read.

99: typo: “As *with* its predecessor,…”

Figures S1-S5: The one-sentence description of these figures in the main text is far too brief. While I appreciate that these figures are reproducing previous results, some additional text in supplementary material guiding the reader through what is depicted is necessary.

Comparison to experimental data is frequently lacking in these. There are no methods relating to these figure, so it is difficult to know what is being reported or why. If this is relying on publications of previous versions of the model, this should be made clearer.

Fig S2: what are the dotted lines?

Fig S3: Why use a different color scheme for contrast to that used in Fig S2. Does F0 and F1 refer to the temporal Fourier components to drifting gratings? What about complex cells? It is not clear that complex cell are the L2/3 cells, without referring to ref 23.

Fig S4: The distribution of F1/F0 by layer is more binary than experimental data.

Fig S5: A) Color code is not given. C) I would average cross-correlation should be very weak given that neurons are tuned to different orientations. What are the experimental distributions? F) Correlations for cat data do not match the model (E) well for negative correlations. G-J) How is 1/SD measured in percent? Percent of what?

Fig 2A: What is spatial scale? Color map scale? It would be good to have these available as movies in supplementary material. Some experimental reference would be helpful.

It is not clear these are waves. Or more importantly, are they similar to experimentally recorded waves.

134: What is the band of the wideband filtered signal? This should be in the main text. In the Methods it is given as 5-100 Hz, whereas in the ref 7 it is given as 5-40 Hz. The filter applied in ref 7 was also a zero-phase 8th-order Butterworth filter, by which I assume it is meant that it is applied in backward as well as forward direction to prevent phase distortion. However, in the current methods, no mention of forward-backward filtering is mentioned.

135: Why is this sentence suddenly referring to firing rates when this section is on the LFP? Is this the data plotted in Fig 6a and 6b in ref 18? How were spikes binned to calculate rates?

Fig 2F and 2G: These data for ref (7) are for area MT not V1.

175: Some introduction to the importance of stLFP would be helpful.

177: What is meant by an stLFP residual? Without any mention of trying to predict LFP based on spikes, it is not clear what is meant.

252: typo: “stimulation from *the* cortical surface”

Fig 2I and 2J: Why not compare to experimental data from Fig. 2 in ref 8?

Fig 2I: Why the large number of points with ~0 ms time-to-peak across most distances up to 3.5 mm?

Fig 2K: Why not compare to experimental data from Fig. 4 in ref 8? The shape but not the magnitude is similar. Why?

In the text it is stated that the stLFP analysis is made with 269 neurons, but in the Methods the number is given as 316.

214: The cited study ref 10, is a developmental study performed soon after eye opening. As stated in this study: as the modular correlation maps “are present prior to the maturation and elaboration of long-range horizontal connectivity, these results also present a conundrum.” The authors demonstrated that local, but heterogeneous connections can explain the modular correlations maps. So, the patchy L2/3 connectivity emphasized in the current manuscript is not necessary and infact only weakly present before eye-opening. At line 219 it is stated that “Crucially, the orientation map similarity diminishes when we remove the orientation bias of Layer 2/3 long-range horizontal connections (Figure 3E (right), 4F), further underlining the importance of the functional biased lateral connectivity in shaping spontaneous cortical dynamics.” Furthermore, the degree of similarity in correlation maps is not more than the experimental values with L2/3 orientation bias. In Fig 3 H left, it seems the model prediction are much higher than experimental values for E->Or (Note: it is not clear what E->Or means).

The model in ref 10 requires short range excitation and longer-range inhibition in cortico-cortical connections. In the model in the current manuscript, it is difficult to tell if this is also the case because the local inhibitory to excitatory connections have a hyperbolic distribution (Eq.4) while the local excitatory to excitatory connections have a Gaussian distribution (Eq. 7). Please clarify relative extent of local inhibition vs excitation. If inhibition does not spread further than excitation, then this regime should be explored.

208: It is stated that the methods of Smith et al, (ref 10) were followed, however, they explicitly screened for large-scale spontaneous events. No mention of this screening step is mentioned in the Methods of the current manuscript.

259: Once again the study in ref 14 is a developmental study, in which “the long-range axonal connections that eventually linked correlated and co-tuned domain are still poorly developed.”(ref 12). The subjects are ferret pups prior to eyes opening, in which the long-range connections are believed to be poorly developed. So, the current model, which features

such long-range connections, is not appropriate.

259: In ref (14) the pattern of activity during full-field optogenetic stimulation changes from trial-to-trial (Fig 2e). Does the current model replicate these results?

271: "removing the orientation bias of long-range connections from the model decreased the similarity to chance level, demonstrating that the functional biases in later-connectivity mediate the link between orientation maps and optogenetically evoked activity". Following my remarks for line 214, could the model in ref 10 without long-range connections also reproduce these results with full-field optogenetic stimulation?

275: Once again, the study in ref 12 is a developmental model of ferret, postnatal day 25-31, prior to eye opening. Again, the long-range connections can be expected to be weakly developed or even absent.

279: I can find no mention of "surrogate stimulation pattern" in Mulholland et al (ref 12). The description pertaining for Fig 5 in ref 12 does not mention "surrogate stimulation patterns" nor any relationship to orientation maps. If this is the case, then the manuscript does not cite any experimental work connecting optogenetic stimulation and orientation maps. This is problematic, because the next two subsections of the Results focus on the relationship between orientation maps and optogenetic stimulation. I appreciate that these two subsections can be considered to be predictions of the model. However, these predictions are then made without any data supporting link between optogenetic stimulation and orientation maps.

803: Explain the 400um spacing of the LFP grid.

807: Were the z-scores calculated in this analysis as per ref 8?

Fig 3C: give similarity index in caption or text.

Fig 3H: it is not clear what is plotted in Fig 3H either in the caption or the text. A more precise definition in the caption is required.

309: Could you clarify how "activation" is being measured here.

317: A "winner take-all mechanism in layer 2/3" is mentioned. What is this mechanism in the model? How is implemented or how does it emerge?

352: The method use in the paragraph is not very clear, mainly because it is technical without explaining the basic idea.

354: What is meant by the Orientation Domain here?

356: Should make it clear that the normalisation of the DOAD makes it a probability density function.

359: What is the ROI being referred to here? The Orientation Domain?

361: While the entropy analysis is nice in some ways, the values are not intuitive to interpret. Giving the entropy values for Fig 5C would help, but additional supplementary graphs like Fig 5C and their entropy values would help more.

Fig 6C: It is odd to measure orientation in radians in previous figures and then degrees here.

Fig 6E is not referred to in the text.

584: The ratio of LGN to L4 V1 neurons is much lower than biology, even for area centralis.

631: In Eq.2, I take it that x and y are the positions of LGN neurons? How should the function g be interpreted here? It is not really a distribution, as it takes negative values and is not normalised. Is it a modulation of the synaptic conductance from LGN to L4? How does the position of the L4 neurons affect g ? – this is missing. Presumably this should be via determining the center of the Gaussian factor in Eq. 2. A pre-computed map for the orientation, θ , is mentioned, but no details are given. Please provide this.

656: Eq(4) describes the connectivity pdf in terms of x , which is the "distance between two neurons". In ref 23 the pdf is described in terms of x , y and z where x is cortical depth of presynaptic neuron, y is cortical depth of postsynaptic neurons and z is lateral (radial) displacement. So in Eq.4 in the current manuscript, is x the lateral displacement, or the combination of lateral and vertical (between layers) displacement.

784: It is not explicitly stated how the phase is calculated from the analytical signal. Nor how the Hilbert transform was calculated.

796: Using a "*" for multiplication is not generally accepted mathematical notation and not consistent with the rest of the manuscript.

874: I initially thought phase referred to spatial phase and direction referred to drift direction of a Gabor. Perhaps something can be said here to make it clear that phase and direction refer to an ambiguity of orientation. I.e. the self-organizing map can be shifted or reversed.

876: "between the estimated orientations and the orientation map" – can you clarify that the latter means the true orientation map.

981: Can you provide a reference for "the light intensity falloff from the surface is exponential"?

Generally the mathematical notation in the manuscript should not use abbreviations like 'wl' for 'wavelength' as this is confusable with 'w' multiplied by 'l'. There are many other instances of this type of problematic notation (e.g. LFP, ISI, CV etc.)

Refs 6, 13 are incomplete

(Remarks on code availability)

The code is extensive and would require considerable time to review.

Reviewer #2

(Remarks to the Author)

Summary:

R{ }za et al. investigate the relationship between recurrent connectivity and brain dynamics in cat primary visual cortex using computer simulations of a spatially organized spiking network model.

Structurally, the model focuses on layer 2/3 and layer 4, includes a retino-thalamic, orientation specific long-range connectivity in layer 2/3, and push-pull connectivity in layer 4.

Dynamically, the model reproduces a broad range of experimentally observed phenomena. By varying the structure and other key parameters of the network model, the authors relate network structure to the exhibited activity in a mechanistic manner, and are able to generate predictions to be tested in future electrophysiological experiments.

Review:

Introduction:

L36 - L40

The authors establish a distinction between spontaneous traveling waves and module spontaneous activity. Regarding the former they write "[...] with propagation biases aligned with cortical functional organisation (8)" while for the latter they assert that

"emerging patterning in V1 are correlated with functional maps".

Later in the manuscript (L173ff) they write regarding reference (8):

"Crucially, they also observed significantly higher amplitudes of stLFP (spike triggered LFP, reviewer) residuals at recording sites with orientation preference [...]."

To my understanding this implies that also the traveling waves are related to the functional map of V1.

The authors should expand on the difference between spontaneous traveling waves and modular spontaneous activity. Otherwise, why not group the former as a part of the latter?

L45 - L47

The authors write:

"Ultimately, visual spatial integration across these cortical regimes (spontaneous, visually driven, and optogenetically driven) relies on a shared neural substrate, especially the functionally specific long-range lateral connectivity in layer 2/3."

I do not fully understand what the authors want to communicate here. What is visual spatial integration during spontaneous and optogenetically driven activity? To my understanding, there is no visual spatial integration during these regimes by definition since there is no visual input. Do the authors want to refer to cortical dynamics here? The authors should expand on this.

Additionally, the authors should expand why

"especially the functionally specific long-range lateral connectivity in layer 2/3"

is relevant for "visual spatial integration" (i.e. why it is most relevant). What about for example patchy connectivity in deeper layers, see e.g. Gabbott et al. 1987.

Results

L127 - 130

The authors write:

"To determine if the model's spontaneous activity exhibits wave-like patterns we generated a synthetic LFP signal from the neurons' input current (27), which we then convolved with a 2D Gaussian ($SD=250\ \mu\text{m}$) to account for the spatial pooling of LFPs in electrode recordings.

Could the authors comment on the choice of SD for the Gaussian kernel? $250\ \mu\text{m}$ appears rather large. Additionally, in light of e.g. Linden et al., Neuron, 2011: Why choose a Gaussian kernel at all? Linden et al. show that for large enough distances the contribution of a single cell to the LFP goes like $d^{-\gamma}$ (see their Figure 2 and eq 8). Wouldn't this be a more appropriate model for the pooling undertaken by the authors?

L130 - 131

The author write:

"As shown in Figure 2A, the synthetic LFP signal propagates across the model cortical surface as a STW (spontaneous traveling wave, reviewer)"

To what extent can the STW already be observed in the spiking activity? Would the wavefront be "sharper" (in the sense or

more clearly discernible)? Can one make a prediction of how the STW on the spiking level would look like if experimenters could record from neurons with a higher density?

Figure 1:

It is confusing that the authors first refer to Figure 1 panel B and only after that to Figure 1 panel A while it seems that the panels also could be swapped without breaking the design of the figure.

Figure 2:

They authors write:

"(B) Synthetic LFP signal (black) and corresponding generalised phase analysis, with the colour indicating the instantaneous phase."

It is not clear what the authors plot here. To my understanding they compute a spatially resolved LFP signal (shown in Figure 2 panel A). How do they get to a single time series in panel B? Is this the mean LFP? Is this for one "channel"? Also: To my understanding the authors overly the LFP signal with the instantaneous amplitude and instantaneous phase (color-coded). Is this correct?

The authors should make this more clear.

Panel C, D, G: The label of the vertical axes reads "Normalized density". The normalized is redundant - a density is by definition normalized.

Panel F: The label of the vertical axis reads "Cumulative distributio". Probably "Cumulative distribution" is meant, correct?

Panel I, J: I could not find references to these panels in the text. Please elaborate on the interpretation of the results.

L176 - L178:

The authors write:

"Crucially, they also observed significantly higher amplitudes of stLFP residuals at recording sites with orientation preference [...]."

It would be good to briefly explain what they mean by "residuals". I think it is necessary to make this clear not only in the methods but also here!

L197 - L200:

The authors write:

"Interestingly, we also find that propagation speeds and wavelenghts in model layer 4 are lower then in layer 2/3 (Figure 2EH), which we attribute to shorter-range connectivity in layer 4 which requires activity to make multiple synaptic jumps to propagate to longer distances"

An alternative explanation could be that the additional synaptic jump comes the time it takes for activity to propagate from L23 to L4. Can the authors exclude this? Why do the authors favor their explanation?

Figure 3:

Panel F: The color bar for the similarity attains values between -1 and 1. It is impossible to tell from the chosen color bar whether negative values are attained (my hunch would be yes). To my understanding the definition of the similarity in the appendix implies that the quantity is always non-negative.

Please resolve this.

L227 - L231

The authors write:

"Finally, the model accurately matches several key metrics of spontaneous ferret V1 activity: [...] correlation wavelength [...] excitatory dimensionality [...] local correlation eccentricity"

Why do the authors only here refer to ferrets but refrain from doing so for the findings regarding the correlations maps (Smith et al.)? I think it would overall help if the authors would more consistently highlight from which animal model the findings they are replicating are coming from. Surely, I agree with the authors that many of these findings are shared between highly visual mammals. But this would highlight even better the generality of the derived model.

Also: What do the authors refer to when they write "correlation wavelength"? I could not find a definition in the manuscript. Same with "excitatory dimensionality" (probably referring to dimensionality of excitatory neurons, but it's still unclear how the dimensionality was assessed). . Same with "correlation eccentricity".

Figure 4:
Panel D: Similarity between -1 and 1.

Figure 5:
The authors write:

"(C) The mean Distribution of Activity across the Orientation Domain (DAOD) over the stimulation period and trials is more sharply oriented for both endogenous (orange) and surrogate (purple) stimuli, than the distribution of orientation preference (OD) in the stimulated area. [...]."

While I agree that this can be concluded for the graph showing the "Endog. DAOD" I do not see how the claim follows from the graph for the "Surr. DAOD." Could the author please expand on this.

L361 - L363
The authors write:

"To test the existence of orientation tuning sharpening, we compared the entropies of the distribution of orientations under the stimulated regions and DAODs with lower entropies indicating broader tuning in the orientation domain."

Could the authors please expand on why a lower entropy indicates broader tuning? To my understanding, the authors are considering distributions on the domain of orientations $[-\pi/2, \pi/2]$. On this domain, the distribution with the highest entropy is the uniform distribution. I would argue, that the uniform distribution is the least broadly tuned, and that lower entropies are indicative of (but not proving) a sharper tuning.

From the displayed data in Fig 5 Panel C I would conclude that the endogenous stimulation has a lower entropy (less uniform than the surrogate stimulation), which is then also shown in panel D vertical axis.

Could the authors please comment on this issue?

Figure 6
Panel E Vertical axis tick labels are both 3.4. Please fix.

L389 - L392
The authors write:

"To understand these network phenomena more mechanistically, we performed simulations focusing on how lateral centre-surround interactions emanating from a single column shape the response to optogenetic stimulation."

Subsequently, the authors stimulate a circular region in their model. Do the authors observe a traveling wave after that stimulation, akin to the traveling waves after thalamic stimulation observed in Senk et al. Cerebral Cortex, 2024?

L418 - L419
The authors write:

"In line with iso-orientation biased long-range lateral connectivity, we found that neurons in the surround of directly simulated cortex are biased towards [...]"

"biased" -> "biased"

Do the authors mean here "simulated" or do they mean "stimulated"?

Discussion

General

Pre-computed Orientation Map

Does the pre-computed orientation map satisfy the pinwheel density derived by Kaschube et al. 2010?

Complex nonlinear interaction across the cortical space underline population responses to optogenetic stimulation in V1

The authors show that when increasing an optogenetic stimulus that especially the response of excitatory neurons show a highly non-linear behavior: first increasing the firing rate, then decreasing (Fig6D).

I was wondering whether this can be related to extra-classical receptive field effects like end-inhibition (Bolz and Gilbert, Nature 1986). Famously, predictive coding was suggested as a means of explaining these effects in an hierarchical network

(Rao and Ballard, Nature Neuroscience 1999)

However, to my knowledge it is still not resolved whether feedback is necessary to explain end-stopping in the brain. It would be interesting to see whether your network can explain this effect without additional areas just via the local recurrent connectivity.

Downscaling of model

Even though in comparison to the previous version of the model the density of neurons has increased, the model is not full scale (the authors write that they increase from a certain density of neurons to a higher density of neurons "to reduce downsampling of the number of neurons and synapses" implying that the network is still downsampled).

This is a potential problem since a substantial number of findings in this paper are based on correlations which can be strongly affected in models that are downsampled (van Albada et al. Plos CB 2017). Yet, the downscaling is only insufficiently discussed (in the main text only in relation to low wavelengths). The authors should not en passant mention this but rather discuss this fact and the potential limitations that follow from the downscaling.

Spike-triggered LFP

In the methods the authors describe how they calculate the stLFP residual. According to their description, first an expected stLFP amplitude is determined via an exponential fit, then the actual stLFP is subtracted. The authors show in Figure 2K that if the biased connectivity is removed the residuals are small for L2/3 neurons (is this the case in L4). It would be helpful to see how expected stLFPs look like to convince the reader that an exponential fit is indeed a good model.

Minor

In the equations many of the labels like "dist" and "func" are typeset as variables using an italics font. This is not correct, they should be typeset in an upright font using a command like $\mathrm{\}$ or similar to distinguish labels from variables or indices.

(Remarks on code availability)

I have not reviewed the model code but the fact that the authors use the established simulation codes Mozaik and NEST in recent versions substantially increases the probability that the results can be reproduced. This is how computational studies should be done today. In terms of reproducibility the work is well ahead of a typical study in computational neuroscience today.

Reviewer #3

(Remarks to the Author)

This manuscript by Dr. Antolik et al. presents a neural network model of V1 layer 4 receiving LGN input, and its primary downstream target, layer 2/3. Based on anatomical connectivity studies, the model connects neurons in a biologically realistic manner, incorporating distance-dependent strength and synaptic delay, patchiness given by an orientation map, a push-pull scheme in layer 4, and iso-orientation biased long-range cortical connectivity in layer 2/3. The model successfully captures numerous aspects of experimental observations from the early visual system. Various parameters have been explored, and overall, the study is carefully and comprehensively implemented, effectively focusing on the role of iso-orientation biased long-range cortical connectivity in V1 layer 2/3 across a series of experimental results. The study provides substantial computational insights and intuition, aiding understanding.

Major comments:

1. As referenced in the manuscript, Z. Davis (2024) presented a model with a highly similar recurrent structure and demonstrated functionally biased spontaneous activity. The authors claim to extend this by linking it to visually or optogenetically driven processing. However, for the results investigating how the functional specificity of cortical connections dictates the correlation structure of spontaneous activity, it is unclear how the biased long-range horizontal connections give rise to the similarity between CMs and the orientation map differently compared to the previous model (Fitzpatrick, M. Kaschube et al, 2018) with heterogeneous Mexican hat connectivity giving rise to modular activity pattern. Or say, how are these two models essentially different?

2. As a modeling study, the strength of the work lies in the solid analysis proving the ability of the model to capture a number of important experimental results in the field. The weakness is the testable predictions generated by the model. I am skeptical about the significance of the predictions listed in the manuscript. The authors predict slower wave propagation lacking orientation bias in the layer 4. How this could influence behavior given the established finding of the characteristics of traveling wave in its downstream target layer 2/3 remains to be addressed. Also, the prediction of the non-monotonicity of optogenetically driven cortical response to increasingly stimulated area is not surprising since it can be explained by the extensively studied inhibition-stabilized network (ISN) regime, as the authors mentioned in the discussion. This also pertains to my first major comment.

3. Since the paper covers a broad range of experimental results (traveling wave, distal correlated activities, optogenetically driven cortical responses), a brief introduction on each metric being investigated in the result section may provide useful intuition and help communicating the concept for readers with diverse research interests. For example, fig. 3H is abruptly presented with little introduction. The same comment applies to the concept of stLFP residual, entropy of DAOD distributions.

Minor comments:

- Figure 1A inset illustrating the lateral connectivity properties: incorrect x-axis label in the right panel.
- Figure 2K: x-axis label should be '.... in orientation preference'.
- Figure 6B, right: incorrect x axis label.

- as the model is claimed to be applicable to primate V1, a discussion of eccentricity-dependent cortical organization should be included.
- the model includes feedback pathway from the layer 4 to the layer 2/3. This is a bit confusing since the output superficial layers don't project significantly to the input layers to my knowledge. Is this necessary to any of the key results?

(Remarks on code availability)

Reviewer #4

(Remarks to the Author)

In this work Rózsa et al. use a spiking recurrent network model to simulate activity patterns in the visual cortex under a variety of stimulus conditions. They show the model can describe data on spontaneous traveling waves and the spatial patterns that arise from spontaneous activity or optogenetic stimulation. This model is a combination of two previously models developed by these authors: 1) a multi-layer model of feedforward visual stimuli (Antolík et al., 2024), and 2) a brain-machine interface model to simulate optogenetic stimulation of cortical networks (Antolík et al., 2021).

Overall, this work is an extension of previous work studying spatially biased functional connectivity within recurrent networks. As the authors note in the introduction, biased topographic connectivity is required for orientation preference biased spontaneous traveling waves (Davis et al., 2021), spontaneous activity patterns with long-range correlations (Smith et al., 2018), and optogenetically evoked activity patterns (Mulholland et al., 2024). The main advance here is repeating this in a spiking network where temporal dynamics are better constrained by synaptic time constants, delays, and biophysical mechanisms. The authors use this model to make several testable predictions: 1) recurrent network amplification of input patterns that align with the iso-orientation-biased functional connectivity, 2) and the nonlinear effects of local recurrent connections on surround activity when provided a localized input.

The primary result here is that moderate-strength iso-oriented long-range connectivity improves model description of the data. My enthusiasm is limited by several factors. The level of bias is not systematically varied and at times the effects of adding bias can be weak (Fig. 4I). This connects to a larger point: the manuscript does not deeply explore the circuit mechanisms that allow the data to be described. This is true especially for the bias parameter but also holds for other parameters. There is some of this in Fig. 7, but the parameter optimization is relatively limited and in some cases (Fig. 7D) parameters have little influence. I would rather have seen this manuscript focus on the parameter regimes — the network mechanisms — that describe the data. Why is this model the model of greatest interest, and what are the essential mechanisms that make it so?

That said, I believe the paper in the present form will be a contribution to the field. I have some comments below but none are serious obstacles to their results.

Major:

I would have liked to see the authors take advantage of the multiple layers of their model to study the interaction between feedforward stimuli (visual input) and local recurrent network activity (spontaneous or optogenetically-evoked), which is not examined here.

In Figure 3H the authors argue that the model results replicate in vivo ferret data, but for several of the metrics the model falls outside the in vivo ranges. Specifically, there are discrepancies in the similarity between excitatory correlation maps and orientation preference maps, inhibitory event dimensionality, and inhibitory eccentricity. How do you account for these differences, is there a mechanistic explanation? The data taken from Smith et al., 2018 and Mulholland et al., 2021 was measured in young, postnatal ferrets—could this be due to differences in the developing brain? Overall, the authors should modify their statement to highlight where the model differs from the in vivo data, and offer a discussion of the possible reasons why.

This paper would benefit from a more detailed explanation of the Distribution Across Orientation Domains (DAODs) and distribution of orientation preferences within the stimulated ROIs (OD) (Fig 5D), which was non-intuitive and initially difficult to evaluate how it relates to functional network connectivity. What specifically is the difference between these two measures and their entropies? I presume that DAODs are derived from the activity during modeled optogenetic stimulation and ODs are derived from the activity of modeled spontaneous activity within stimulated ROIs (based on the black labels in Figure 5E), but this was not clearly stated in the legend or text.

Furthermore, interpreting this was occasionally made more difficult by apparent contradictions in the text. For example, in lines 362-363, it states: “with lower entropies indicating broader tuning in the orientation domain”. However, Figure 5C suggests that endogenous patterns—which in Figure 5D and E have lower entropies—have sharper tuning. The fact that patterns with lower entropies correlates with higher firing rates is consistent with the finding that endogenous patterns are more strongly amplified than surrogate, which suggests to me that the statement in lines 362-363 is a typo.

Overall, a more explicit, simplified explanation of these terms in section will help readers evaluate and interpret the results in figures 5&6.

How does the location of the local stimulus affect the results of Figure 6? If the stimulus was not centered on an iso-orientation domain, but instead overlapping over two domains or a pinwheel center, how would that affect the firing rates for

excitatory and inhibitory cells? Would the difference in firing rates between the 'iso' and the 'ortho' surround disappear, or would there be a winner-take-all effect?

Minor comments:

In lines 137-139, you state "Prior to convolution, the wavelengths of our synthetic LFP signal are significantly higher than in the spatially shuffled signal, demonstrating that the signal is indeed spatially structured". I assume you mean that the wavelengths are significantly smaller (higher frequency), as shown in Figure 2F. Also, here and in Figures 2E,F it was not immediately clear in the text or the legend whether you were measuring the spatial wavelength of the traveling wave or the temporal wavelength calculated from the Generalized Phase analysis.

You predict that Layer 4 would have slower spontaneous traveling wave propagation than Layer 2/3. Given that Davis et al., 2020 shows that the phase of spontaneous activity affects the sensitivity to visual stimuli, could you speculate what effect, if any, this slight temporal de-synchrony could have on visual perception?

In the legend of Figure 5C, it says "The mean Distribution of Activity across the Orientation Domain (DAOD) (see Methods) over the stimulation period and trials is more sharply oriented for both endogenous (orange) and surrogate (purple) stimuli, than the distribution of orientation preferences (OD) in the stimulated area." However in Fig 5C, the surrogate DAOD values (purple line) do not look different from the stimulated area (black line). Is there a significant difference here, or is this an error? I assume so based on Fig 5E, but this was not clear from the legend or text.

Other:

What is the scale bar on the traveling waves in Fig 2A?

Fig 2 C, D and G do not have y-axes.

In Fig 4C please add labels for which map was made from spontaneous activity and optogenetic stimulation.

In Fig 4I, is the 'Endogenous No Bias' significantly different from 'Surrogate' or 'Endogenous'? Please include this analysis and add errorbars / variance metric to the panel.

In Figure 6 C-F, to evaluate how changing different model parameters affects the different metrics, it would be helpful if you visually indicated the biological range of the in vivo data and/or the model parameters used in the previous figures.

(Remarks on code availability)

Reviewer #5

(Remarks to the Author)

(Remarks on code availability)

Reviewer #6

(Remarks to the Author)

(Remarks on code availability)

Version 1:

Reviewer comments:

Reviewer #1

(Remarks to the Author)

The authors have done a commendable job at addressing the issues I raised in my previous review. A few are worth noting.

First, on the issue of young (pre-eye opening) vs adult animals and the absence vs presence of long-range patch connections: I accept the point that the model of Smith et al., for young animals, required very strong recurrent connections relative to afferent connections, which is not appropriate for the adult. So the phenomena seems to require two models for a complete explanation: the Smith model for young animals and the model in the current study for adult animal. While this is not parsimonious, it is not unreasonable. This is acknowledged and discussed in the revised manuscript. Meanwhile, there are many other contributions of the paper, so issue should not be an impediment to publication.

Second, the explanation about the different observed time scales for spontaneous activity in V1, as a measurement artifact, in terms of the effect of the dynamics of calcium imaging was interesting.

Third, it is also reasonable to leave to further work a study of the functional/computational insights available from the model, and instead focus on firmly establishing the model's explanatory power in relation to diverse experimental observations.

With respect to the explanation about the emergence of complex cells in L2/3: while it is reasonable to point out that L2/3 neurons pool of the responses of L4 neurons with different spatial phase tuning, this is not sufficient - the form of the pooling matters. E.g. linear or threshold-linear pooling will not result in phase invariance (e.g. the threshold-linear model in Carandini 2006, Fig. 1 won't work, with Gabor receptive fields). So some explanation of how the correct nonlinear pooling arises would be helpful.

(Remarks on code availability)

Reviewer #2

(Remarks to the Author)

Review of revised manuscript "Iso-orientation bias of layer 2/3 connections: the unifying mechanism of spontaneous, visually and optogenetically driven V1 dynamics"

(Line numbering of document including changes in the manuscript.)

The reviewer thanks the authors very comprehensively addressing the raised questions (including insightful additional analyses) and correcting the identified minor mistakes.

The additional work greatly improves the strength of the manuscripts and makes it more convincing by the unique opportunities to alter key parameters unique to simulation studies.

The following comments are only minor and meant to further improve the clarity of the manuscript.

Review:

Introduction

L33 f.

The authors write

"Several phenomena highlight the interconnectedness of visually and non-visually driven dynamics: ..."

One of the later mentioned phenomena is grouped under the heading: "Optogenetically evoked patterns of activity".

In my opinion, this is in stark contrast to the other mentioned phenomena, "Spontaneous travelling waves", "Modular spontaneous activity".

Do the authors agree that there seems to be somewhat of a difference? Or could they explain why there is none?

The explanatory text for the "Optogenetically evoked patterns of activity" does not make this clearer, since no concrete "phenomenon" is mentioned, but rather what optogenetic stimulations allows to study.

L70 ff.

The authors write:

"By comparing to data from multiple species, we demonstrate that the model replicates (i) distributions of wavelength and propagation speed in spontaneous waves observed in marmoset, along with biases toward iso-oriented propagation seen in macaque and cat V1, (ii) modular spontaneous activity correlated with orientation maps in both excitatory and inhibitory neurons in ferret V1, (iii) impulse-response properties to both full-field homogeneous and patterned optogenetic stimulation in ferret V1, and (iv) range of visually driven phenomena."

Do they mean "a range of visually driven phenomena" (as in "a number") or the "spatial range" of such phenomena? Please clarify.

L79 ff.

The authors write:

"Parametric analysis of the model reveals ..."

I think what the authors mean is that changing certain parameters leads to activity that does not match to the experimentally observed one. This is of course a strong case for their model. I stumbled, however, over the formulation of "Parametric analysis" which for me immediately evoked "Analysis via parametric statistics". Maybe the authors could reformulate this slightly for the sake of clarity.

Results

L180 f.

The authors write:

"Next, Davis et al. offered a detailed characterization of the spontaneous wave properties in marmoset MT using the Generalized Phase analysis (7, 15)."

I couldn't find the place in the manuscript where the authors resolve the abbreviation MT. If it is not in the manuscript, please add.

L285 ff.

The authors write:

"Activity of both inhibitory and excitatory neurons is correlated over multiple millimetre distances, with excitatory activity resembling orientation preference maps in young and adult carnivore/primate ferret V1 (10, 20, 31, 32)."

To my understanding of the cited literature, the correlation patterns of neural activity (and thus predominantly excitatory activity) resemble orientation preference maps and not the "excitatory activity" itself, as stated by the authors. Please comment on this.

L564 f.

The authors write:

"We have shown that model responses to endogenous (OP map correlated) optogenetic stimuli are stronger than to equivalent surrogate stimuli."

I couldn't find what OP might stand for? Doe the other mean OT for orientation?

L656 ff.

The authors write:

"The precise location of this inflection point is the result of a complex interplay between the spatial scale of the orientation map, the spatial extents of long-range excitatory and short-range inhibitory connections, and balance of excitation and inhibition in the network. Nevertheless, the existence of the inflection point at around 100 μ m radius is an important prediction of the model."

Some (if not all) of the quantities mentioned to underlie the point of inflection depend on the species (e.g. the iso-orientation domain size between cat and macaque differs on average). It is great that you make a specific prediction from your computational model. For which animal model would you expect it to fit best?

Methods

L904

The threshold slope factor is written as ΔT instead of Δ_{T} (the letter version is used in equation 1).

L942

The authors write:

"The phase preference Ψ , is generated randomly."

I assume the authors mean generated randomly following a uniform distribution with on an appropriate interval. Could the authors make this more explicit?

L950 ff.

The authors write:

"Individual layer 4 excitatory and inhibitory cells receive connections from the LGN whose number is drawn from uniform distributions of respective boundaries of [90, 190] and [112, 168]."

I had to read the sentence multiple times to understand what the author want to communicate with the term "boundaries". I think the technical term the authors are looking form is the "support" of a uniform distribution.

L978 ff.

Could the authors add the units for θ and α ?

L1250 ff.

The dimensionality metric employed by the authors is known as participation ration (e.g. Dahmen et al. 2019). Since there many other way to assess the dimensionality of neural activity (for example based on the number of principal components needed to reach a certain value of explained variance) the authors should mention the name of this dimensionality metric and cite the parts of the relevant literature (Abbot, Rajan, Sompolinsky, 2011, "Interactions between intrinsic and stimulus evoked activity in recurrent neural networks", Mazzucato et al. 2016, "Stimuli reduce the dimensionality of cortical activity")

L1323 ff.

I might have missed it here but I do not understand how the authors estimate the temporal scale of spontaneous activity. I do get that they calculate the temporal autocorrelation function. But what then? Do they fit an exponential to extract something. Do they use a method like the one suggested by Zeerati et al 2022. Or do you just visually compare the autocorrelation function of simulated and recorded activity?

(Remarks on code availability)

Reviewer #3

(Remarks to the Author)

The authors have satisfied all my concerns from the from first round of reviews. I appreciated their diligent responses to all four reviewers, the manuscript is now much improved in my opinion. I wish the authors the best of luck in managing the current overwhelming length of the paper moving forward.

(Remarks on code availability)

Reviewer #4

(Remarks to the Author)

Thank you for the work and the new revisions. The authors have made a significant effort in revision to assess reviewer concerns. The edits made to the manuscript have greatly improved its interpretability and transparency. The authors have addressed my concerns. I have one comment on the revisions, which does not require followup.

Regarding the local stimulus location experiments: Thank you for taking the time to investigate this question. By moving the stimulation target around, you have demonstrated that where you stimulate can have variable effects on the response of the network, with column centers providing the strongest amplification, while edges and pinwheels are less effective. This is another piece of evidence supporting a "soft winner-take-all mechanism", which has been demonstrated in other ways throughout the manuscript. For this reason, it may be worthwhile to include these figures in the supplement (both are illuminating, but the first one alone may be enough to make this point without having to do extensive simulations). However, I agree an in-depth study on this topic is outside the scope of this paper, and therefore leave this to the decision of the authors.

(Remarks on code availability)

Reviewer #5

(Remarks to the Author)

(Remarks on code availability)

Reviewer #6

(Remarks to the Author)

(Remarks on code availability)

Our responses to reviewers' comments are in **red**. Changed text in the manuscript cited in our responses is in *italics*, and **line numbers refer to the version of the manuscript with highlighted changes**.

Summary

We thank the reviewers for their thoughtful comments that have substantially improved our manuscript.

Following reviewers' requests we have now substantially revised the manuscript with a range of new analyses. We performed additional comparisons of the model against 4 new experimental studies (Tsodyks et al. 1999, Kenet et al. 2003, Chernov et al. 2018, Kettlewell et al. 2025) further solidifying the anchoring of the model in experimental data. Altogether, we have substantially expanded Figures 2,3,4,5,7 and performed numerous additional control analyses resulting in 9 new supplementary figures and one supplementary video. We have substantially revised the manuscript text to include all the new analyses and figures. We address the major concerns of reviewers by (1) inclusion of new comparisons to adult V1 data on spontaneous structured activity and optogenetic stimulation, (2) substantial extension and refinement of the analysis of the spontaneous travelling waves and (3) additional model parameter exploration experiments.

Overall, we have identified three important recurring themes in reviewers comments for which we would like to offer an overarching response in this summary, but which we further address in full detail in the responses to individual reviewers:

1. Quantitative comparison to animal data: In the first manuscript version, we compared our model to data from 7 different experimental studies against what is arguably an unprecedented range of experimentally observed V1 phenomena spanning 2 different cortical driving regimes (spontaneous and optogenetically driven). This is on top of a range of other visual driven phenomena we compared the model to in our previous work and in supplementary material in this manuscript. In this revision, we have now further extended this battery of tests with other 4 experimental studies. Crucially, to compare against such a range of findings we were forced to select data spanning four species (macaque, marmoset, cat and ferret) as no species offers a complete set of experiments.

Our goal here was hence to develop a generic model of adult columnar primary visual cortex that can reproduce all the targeted phenomena, rather than quantitatively finetune the model for one single species (or single experiment as is often done). Considering the sheer breadth of comparisons and multi-species data, overall our model offers a surprisingly accurate quantitative match to the multi-faceted data. But indeed there is a minority of measures where small but statistically significant discrepancies can be observed. These discrepancies can be explained by the interspecies differences which our generic model cannot simultaneously satisfy, although in some cases we also identify other specific more likely

reasons that we discuss in detail in the direct responses to reviewers' individual concerns (and where appropriate in the revised manuscript).

We hence wish to argue that this minority of quantitative discrepancies from data should not be viewed as a weakness of our study, but rather that the very good overall quantitative match to the multi-faceted data is a major strength of the study. All that said, reviewers were right that in places we should have been more explicit in the text at drawing attention to those quantitative discrepancies, which we have now fully rectified in the updated manuscript.

2. Comparison to data from young animals: As some reviewers pointed out, some studies, to which we compare the adult V1 model, first on existence of spontaneous structured activity (Smith et al. 2018, Mulholland et al. 2021) and second on impact of functionally specific optogenetic stimulation (Mulholland et al. 2024 Feb., Mulholland et al. 2024 March, Mulholland et al. 2024 May), were performed in young ferrets around the time of their eye-opening. Reviewers raised the concern that pre-eye opening V1 circuitry is still developing, and hence questioned the comparison to the adult model of V1. The comparison to the first set of studies was motivated by the fact that it is the most thorough of its kind and it hence allows for more in-depth comparisons with our model than previous studies in adult animals. The comparison to the second set of studies was motivated by the fact that data on functionally specific stimulation in adult columnar V1 remains extremely scarce. We however argue these comparisons are justified because (a) the principal study of Smith et al. 2018 on which all further studies of Mulholland et al. are based off shows that the phenomena of our interest are present also post eye opening and (b) a number of other studies have shown the presence of structured spontaneous activity in adult animals (Tsodyks et al. 1999, Kenet et al. 2003). In this revision we have now added further comparisons to other studies of spontaneous structured activity in adult animals, and to another study of optogenetic stimulation in adult columnar V1. We find that our model is in line with these additional experimental findings, further solidifying the evidence that our model captures the dynamics of adult V1 in both the spontaneous and optogenetically perturbed regimes.

3. Requests for additional experiments: The scope and biological detail of our model mean that the majority of physiological experiments that have been performed in layers 4 and 2/3 of columnar V1 could be meaningfully investigated in our model. We consider this is a testament of the scope and flexibility of our modeling framework, underscoring its potential impact for future visual system research. It, however, also poses a challenge for us as it offers virtually infinite follow-up lines of investigations. Indeed reviewers have suggested quite a wide array of potential additional experiments. We agree these are all exciting future opportunities. We hope that reviewers will agree that we have tried to maximally accommodate their requests, having performed a very long list of additional experiments/analyses. Crucially, we have diligently performed all the requested experiments that were necessary for supporting the key claims made in our manuscript. However some of the requested experiments were firmly outside the scope of this study and were hence left for future work.

We wish to emphasize that we truly share the enthusiasm with reviewers about a number of the suggested lines of future investigation, and hope to pursue them in our future work. Our

group pursues a long-term research program to develop a model that unifies an increasingly broad range of findings on anatomy and function of the early visual system. Our commitment to this plan is evidenced by the series of what is now already 4 studies (Taylor et al. 2021, Antolík et al. 2021, Antolík & Cagnol et al. 2024, Cagnol et al. 2025) in which we gradually build up the modeling framework. However the only practical way such an ambitious long-term goal can be achieved is to report our achievements gradually as individual milestones are met. We are of the opinion that the present study resolutely represents such a milestone, having for the first time unified research on three different driving regimes of functionally specific V1 activity dynamics.

Reviewer #1

Reviewer #1 (Remarks to the Author):

This manuscript investigates the predictions of the computational model of primary visual cortex (V1), focusing on experimental data regarding three phenomena that are all related to the intrinsic (rather than feedforward) V1 circuitry. Namely, spontaneous travelling waves, modular spontaneous activity and optogenetically evoke activity patterns. It can quantitatively explain a good selection of data on these three phenomena, although sometimes a comparison between predictions and data is lacking. A previous variation of the model was recently published, demonstrating it could also account for a variety of visual-driven phenomena, and these results are briefly also summarized here.

The intrinsic circuitry of cortical areas such as V1 play an important role shaping neural activity. Yet there is a lack of consensus on the function and computations performed by such circuitry. A strength of this study is that it explains a range of different experimental phenomena with a single model. This type of integrative approach is important to consolidate the proliferation of many different models, each explaining just a single phenomenon. At the same time, the model does not propose any new mechanism. Nor does it explain any previously unexplained experimental phenomena. Rather its novelty is its ability to explain a broad range of phenomena with a single model.

A second strength of the model is that it carefully chooses and calibrates parameter choices based on a variety of different experiment data.

We thank the reviewer for highlighting the strength of our study. We wish to comment on three issues:

1. **On the lack of comparisons:** In the first manuscript version we compared the model to data from 7 different experimental studies spanning four species (macaque, marmoset, cat and ferret), which were now further extended by comparison to another 4 studies. This in addition to numerous others in our previous work (Antolík et al. 2024, Taylor et al. 2021). We argue that this is well above the typical rigour with which models are validated in neuroscience. Later on the reviewer identifies several opportunities where our model could be compared even at greater level of detail and we have addressed these whenever practically feasible (see direct responses to the respective comments). It is true that in the later (exploratory) sections of the paper we also generate predictions for which data is currently not available, but we argue this is exactly the role theory should play in a well developed field - models being

validated against as broad set of established facts as possible, yet generate plenty of testable predictions for future studies to test.

2. **On the lack of new mechanisms:** we argue that this is actually the key strength of the study rather than its weakness. The current understanding in neuroscience, particularly in the early visual system, is highly fragmented, with a large body of existing models, relying on a wide range of hypothetical mechanisms that each demonstrates a single or few phenomena at a time at most. Yet it is unclear which of these explanations are mutually compatible, and if any combination of them is sufficiently powerful to actually explain the system in its full complexity (or at least its substantial fraction). Similarly, for many phenomena many alternative (but often overlapping) mechanisms were proposed (take orientation tuning as an example) yet it remains unclear which of them actually corresponds to reality. We argue there is a **dire need for synthesis** of existing findings and theories. In this context, demonstrating that a given set of biologically well established mechanisms can explain a truly broad set of phenomena under 3 major different cortical driving regimes should be considered a major achievement, rather than a weakness.
3. **On lack of newly explained experimental phenomena:** while we do not consider this to be the main achievement of the paper (the point 1 and particularly 2 explain the key achievements), to the best of our knowledge, we do model several phenomena for the first time:
 - a. The iso-orientation biased propagation of spontaneous waves demonstrated by Nauhaus et al. 2009 and replicated in the our Figure 2B-D has not been to the best of our knowledge been modeled earlier.
 - b. Quantitative match of inhibitory correlation map metrics to experimental data (Figure 3H)
 - c. Similarity between the correlation maps calculated from excitatory vs. inhibitory neurons (Figure 3F)
 - d. Similarity between the spontaneous and optogenetically evoked correlation maps (Figure 4E)
 - e. Response ratio inside/outside of optogenetically stimulated area for endogenous stimuli (Figure 4H)
 - f. Qualitative match of iso-facilitation and ortho-suppression during simultaneous full-field visual stimulation and optogenetic stimulation targeting an orientation column (Figure 4L)

The primacy of these replications was now more clearly declared in the discussion in following paragraph (lines 729-734):

The model for the first time reproduces: (i) iso-orientation biased propagation of spontaneous waves, (ii) similarity between excitatory and inhibitory correlation maps, (iii) modular activity response to uniform full-field optogenetic stimulation that is correlated with modular spontaneous activity, (iv) response ratio inside/outside of optogenetically stimulated area for endogenous stimuli, and (v) the iso-facilitation and ortho-suppression during simultaneous visual and single-column optogenetic stimulation.

My main concern is related to the modelling of the optogenetically evoked activity patterns. Here the model predictions are compared to data from developmental studies of ferrets prior to eye opening at around postnatal day 31 (refs 12 and 14). The problem is that a critical

component of the model is the incorporation of long-range lateral connectivity in layer 2/3 (L2/3) of cortex, that is patchy and biased to columns of similar orientation. While there is good evidence for this type of connectivity some days to weeks after eye-opening V1 in a variety of species, the evidence is that this type of connectivity is poorly developed prior to eye-opening. See: Jeremy C. Durack, Lawrence C. Katz, Development of Horizontal Projections in Layer 2/3 of Ferret Visual Cortex, Cerebral Cortex, Volume 6, Issue 2, March 1996, Pages 178–183. Therefore, while the key assumption of orientation-biased, long-range lateral connectivity in layer 2/3 is appropriate for adult V1, its not valid for the immature cortex relevant to experimental studies (ref 12 and 14) it is claiming to explain.

This issue (of the lack of patchy long-range connectivity) was addressed in ref 14, in which they showed that a computational model with only local connectivity that was suitably structured, could none-the-less explain the modular activity patterns evoked by optogenetic stimulation. In contrast, in the current manuscript, when the orientation-biased long-range connection are removed, the model is unable to explain such modular activity patterns (Fig. 4G and 4I).

Our study aims to provide a unified explanation of the **adult** primary visual cortex. We do not aim, nor do any claims on explaining the development of the V1 architecture. Upon reflection on the manuscript we now see this was not stated clearly enough in the text, despite it being very important in the context of making comparisons to some studies in young animals (ref 12 and 14). We want to thank the reviewer for bringing this important issue up, and we have now updated the text to make the focus on adult V1 abundantly clear (see lines 11, 59, 67, 84, 89, 287, 328, 330, 476, 725, 740, 826).

This however raises two issues which we wish to address in greater depth. First we address why do we compare adult model to data from young animals in the first place, and second why our observations are not in contradiction with biological findings:

1. Why do we compare the model of adult V1 to studies in young animals: The phenomenon of structured spontaneous activity was observed also in adult animals in numerous previous studies for example in Kenet et al. 2003, to which we compared our model in Figure 3G. We, however, chose to compare primarily to studies 12 and 14 because, to the best of our knowledge, they are by far the most thorough and quantitative examinations of spontaneous structured activity available, and hence it allowed us to make much more deep links between biology and the model.

It is also important to point out that Smith et al. show that the statistics of the spontaneous activity that we investigate here stabilize around the time of eye opening (Figure 4D,E, 5C, SF2F, SF9BDFH of Smith et al. 2018;). It hence justifies using the data provided by these two studies, that pool data from period around eye opening (P21-P47 Smith et al 2018; P23-P29 Mulholland et al. 2021, P23-P29 Mulholland et al. 2024), to perform comparisons to our adult model of V1, although we acknowledge this might introduce some quantitative (**but not qualitative**) deviations from adult data. Acknowledgement of this issue was indeed lacking in our manuscript, and we have now rectified this in the Discussion (lines: 843-848).

That said, to further strengthen this point and demonstrate the validity of the model in terms of spontaneous structured activity in **adult** animals, we have now made new direct

quantitative comparisons to additional previous studies done in adult animals - see added panels I,J,K,L for Figure 3, results paragraph lines, showing excellent agreement with this other adult data further reinforcing the evidence that the model accurately reproduces the correlation structure of spontaneous activity to the orientation preference map in **adult V1** (lines 330-344):

To validate our model also against data from adult animals, we assess how well the model reproduces the level of correlation between the orientation preference map and spontaneous activity measured in adult cat (Kenet et al. (31), Tsodyks et al. (33)) using alternative metrics. The distribution of correlation coefficients between spontaneous activity frames and single-condition orientation maps (activity maps in response to gratings of a single orientation; see Methods) matches that of Kenet et al. (31), including the control distribution computed using horizontally flipped single-condition maps (Figure 3I). Next, the model matches the correlation distributions of spontaneous activity frames to the Preferred Cortical State (PCS) of neurons (Figure 3J), a proxy for single-condition orientation maps (Tsodyks et al. (33), see Methods). Additionally, the subset of the above distribution consisting of only spontaneous frames during which the neuron fired an action potential (Figure 3K), exhibits a significant positive bias ($p < 10^{-16}$, $n = 367169$, 1-sided one-sample T-test; mean = 0.1007 ± 0.0003 for model, 0.1591 ± 0.0045 for experiment). Thus, the predicted firing rate (Figure 3L, ratio of histograms K,J) increases with stronger correlations to the orientation map, in line with cat data.

Figure 3: The structure of correlations in spontaneous activity is related to orientation maps

... (I) Validation against Kenet et al. (29): The distribution of correlation coefficients between model spontaneous activity frames and single-condition orientation preference map is consistent with experimental data, including correlations to control (horizontally flipped) maps. (J-L) Validation against Tsodyks et al. (31): (J) The distribution of correlations between model spontaneous activity frames and Preferred Cortical States (PCS) (see Methods) closely matches experimental data (K) The distribution of correlations to PCS for spontaneous frames at which the monitored neuron fired an action potential exhibits a significant positive bias ($p < 10^{-16}$, $n = 367169$, 1-sided one-sample T-test), consistent with experimental data (mean = 0.1007 ± 0.0003 for model, 0.1591 ± 0.0045 for experiment). (L) The predicted instantaneous firing rate increases with PCS correlation for both model and experiment. It is calculated as the ratio of histograms (L) and (K) divided by the duration of the time frame.

2. Why can a model without long-range connections explain long-range spontaneous correlations, but these long-range correlations substantially weaken, and the similarity of correlation and orientation maps disappears in our model once long-range connections lose their iso-orientation bias or are removed completely.

Indeed, as already shown in Smith et al. 2018, a simple model relying on anisotropic Mexican hat-like lateral interactions can explain long-range spontaneous correlations without the need for long-range co-orientation biased connections. It has to however be emphasized that even if during development long-range connections might not be necessary for inducing long-range correlations this does not imply that this might not be so in the adult cortex, that we model here. For example during the development these short-range interactions might need to be very strong relatively to the afferent input, as indicated by Smith et al. 2018 in the original modeling study, where authors state that the long-range correlations develop only once the model dynamics are strongly dominated by lateral interactions (see Methods of Smith et al. 2018), which seems to occur when the afferent input is about **1/100th** of the lateral interaction strength (Fig 7D in Smith et al. 2018). Such very weak contributions from the afferent pathway might however not be sufficient for proper visually driven processing required of adult V1, as corroborated by the fact that in young animals visual stimulation is ineffective in evoking reliable neuronal responses (44, 45). Hence, in adult animals, where the afferent pathway is the driver to cortical activity, and long-range orientation-biased connections are an experimentally established fact, upon their removal (or removal of their iso-orientation bias), the relatively weaker lateral connections (as opposed in young animals) might not be sufficient anymore to drive the long-range correlations as suggested by our model. We have now added a paragraph to the Discussion that discusses this issue (lines 823-833):

... It is important to emphasize that even if during development long-range connections might not be necessary for inducing long-range correlations, this does not imply this is also the case in the adult cortex. For example, the cortex is much more driven by the afferent pathway in adult animals than in the young in which visual input is ineffective in evoking reliable neuronal responses (10, 48). This is further corroborated by modeling in Smith et al. (10) that shows that the long-range correlations emerge only once the model dynamics are strongly dominated by lateral interactions, i.e. when the afferent input is about 1/100th of the lateral interaction strength. Under the weaker lateral interactions in the adult condition, the long-range connections might be necessary for the emergence of the long-range correlations in spontaneous activity, as our model suggests.

This is related to a second shortcoming of the manuscript, which is that there is a lack of comparison with other models that explained each of these three phenomena in isolation, including the models in refs 4, 5, 10, 14 (and Muller L, Reynaud A, Chavane F, Destexhe A. The stimulus-evoked population response in visual cortex of awake monkey is a propagating wave. Nat Commun. 2014; 5:3675). While the models are cited, there is no comparison of the whether the models involve similar or quite different mechanisms and assumptions.

We acknowledge that the manuscript lacked a detailed discussion and comparison of our model to the ones mentioned. Our intention was to keep the discussion concise due to the length limits imposed by the Nature Communications. Nevertheless, we have now added the following detailed model comparison section into the Discussion:

Several other models aimed at studying some of the individual phenomena we reproduce in our model. Spontaneous travelling waves have been generated in small-scale spiking neural networks with local distant-dependent recurrent connectivity (45, 46), but these waves are too dense in comparison to the sparse traveling waves observed in vivo (4). In a simple model with ring architecture, Muller et al. have shown that the oscillatory activity of the model could propagate laterally only when the length of the model was within a specific range (5). Davis et al. designed a topologically organised spiking neural network which also exhibited spontaneous traveling waves with realistic wavelengths and propagation speeds properties (15). Importantly, they showed that the presence of the topologically-organized recurrent connectivity is essential for the emergence of these properties. In a follow up study, they present an iteration of the model with functionally-biased long-range connectivity (9). They demonstrate that this connectivity scheme is needed to reproduce activity phase patterns associated with functionally biased spiking activity which they observed in vivo. Their work is complementary to ours as they focus on the functional bias of modulation of spiking activity induced by phase patterns of waves whereas we focus on the functional bias of the propagation of the traveling waves, with both studies emphasizing the importance of functionally-biased connectivity. Unlike our study, however, Davis et al. do not investigate iso-orientation bias of activity propagation, do not consider retino-cortical pathway and hence visual driven regime, and do not investigate optogenetic stimulation.

Next, spontaneous structured activity was modeled by Cai et al. (19) in a single-layer spiking V1 model. Just as our study, Cai et al. assumed iso-orientation biased excitatory long-range lateral connectivity and demonstrated spontaneous events that were dominated by a narrow range of orientations, but did not provide direct quantitative comparison of these spontaneous correlations to experimental data. Interestingly, spontaneous structured activity is present already in young animals before eye-opening (10), before long-range correlations are fully formed (47). Smith et al. offered a possible resolution to this conundrum by demonstrating that a model with only short range Mexican-hat-like interactions can explain long-range correlations in spontaneous activity (10), but their model did not account for afferent processing, and hence could not demonstrate the direct correlation of the spontaneous activity with orientation maps. It is important to emphasize that even if during development long-range connections might not be necessary for inducing long-range correlations, this does not imply this is also the case in the adult cortex. For example, the cortex is much more driven by the afferent pathway in adult animals than in the young in which visual input is ineffective in evoking reliable neuronal responses (10, 48). This is further corroborated by modeling in Smith et al. (10) that shows that the long-range correlations emerge only if the model dynamics are strongly dominated by lateral interactions, i.e. when the afferent input is about 1/100th of the lateral interaction strength. Under the weaker lateral interactions in the adult condition, the long-range connections might be necessary for the emergence of the long-range correlations in spontaneous activity, as our model suggests.

We didn't add reference to one of the studies you suggested (Muller et al. 2014) as in this case the traveling waves are generated during evoked activity which is out of the scope of this study, and as the architecture of that model is in any case similar to Muller et al. 2012.

At the same time there is a lack of discussion about how the integrative model considered in the manuscript can account for diverse phenomena. Particularly, how does the model

account for two different accounts of spontaneous activity in visual cortex: (1) cortical waves as refs 7 & 18, which are dynamic and changing over 10 to 100 ms, (2) modular spontaneous activity, as in ref 10, that changes very little over 100s of ms (e.g. Supp Fig. 2c in ref 10)?

Thank you for pointing out this opportunity for offering additional insight from our model that we have now incorporated in the paper (lines 502-511):

We find that the observed difference in the timescales of cortical waves and modular spontaneous activity are actually not due to the underlying neural activity, but rather due to the temporal properties of the calcium imaging signal which is used in the spontaneous structured activity analysis. In Figure 5B we show the strong temporal smoothing effect of our model calcium imaging construct, which we set based on the temporal kernel of the calcium imaging response measured by Lützke et al. 2013 (as described in Methods 2.5).

This explanation is corroborated by experimental findings in a recent preprint (Kettlewell et al. 2024) from the lab of G.B. Smith which explores this issue in further detail. The authors removed the temporal smoothing effect by applying the Prior Frame Subtraction (PFS) technique to deconvolve the calcium imaging signals. We compared the temporal autocorrelation functions calculated from the time courses extracted from Figure 1A,C of Kettlewell et al. with the temporal autocorrelation functions of model time courses under three conditions: (1) with simulated calcium imaging (SCI), (2) with SCI without convolution with the temporal kernel, and (3) SCI after PFS deconvolution (see the added panel Figure 5C and below). PFS deconvolution produces model results similar to those obtained without convolution, suggesting that any major differences between autocorrelation functions are not methodological artifacts. The deconvolved time courses in both ferret and model are of a similar time scale, with the first autocorrelation peak occurring at ~226ms in ferret and ~200ms in our model.

We added this comparison to the results section detailing the temporal smoothing effect of calcium imaging (lines 504-510):

Notably, this temporal smoothing effect of calcium imaging has been recently demonstrated by Kettlewell et al. (36), who deconvolved the ferret calcium imaging spontaneous activity recordings using Prior Frame Subtraction (see Methods), revealing faster temporal dynamics. Our model spike data matches the timescale of the example deconvolved signal provided in their article (36), with the first peak of the temporal autocorrelation function occurring at ~226ms in ferret and ~200ms in our model (Figure 5C).

Kettlewell, L., Sederberg, A. & Smith, G. B. Stereotyped spatiotemporal dynamics of spontaneous activity in visual cortex prior to eye-opening. (2024) doi:10.1101/2024.06.25.600611.

A final concern is that the study ultimately does not clearly describe a functional/computational role for the long-range orientation-biased connectivity, that is the key component of the model. This is despite this being an implied aim of the study: “Yet, a systematic understanding of how this common substrate supports the diverse computations involved in visual integration, and under different cortical regimes, remains elusive” (line 47)

The possible functional/computational role of long-range orientation biased connectivity has been already widely hypothesized in broad-range of studies (Field et al. 1993; Das & Gilbert 1995; Angelucci & Bressloff 2006;) with number of models (including ours) demonstrating their possible involvement in specific phenomena/computations (Li et al. 1998; Somers et al. 1998; Rao & Ballard 1999; Cai et al. 2005, Chariker et al. 2016; Davis et al. 2024, Antolik et al. 2024).

The main goal of the present study was to unify across an unprecedented range of phenomena spanning 3 cortical driving regimes with a single unified cortical circuit with particular focus on the lateral connectivity, hence providing a unified mechanistic explanation for all these phenomena. Having reviewed our abstract and introduction we feel this has been clearly explained in the introduction and abstract with the last paragraph in the introduction very specifically outlining all achievements without making any unfounded claims.

We acknowledge there is more work to be done to translate the understanding of the common substrate and the various emerging phenomena into understanding of which computations are performed (and which are not!) by early visual systems to integrate visual information, and turn this into a unified computational theory of early visual processing. But having re-reviewed the manuscript text we argue that:

1. We did not make any claims to attempt this ultimate goal in this single study. Note that the abstract does not mention function/computation at all and in the introduction the cited sentence is the only one containing the word computation.

2. The present study does represent a **major and necessary** step towards such an ultimate goal by providing a unified model spanning many phenomena and dynamic regimes in which the exact computational roles of the substrate and its emerging phenomena can be studied in a holistic manner.

The role of the specific sentence cited by the reviewer was to establish the existing gap (the lack of systematic integration across phenomena and different dynamical regimes all supported by a single substrate). The use of the word 'computation' was to point out that ultimately all these targeted phenomena link to computations which makes it important to study. That said, to prevent the slightest chance of misinterpretation of our claims, we have now replaced the word 'computation' in the cited sentence with 'phenomena' (line 49).

Specific comments:

102: With respect to the emergence of complex cells in L2/3, what is the mechanism by which complex achieve the spatial phase insensitivity in L2/3? Please provide evidence.

In the model, complex cells in L2/3 arise mainly due to how the feedforward hierarchy in V1 is constructed. Layer 4 is predominantly occupied by simple cells, whose afferent connections from thalamus are constructed based on Gabor templates with orientation preference following orientation maps but phase being diverse. Some complex cells can emerge in layer 4 due to the pooling of inputs via local lateral connections from other simple cells with a variety of phases but similar retinotopic position, but these will be a minority due to the functional biases of the push-pull connectivity. Next, there are narrow feed-forward connections from layer 4 to layer 2/3 which feed from neighboring cells in layer 4 sharing similar orientation tuning preference and retinotopy. Thus, layer 2/3 cells will predominantly pool from layer 4 neurons of similar retinotopy and orientations, yet varying phase as explained in Methods 1.4. As a consequence, layer 2/3 cells pool their input across cells similarly tuned to orientation but with a wide variety of phase preference profile, therefore inducing spatial phase insensitivity. Such a mechanism of pooling input from simple cells tuned to different spatial phases was already proposed by Hubel & Wiesel 1962 and Movshon et al. 1978. This mechanism is further described in the subunit model in Carandini 2006 (see Figure 1B). This hierarchical model is consistent with experimental findings indicating that complex cells' responses display higher phase modulation for very low contrast stimuli (Van Kleef et al. 2010) and the bias towards simple cells in granular and complex cells in supra-granular layers.

94: Does the feedforward organisation include distinct ON and OFF regions (Jin, J. Z., Weng, C., Yeh, C. I., Gordon, J. A., Ruthazer, E. S., Stryker, M. P., ... & Alonso, J. M. (2008). On and off domains of geniculate afferents in cat primary visual cortex. *Nature neuroscience*, 11(1), 88-94.)?

The LGN model has distinct ON and OFF-center populations. We reformulated the sentence in line 912 to explicitly indicate this:

ON and OFF-center LGN neurons are modelled as leaky integrate-and-fire cells.

Regarding cortical neurons, each simple V1 cell exhibits separate ON and OFF subregions, and this has been extensively examined in our recent paper (Taylor et al. 2021) where we showed that in this respect the model very well conforms to intracellular recordings from the Hirsch lab examining this issue.

The model, however, does not model the systematical topological organisation of ON and OFF dominance across cortical surfaces as discovered in a series of studies from the Alonso group starting by Jin et al. 2008 that you mention. This fact is clearly declared in our previous studies from which the model here is derived. We now also explicitly declare it in the present manuscript (lines 952-954):

A limitation of the model is that we do not model the dominance of the OFF pathway and systematic variations of the ON-OFF pattern of RF following the orientation maps revealed in recent electrophysiological experiments (Jin et al. 2008, Wang et al. 2014).

We agree that these findings are important, and we have an ongoing project to incorporate them into the V1 model, however it is important to note that the range of experimental findings and associated model changes this series of studies entails warrants a fully dedicated paper (or several!) and is out of the scope of the present manuscript. Crucially, while important, these features of cortical architecture do not have significant implications for the present study that focuses on ongoing and externally evoked activity that does not involve the thalamo-cortical pathway.

Taylor, M. M., Contreras, D., Destexhe, A., Frégnac, Y., & Antolik, J. (2021). An anatomically constrained model of V1 simple cells predicts the coexistence of push-pull and broad inhibition. *Journal of Neuroscience*.

94: The smaller text in Fig 1A is hard to read.

Thank you for notifying us about this oversight, we increased the smallest fonts across the figure.

99: typo: "As *with* its predecessor,..."

The typo is now corrected.

Figures S1-S5: The one-sentence description of these figures in the main text is far too brief. While I appreciate that these figures are reproducing previous results, some additional text in supplementary material guiding the reader through what is depicted is necessary.

Comparison to experimental data is frequently lacking in these. There are no methods relating to these figure, so it is difficult to know what is being reported or why. If this is relying on publications of previous versions of the model, this should be made clearer.

We apologize for the lack of clarity of the description of these supplementary figures. These figures are indeed a direct replication of analogous figures from our earlier paper where quantitative comparisons to population averages was discussed in detail. We added the following additional text to guide the reader better and refer to the publication of our previous model version:

S1

The activity of the model in spontaneous state is low and exhibits irregular-asynchronous dynamics, membrane potential resting state of about 70mV and overall realistic subthreshold signals. Refer to Antolík et al. 2024 (Fig. 4A-F) for an extensive description of these analyses and for comparison to the previous iteration of the model (Antolík et al. 2024).

S2:

The orientation tuning curves are computed from the neurons' spiking activity in response to sinusoidal drifting gratings. In all layers, the orientation selectivity of the neurons of the model, measured with the Half-Width at Half-Height (HWHH) of the orientation tuning curves, is contrast-invariant (Finn et al. 2007). Inhibitory neurons remain well tuned with only slightly broader tuning than excitatory ones in line with experimental data (Cardin et al. 2007, Nowak et al. 2008). Refer to Antolík et al. 2024 (Fig. 6AB) for an extensive description of these analyses and for comparison to the previous iteration of the model (Antolík et al. 2024).

S3:

All subthreshold signals exhibit orientation tuning. F0 and F1 refer to the DC and fundamental Fourier components of the neurons' responses to optimally oriented drifting gratings in line with previous intracellular recordings in cat (Anderson et al. 2000). Refer to Antolík et al. 2024 (Fig. 7) for an extensive description of these analyses and for comparison to the previous iteration of the model (Antolík et al. 2024).

S4:

The modulation ratio of the neurons was computed following a drifting sinusoidal gratings protocol. It exhibits a clear bimodal distribution, although with less overlap than observed experimentally in cat and macaque V1 (Ringach et al. 2002, Priebe et al. 2004). Layer 4 is predominantly populated by simple cells and Layer 2/3 is populated by complex cells. Refer to Antolík et al. 2024 (Fig. 8A) for an extensive description of these analyses and for comparison to the previous iteration of the model (Antolík et al. 2024).

S5:

Neurons' responses to a natural movie stimulus with simulated eye movement (NI) exhibit higher reliability and precision than to drifting gratings stimuli (DG). The trial-to-trial cross-correlation response profiles are in good qualitative agreement with the experimental data of Baudot et al. (2013), except for layer 4 membrane potential response which exhibit a marginally larger peak for DG than for NI. When pooled across the entire model, the inverse of the standard deviation of the response of neurons relative to spontaneous activity ($1/SD$) has a profile corresponding to the data of Baudot et al.. Refer to Antolík et al. 2024 (Fig. 10) for an extensive description of these analyses and for comparison to the previous iteration of the model (Antolík et al. 2024).

Fig S2: what are the dotted lines?

Thank you for pointing out the omission. We have now added the following sentence in the captions of Figure S2:

The baseline levels of activity are marked with dashed lines.

Fig S3: Why use a different color scheme for contrast to that used in Fig S2. Does F0 and F1 refer to the temporal Fourier components to drifting gratings? What about complex cells? It is not clear that complex cells are the L2/3 cells, without referring to ref 23.

We agree with the reviewer that Figures S2 and S3 should ideally have a matching color scheme. We however prefer to not modify these, as the color schemes match the corresponding Fig. 6 and 7 in Antolík et al. 2024 to which we refer the reader here for details. We are concerned that changing the color schemes in the new manuscript would cause confusion. We regret that we made this imperfection in our previous paper.

The F0 and F1 indeed correspond to the temporal Fourier components in response to drifting gratings. As mentioned earlier, we have modified the caption of Figure S3 to indicate this explicitly:

All subthreshold signals exhibit orientation tuning. F0 and F1 refer to the DC and fundamental Fourier components of the neurons' responses to optimally oriented drifting gratings in line with previous intracellular recordings in cat (Anderson et al. 2000).

In this figure we average the neurons' responses depending on their layer of origin, and not based on whether they are simple or complex. The intended role of this figure is not to indicate that layer 2/3 cells are complex cells. This property of the model is better illustrated by Figure S4. Instead the goal of this figure is to quantify the magnitude and tuning of the F1 vs F0 components per layer. Note that most cells are not purely simple or complex, thus each contributing to both F0 and F1 components, but of course the ratio of these contributions varies by neural type (simple vs. complex) and layer.

However, to clarify the fact that layer 2/3 cells are complex cells, we have added the following caption in the caption of Figure S4:

Layer 4 is predominantly populated by simple cells and Layer 2/3 is populated by complex cells.

Fig S4: The distribution of F1/F0 by layer is more binary than experimental data.

We indeed acknowledge that there is greater separation between the F1/F0 indexes in model layer 4 and layer 2/3 than observed in comparable analysis in cat. To be fully transparent about this issue we have made the following addition in the caption of Figure S4: *It exhibits a clear bimodal distribution, although with less overlap than observed experimentally in cat and macaque V1 (Ringach et al. 2002, Priebe et al. 2004).*

Fig S5: A) Color code is not given. C) I would average cross-correlation should be very weak given that neurons are tuned to different orientations. What are the experimental distributions? F) Correlations for cat data do not match the model (E) well for negative correlations. G-J) How is 1/SD measured in percent? Percent of what?

Thanks for noting the lack of color code for panel A. Excitatory conductance is represented in red, and inhibitory conductance in blue. We added this information in the part of the caption relative to panel A.

The cross-correlation are computed trial-to-trial for each individual neuron, and not across different neurons following the analysis in Baudot *et al.* (Baudot *et al.* 2013). Moreover in this figure we analyze only the response of neurons tuned to the orientation of the drifting grating stimulus. To emphasize this, we have added 'trial-to-trial' in front of every occurrence of 'cross-correlation' in the caption of Figure S5. As already indicated in response to a previous comment, we have also added further explanatory text to the caption. Unfortunately Baudot *et al.* do not provide distributions to which we can compare.

In the description that we added in the caption of this figure, we now acknowledge that we do not match the experimental data when it comes to the reliability of layer 4 membrane potential (this has been also discussed in depth in the original model paper):

The trial-to-trial cross-correlation response profiles are in good qualitative agreement with the experimental data of Baudot et al. (2013), except for layer 4 membrane potential response which exhibit a marginally larger peak for DG than for NI.

That said it is worth pointing out that Baudot *et al.* do not provide detailed information of their experimental protocol, for example the contrast of the stimuli and the number of repetitions of stimuli presentations (which differ from cell-to-cell). All these experimental parameters would affect the exact quantitative results of this analysis. In Figure S5E we observe negative correlations for the drifting gratings stimuli when there's a time lag approaching 250 ms, when the gratings are getting in antiphase (as the period is 500 ms)), just as in the experimental data shown in Figure S5F. In both model and experimental data, there is no correlation when the naturalistic movies have a time lag. We therefore consider that we are in good qualitative agreement with the data when it comes to the presence or the absence of negative correlations. Note also that Baudot *et al.* stimulated with gratings drifting at the optimal temporal frequency, which is cell dependent (2 to 3 Hz in their study). This makes quantitative comparison of negative peaks impossible as their data is therefore pooled across cells whose negative peaks take place within a broad temporal range. We have extended the y-axis of Figure S5CDE in order to make negative correlations visible in those panels.

The description of 1/SD was indeed incomplete, it represents the inverse of the standard deviation of the response of neurons relative to spontaneous activity, which is why it is quantified in percentage. We have modified the caption so that 1/SD is properly defined:
(G) *Inverse of the standard deviation of the response of neurons (1/SD) expressed in percentage relative to spontaneous activity for DG (red) and NI (black) stimuli averaged across all record excitatory neurons in layer 4.*

Fig 2A: What is spatial scale? Color map scale? It would be good to have these available as movies in supplementary material. Some experimental reference would be helpful. It is not clear these are waves. Or more importantly, are they similar to experimentally recorded waves.

Thank you for pointing out the missing spatial scale and colorbar, which we now added on the right of Figure 2A. We also added a video of this example as supplementary material,

and have added two more frames for this example wave in Figure 2A. The caption of Figure 2A now contains a reference to Supplementary Video 1.

We agree that it is important to compare the waves produced by the activity of the model with experimental data. This is what we try to achieve when comparing the wavelengths and propagation speeds distribution to the experimental values of Davis et al. 2021, which investigated thoroughly the properties of mesoscopic waves in the visual system.

134: What is the band of the wideband filtered signal? This should be in the main text. In the Methods it is given as 5-100 Hz, whereas in the ref 7 it is given as 5-40 Hz. The filter applied in ref 7 was also a zero-phase 8th-order Butterworth filter, by which I assume it is meant that it is applied in backward as well as forward direction to prevent phase distortion. However, in the current methods, no mention of forward-backward filtering is mentioned.

As requested by the reviewer, we have now added the band of the filtered signal in the main text at lines 116-117

The signal was then wide-band filtered from 5 to 100 Hz.

We bandpass filtered our LFP signal between 5 and 100 Hz because the same filter parameters were used by Davis et al. 2021 (ref 18, see Methods - Analysis of spatiotemporal dynamics) for their computational model data. However, the reviewer is right that the same authors used a 5-40 Hz bandpass filter when processing their experimental recordings (ref 7 and 18). We do not know the reason for this discrepancy. For full transparency, we have repeated our analysis with a 5-40 Hz 4th order butterworth filter:

Figure S19: Dependence of the LFP signal properties on the range of frequencies used in the band-pass filter. (A) The average wavelengths for model layer 2/3 for band-pass filtering ranges of 5-100 Hz (red, full) and 5-40 Hz (red, hollow), and for the experimental data of Davis et al. (2021) for marmoset MT (blue), as computed with the Generalized phase analysis. **(B)** Same as (A) for the average propagation speeds. **(C)** The average of the spike-triggered LFP residuals for model layer 2/3 for band-pass filtering ranges of 5-100 Hz (red, solid) and 5-40 Hz (red, dashed) across reference neurons and recording sites, binned according to their orientation preference differences. Error bars: 95% confidence interval.

As can be seen here, changing the bandwidth of the filter has only a marginal impact on the wave properties under scrutiny in this manuscript. There is still a significant propagation bias

of the traveling waves and its magnitude is only weakly reduced. Conclusions drawn from our results are therefore not altered by the choice of frequency bandwidth. We now added this figure as Figure S19 and have added the following sentence in methods at lines 1146-1147:

Our results are however not significantly affected if using a narrower band of frequency in the filtering (Figure S19).

The reviewer is also correct in that Davis et al. apply forward-backward filtering, which we indeed do as well to avoid phase distortion. We explicitly mention this now in Methods at line 1145.

135: Why is this sentence suddenly referring to firing rates when this section is on the LFP? Is this the data plotted in Fig 6a and 6b in ref 18? How were spikes binned to calculate rates?

We thank the reviewer for pointing out that the formulation of this paragraph is confusing. The purpose of this sentence was to indicate that we ensured that our model was in a similar regime of activity as in the experimental data, which is indeed shown in Fig. 6AB of ref 18 (Davis et al. 2021). We moved this sentence at the end of the paragraph and modified it so that it now reads as (lines 244-247):

The peaks of both the distribution of firing rates and coefficients of variation (CV) match well the experimental data from marmoset MT by Davis et al. (Figure 2GH), ensuring that our network is in a realistic regime of activity similar to that of the animal visual cortex.

We didn't perform any spikes binning to compute the firing rates, we simply divided the total numbers of spikes for each neuron by the length of the simulation.

Fig 2F and 2G: These data for ref (7) are for area MT not V1.

The wavelengths and propagation speeds data that we were comparing our model to were indeed recorded in area MT in marmosets and not in V1. We made this choice because Davis et al (2020,2021) provide the most complete data regarding these two metrics, with full distributions we can compare our model to. We would not expect any major difference between V1 and MT regarding that the general hypothesis is that the propagation speeds of the traveling waves reflect the underlying conduction speed of unmyelinated axons which shouldn't exhibit large variations across cortical areas. Explicit indications that this data originated from MT and not from V1 was lacking, and we now indicate this in lines 181,190, 207, 214 and 245. We however acknowledge that it'd strengthen our manuscript to provide direct comparison between our model and V1 data. Together with the generalized phase analysis and comparison with the Davis et al. data, we now also provide a direct comparison with the cat V1 data stLFP propagation speed reported by Nauhaus et al. (2009), which we computed in the same way as them by fitting linearly the time to peak of the stLFP with the distance between the recording sites and the reference neurons. They find stLFP propagation speeds of 0.31 m/s (SD: 0.22 m/s) while the mean model value (mean: 0.52 m/s, SD: 0.25 m/s) is within the range of one SD from the experimental data. As a consequence, we have now reworked the paragraphs from this results subsection, so that

we first present the comparison to the Nauhaus et al. (2009) study and then compare the wavelengths and propagation speeds as computed with the GP analysis to the Marmoset MT from Davis et al. (2020,2021) (lines 120-247):

We next wished to study the relationship between spontaneous wave propagation and the intrinsic functional maps identified by Nauhaus et al. in cats and macaques (Nauhaus et al. 2009). They used spike-triggered LFP (stLFP), which enables investigating how spiking activity at one reference recording location affects the LFP activity across all other recording sites. They found that the delay of the peak of the stLFP increased and that the amplitude of the peak decreased with distance from the reference electrode. We repeated their analysis using the spike trains of 207 model neurons as reference to compute the stLFPs (see Methods). As in the experimental data, we observe that the delay of the peak of stLFP increases with distance between recording sites (Figure 2B), and that its amplitude decreases exponentially based on distance (Figure 2C), confirming the validity of the stLFP method implementation in our model. The mean propagation speed of the stLFP signal is 0.52 m/s, within one standard deviation from the value of 0.31 m/s reported in cat V1 by Nauhaus et al. (Figure 2D). For each reference electrode, Nauhaus et al. used an exponential function to fit the relationship between the stLFP amplitude and the distance to the reference electrode, and then computed the stLFP residuals by subtracting the value predicted by the exponential fit from the stLFP amplitude at each electrode. We applied the same fitting procedure (Figure 2C) and computed stLFP residuals for all but two reference neurons for which the fitting performances were poor (see Methods). Crucially, Nauhaus et al. observed significantly higher amplitudes of stLFP residuals at recording sites with orientation preference congruent to the reference electrode, than for sites with orthogonal preference, indicating functionally specific wave propagation. We find that, for a given reference neuron, iso-oriented recording sites has significantly higher stLFP residuals than recording sites in orthogonal domains (Iso-oriented: 0.0124 ± 0.0008 , ortho-oriented: -0.0129 ± 0.0008 , $p = 4.74 \times 10^{-31}$, $n = 205$, two-sided Wilcoxon signed-rank test) (Figure 2E). This is inline with the observations of Nauhaus et al., although the magnitude of the iso-orientation bias they report is higher. Overall these findings indicate that cortical STW propagate preferentially towards iso-oriented domains in the model as observed experimentally, confirming that we successfully account for functionally specific spontaneous V1 dynamics. Furthermore, we performed the same analysis after removing the functional bias in the long-range excitatory connectivity in the model layer 2/3. After this manipulation, the stLFP residuals are no longer higher for iso-oriented recording sites than for orthogonal recording sites (Iso-oriented: -0.0020 ± 0.0008 , ortho-oriented: -0.0001 ± 0.0008 , $p = 0.07$, $n = 201$, two-sided Wilcoxon signed-rank test) (Figure 2E), demonstrating that functionally specific connectivity is critical for the iso-oriented bias of STW propagation.

Next, Davis et al. offered a detailed characterization of the spontaneous wave properties in marmoset MT using the Generalized Phase analysis (Davis et al. 2020, 2021). We applied the Generalized Phase analysis to compute the instantaneous phases of the wideband filtered signal (Figure 2F), yielding the distribution of wavelengths and propagation speeds of the synthetic LFP (see Methods) which we compared to Davis et al. marmoset data. Prior to convolution, the wavelengths of our synthetic LFP signal are significantly higher than in the spatially shuffled signal, demonstrating that the signal is indeed spatially structured (Figure

2J). After applying the Gaussian convolution to account for spatial pooling, the distribution of the wavelengths in the signal closely matches experimental data in marmoset MT at high and intermediate wavelengths. However, we observe a greater proportion of low wavelengths, likely due to the limited simulated cortical area and the absence of extra-areal input which increases global network synchronisation. The distribution of propagation speeds also lies within the inter-subject variability identified by Davis et al. (Figure 2K). The peaks of both the distribution of firing rates and coefficients of variation (CV) match well the experimental data from marmoset MT by Davis et al. (Figure 2GH), ensuring that our network is in a realistic regime of activity similar to that of animal visual cortex.

We also have now reorganized the panel orders in Figure 2 to account for these changes and have added the following figure as the newly added panel D:

(D) Propagation speeds of the stLFP in cat V1 (black) from Nauhaus et al. (Nauhaus et al. 2009), and for the model. Error bars represent the standard deviations.

175: Some introduction to the importance of stLFP would be helpful.

We added the following sentence at lines 121-122:

They used spike-triggered LFP (stLFP), which enables investigating how spiking activity at one location affects the LFP activity across all other recording sites.

177: What is meant by an stLFP residual? Without any mention of trying to predict LFP based on spikes, it is not clear what is meant.

We improved the text so that lines 156-161 now read as:

For each reference electrode, Nauhaus et al. used an exponential function to fit the relationship between the stLFP amplitude and the distance to the reference electrode, and then computed the stLFP residuals by subtracting the value predicted by the exponential fit from the stLFP amplitude at each electrode. We applied the same fitting procedure (Figure

2C) and computed stLFP residuals for all but two reference neurons for which the fitting performances were poor (see Methods).

This text refers to Methods 2.4 which also contains a description of stLFP at lines 1174-1177: *We then computed the stLFP residuals by subtracting the expected stLFP amplitude (determined by the exponential fit) at each electrode from the actually measured amplitude. We assigned an orientation preference to each recording site based on the pre-computed orientation map of the model.*

252: typo: "stimulation from *the* cortical surface"

It is now corrected.

Fig 2I and 2J: Why not compare to experimental data from Fig. 2 in ref 8?

The data shown here is specific for stLFP computed from a single reference recording site, and the same is true for the data shown in each row of Figure 2 in Nauhaus et al. (Nauhaus et al. 2009). There is no point in comparing data points from one arbitrary electrode in the model to another arbitrary electrode from in-vivo experiment due to the large variations from site to site. The purpose of panels 2B and 2C (previously 2I and 2J) was to illustrate that just as in the data, in the model the time to peak of stLFP increases with distance from the reference site, and that the stLFP amplitude decreases exponentially with distance from the reference site. However the reviewer is right that a clear description of this fact was missing in the text, so we added the following sentence in lines 151-154:

As in the experimental data, we observe that the delay of the peak of stLFP increases with distance between recording sites (Figure 2B), and that its amplitude decreases exponentially based on distance (Figure 2C), confirming the validity of the stLFP method implementation in our model.

Fig 2I: Why the large number of points with ~0 ms time-to-peak across most distances up to 3.5 mm?

We are grateful to the reviewer for noticing this anomaly! For calculation of stLFP in our model we relied on the reference weighted sum LFP proxy method from Mazzone et al., 2015. This method suggests taking the contributions of excitatory currents with 6ms delay, which, as it turns out, is the source of this anomaly. Indeed, in the case of our stLFP analysis, this induces a 6 ms lag between the spiking times and the LFP signal. Thus, for many recording sites the actual peak of stLFP occurs before the spiking time of the reference neurons. Given that our stLFP analysis takes into account only the 20 ms subsequent to those spiking times, the stLFP peaks take place at the first time point after the reference neuron's spiking times.

This is illustrated by the following figure, with each plot representing the stLFP at a given recording site in respect to the spikes of one given reference neuron (location marked in red, same reference neuron as for Figure 2BC, previously Figure 2IJ). Note that when creating

this figure we further downsampled the number of recording sites by a factor of 2 to improve its clarity, so each recording site shown here is separated by 800 μm :

At many recording sites, even distant ones such as the upper left one, the stLFP peaks occur before the spiking times. However, if, instead of using the weighted sum proxy, we use unweighted sum of excitatory and inhibitory currents with no time delay applied (eq 2 in Mazzoni et al. 2015), this anomaly disappears:

In this figure it is clear that more distant recording sites tend to have more delayed peaks and lower amplitudes. For many recording sites, the stLFP rises before the spiking times of the reference neuron, but this 'non-causal response' coincides well with the observations of Nauhaus et al. (see their Figure 2, right column).

It is important to point out that the 6ms delay choice in Mazzoni et al. was an ad-hoc choice to fit reference multi-compartmental simulation data, not grounded in any biological or theoretical basis. The aim of Mazzoni et al. was to build a synthetic LFP signal from the response of a point-neuron current-based model which would fit best the LFP signal generated by the transmembrane currents of a multi-compartmental mode with similar parameters. There are several details in Mazzoni et al. which convince us that using this ad-hoc parameter would be arbitrary and not well justified in our case. For example Mazzoni using current based vs. us using conductance based LIF neurons which necessarily has an impact on the time course of the amplitude of postsynaptic currents. Additionally, they use different decay time constants for excitatory synaptic current across their point neuron (2 ms, Table 2) and multi-compartmental (1 ms, Table 4) models, as well as different synaptic

weight values, which all can have a considerable impact on the need of such a delay correction factor and its exact numerical values. Importantly, Mazzoni et al. show that the sum of absolute input currents also makes a very good proxy for LFP signal generated from pyramidal cells, and showed that it provides a good match to spectral properties of LFP signals recorded in the cortex (Mazzoni et al., 2008). Thus, we have made the choice to use this proxy to synthesize the LFP signal in the layer 2/3 of our model in the revised version of our manuscript.

In the newly updated Figure 2C, we thus do not observe anymore distant recording sites with time to peak close to 0ms:

We have therefore updated the methods at lines 1105-1110

To synthesise LFP signal from our point neuron population of L2/3 excitatory neurons we used a proxy method that approximates an LFP signal as the sum of the absolute values of the excitatory and inhibitory currents to estimate the participation of single neurons in the LFP signal of the population (Mazzoni et al. 2015).

These modifications altered the LFP signal; we have therefore also updated Figure 2 following these changes. Note that ultimately, these changes in the analysis only marginally affect the distributions of wavelengths and propagation speeds of the LFP signal. It has only a minor impact on the values of stLFP residuals in the model, which still exhibits significant iso-orientation propagation bias as shown in the newly updated Figure 2E.

Fig 2K: Why not compare to experimental data from Fig. 4 in ref 8? The shape but not the magnitude is similar. Why?

It is correct that the magnitude of the stLFP residuals is smaller in our model than in experimental data. However, precise quantitative match is difficult to obtain given that we rely on a proxy of an LFP signal based on input currents of our layer 2/3 excitatory cells. Although Nauhaus et al. recorded in superficial layers, which is analogous to the layer 2/3 population of the model, these layers also contain apical dendrites of layer 5 cells whose participation in the LFP signal cannot be accounted for in the model. Finally, Nauhaus et al.

themselves observed a significant inter-individual variability as in one of the four animals they used for the experiment they do not observe a significant bias of propagation of traveling waves. Overall, given the very small magnitude of the effect (0.04) and the involved approximations we would not expect a perfect quantitative match.

For full transparency, we now explicitly acknowledge the fact that in the model the magnitude of the propagation bias is smaller than in Nauhaus data at lines 168-169:

This is inline with the observations of Nauhaus et al., although the magnitude of the iso-orientation bias they report is higher.

We also now added the following in the discussion (lines 853-858):

Apical dendrites of layer 5 cells extend up to the supragranular layers and participate in the generation of the LFP signal in these layers. Their absence in the model therefore impacts the magnitude of the LFP signal and might contribute to the discrepancy between the magnitude of the stLFP residuals of our model and the ones observed in Nauhaus et al. data (Nauhaus et al., 2009).

We have also modified Figure 2E (previously Figure 2K) such that it includes the experimental data from Nauhaus et al.. Consequently to this change, we have modified the error bars such that they now represent the standard error of the mean as in Nauhaus et al.. We have adapted its caption accordingly.

In the text it is stated that the stLFP analysis is made with 269 neurons, but in the Methods the number is given as 316.

Thank you for your vigilance. This discrepancy arises from the fact that, as stated at lines 161-162, we exclude from further analysis reference neurons for which the relationship between amplitude at recording sites and distance from recording sites could not be fitted well with the exponential (using explained variance < 0.5 as a criterion, similarly as Nauhaus et al 2009) In the main text of the article, we reported only the number of reference neurons which were used for the stLFP residual analysis.

However, due to the correction in the computation of the LFP signal of the model which was made to address one of the previous comments of the reviewer, these numbers have to be updated. We now analyze the stLFP for 207 reference neurons (to roughly match the number of selected neurons in Nauhaus et al.), from which 2 are discarded due to poor fitting, resulting in 205 reference neurons whose stLFP residuals were analyzed. We have therefore updated those numbers in lines 126, 161, 167, 1166, 1173.

We also now have reformulated the main text so that it mentions the original number of reference neurons and we discarded two neurons from stLFP residual analysis in lines 125-161

We repeated their analysis using the spike trains of 207 model neurons as reference to compute the stLFPs (see Methods).

[...]

We applied the same fitting procedure (Figure 2C) and computed stLFP residuals for all but two reference neurons for which the fitting performances were poor (see Methods).

214: The cited study ref 10, is a developmental study performed soon after eye opening. As stated in this study: as the modular correlation maps “are present prior to the maturation and elaboration of long-range horizontal connectivity, these results also present a conundrum.” The authors demonstrated that local, but heterogeneous connections can explain the modular correlations maps. So, the patchy L2/3 connectivity emphasized in the current manuscript is not necessary and in fact only weakly present before eye-opening. At line 219 it is stated that “Crucially, the orientation map similarity diminishes when we remove the orientation bias of Layer 2/3 long-range horizontal connections (Figure 3E (right), 4F), further underlining the importance of the functional biased lateral connectivity in shaping spontaneous cortical dynamics.” Furthermore, the degree of similarity in correlation maps is not more than the experimental values with L2/3 orientation bias. In Fig 3 H left, it seems the model prediction are much higher than experimental values for E->Or (Note: it is not clear what E->Or means).

The model in ref 10 requires short range excitation and longer-range inhibition in cortico-cortical connections. In the model in the current manuscript, it is difficult to tell if this is also the case because the local inhibitory to excitatory connections have a hyperbolic distribution (Eq.4) while the local excitatory to excitatory connections have a Gaussian distribution (Eq. 7). Please clarify relative extent of local inhibition vs excitation. If inhibition does not spread further than excitation, then this regime should be explored.

In response to the major concerns of the reviewer above we have already justified the rationale why we compare our adult model to young animals, and why our study is not contradictory with the lack of long-range connections in young animals.

Let us now address the additional specific comments made by the reviewer here:

1. **On the higher similarity between correlation maps and orientation preference maps:** indeed we observe higher degree of similarity between orientation maps and correlation maps in our model than in the young ferrets. This could be simply because of the additional measurement noise in real experimental data that is not present in our simulations, decreasing the observed correlations in ferret. Alternatively, this could be explained by the fact that, as reviewer points out, recordings in (10) were done in young ferrets, whereas with increasing age the representation of orientation becomes more refined, as indicated in Figure 5C in (10), driving up the correlations between orientation maps and spontaneous activity. Hence our adult model predicts higher alignment between OR and correlation maps than that observed in young animals. We have now more explicitly stated this discrepancy and potential explanation in the manuscript text:

Finally, we quantitatively assess how well the model captures the spatial properties of in-vivo spontaneous activity using several key metrics of spontaneous V1 activity (10, 20) (Figure 3H), overall finding a good match with young ferret data (see Methods for a detailed description of all metrics). The model reproduces the degree of similarity of excitatory and inhibitory CMs, long-range correlations and correlation wavelength for both excitatory and inhibitory activity, while the inhibitory dimensionality and local correlation eccentricity lie only slightly out of experimental data ranges. Our model does exhibit somewhat higher similarity between spontaneous activity and orientation preference maps than observed in young ferrets, which could be due to the absence of measurement noise in the model. Alternatively, this discrepancy could be

explained by the young age of the imaged ferrets, whereby the orientation representation becomes more refined as the animal matures (see figure 5C in (10)), reflecting the stronger alignment between orientation maps and spontaneous activity we observe in the model of adult V1.

2. **On the lack of explanations for abbreviations in figure 3H:** we have now added the descriptions of all abbreviations to the corresponding figure caption.
3. **On clarifying the extent of lateral connectivity:** The spatial profiles of lateral connectivity in our model are set by cat V1 anatomical measurements. We set the spatial constants for the excitatory-to-excitatory lateral connectivity in Layer 2/3 based on the bouton density profiles in Buzás et al. (2006). For local connectivity, we use hyperbolic distributions that were fitted to the connection probability profiles in Stepanyats et al. (2008) obtained in cat V1. The process is described in greater detail in the publication of the previous version of this model (20). The resulting spatial distributions, added as Supplementary Figure S16 (also shown below), show that for both local-only and local + long-range excitatory connectivity, the local spatial connectivity profile of excitatory neurons is wider than that of inhibitory neurons. Since these profiles are directly constrained by anatomical data, exploring regimes where inhibitory connectivity extends further than excitatory connectivity would not be biologically justified. In fact, we see our adherence to anatomically realistic connectivity as a key strength of our adult model compared to the connectivity assumptions in Smith et al. (10), although it should be acknowledged that the connectivity profiles in young animals are experimentally less clearly determined.

Figure S16: Spatial extent of excitatory and inhibitory lateral connectivity in model Layer 2/3

The spatial constants for the excitatory-to-excitatory lateral connectivity in Layer 2/3 are based on the bouton density profiles in Buzás et al. (14). All other local connectivity, including the local connectivity of inhibitory neurons, is the result of fitting a hyperbolic distribution to the connection probability profiles in Stepanyats et al. (15). The resulting spatial distributions, thus constrained by anatomical data, show a wider local spatial connectivity profile for excitatory neurons than for inhibitory neurons.

208: It is stated that the methods of Smith et al, (ref 10) were followed, however, they explicitly screened for large-scale spontaneous events. No mention of this screening step is mentioned in the Methods of the current manuscript.

We thank the reviewer for pointing this methodological detail out. During our initial analyses, we discovered that in general, we obtained very similar results (see added Supplementary Figure S7 A,B, also below) whether we screened for large-scale spontaneous events or not, given long enough recordings to record a wide range of events. As such, for the sake of simplicity, we omitted this screening step. We now clarify this choice in Methods, lines 1201-1202:

For simplicity, we omitted the screening for large-scale events used by Smith et al. (10), as it had minimal impact on results (see Supplementary Figure S7).

The parameters of event detection (the threshold at which a pixel is considered “active” and the percentage of active pixels required for a frame to be considered active) had a minor effect on our metrics shown in Figure 3H (see Supplementary Figure S7C, blue), somewhat improving our match to experimental data. However, these parameters are strongly dependent on the noise properties of in vivo calcium imaging signals. Since our model does not include modeling such in vivo imaging noise, we would need to set these thresholds arbitrarily. To avoid this, we opted for the simplified, conservative approach of omitting event detection altogether.

Figure S7: Comparison of results calculated from entire activity time course vs. only large-scale event maxima

(A) Spatial correlation maps are similar whether screening for large-scale events is applied or not, on a 120s recording of spontaneous activity data. Screening is done as per Smith et al. (13), but with a pixel activation threshold of 2 s.d. and an active frame threshold of 0.8. The decreased pixel activation threshold (from 4-5 s.d. in Smith et al. (13)) is due to the lower levels of noise in our model compared to experiment. **(B)** Correlation similarity (see Methods) between non-event- and event-based correlation maps is consistently very high (mean=0.78 ± 0.001), regardless of seed position **(C)** Comparison of metrics measuring the spatial properties of spontaneous activity (see also Figure 3H). Most metrics stay within ferret cortex (13, 16) data ranges, regardless of screening for large-scale events. Omitting screening results in a slight weakening of spatial correlations (still within experimental

bounds), and increase in dimensionality (excitatory dim. within, inhibitory dim. outside experimental bounds).

259: Once again the study in ref 14 is a developmental study, in which “the long-range axonal connections that eventually linked correlated and co-tuned domain are still poorly developed.”(ref 12). The subjects are ferret pups prior to eyes opening, in which the long-range connections are believed to be poorly developed. So, the current model, which features such long-range connections, is not appropriate.

We have already provided earlier a general justification of why we compare our adult model also to studies in young animals and why our adult model is not in contradiction with lack of long-range connections prior to eye opening. We have now updated the manuscript text so that it is abundantly clear that we model adult V1 (lines 11, 59, 67, 84, 89, 287, 328, 330, 476, 725, 740, 826) and added discussion on the issue of long-range connections prior to eye opening (lines 817-833).

We would make a few additional remarks here:

1. There are very few studies of optogenetic stimulation in animals with columnar functional organisation (i.e. cat, ferret, macaque marmoset) of V1 available other than those we compare to here, which is why we also included studies in young animals.
2. Given that:
 - a. it is well known that structured spontaneous activity exists in adult animals
 - b. and we have now compared the model to additional **adult** studies (see added panels for Figure 3I,J,K,L) demonstrating that the model captures these adult spontaneous correlations
 - c. the model is otherwise well constrained on broad range of other phenomena

it is interesting to demonstrate that our adult model reproduces the phenomenon observed in young animals where, with homogenous stimulation, the similarity between OR maps and correlation maps remains similar to the case without stimulations. Since we know that the structured activity is present in adult animals, we would expect this phenomenon to be preserved in adults as well, justifying its investigation in our model.

3. To further strengthen the validation of our model on adult optogenetic data, we have now added a qualitative comparison to visual+optogenetic stimulation data in macaque by Chernov et al. 2018, into Figure 4 J-L (see also below). They combined optogenetic stimulation of a cortical column with visual stimulation either iso-oriented or orthogonal to the orientation preference column, and observed stronger response compared to visual-only stimulation for the iso- condition, and weakened response in the ortho- condition. We observe an analogous iso-facilitation and ortho-suppression in our model (Figure 4L). Unfortunately, we were limited to a qualitative comparison to the experiment results - converting model spiking firing rates to the intrinsic signal involves multiple unknowns such as local vasculature, species-specific hemodynamics, dura thickness and optical scattering. Thus, we could not quantitatively interpret the difference in the dynamic range of the observed effect (model firing rate range of 14-17 sp/s vs. experiment 0-0.15 dR/R).

Added panels to Figure 4 and caption:

(J) Schematic of Chernov et al. (33) experiment: simultaneous visual stimulation with full-field square gratings and optogenetic stimulation of an orientation preference column (circle, $r=100\mu\text{m}$), (K) Spatial map of regions with a significant (two-tailed paired t-test, for dependent samples; $n = 10$, $P < 0.05$) preference for visual stimuli iso-oriented (iso-regions, blue) or ortho-oriented (ortho-regions, red) relative to the optogenetically stimulated cortical column, within 2mm from the stimulation site. (L) (left) Optogenetic stimulation significantly (two-sided Wilcoxon signed-rank test, $p<0.006$, $n = 10$) increases response in iso-regions (blue) and decreases response in ortho-regions (red, $p<0.004$, $n = 10$) (right). Intrinsic imaging data from Chernov et al. (33) show a qualitatively similar effect.

Added Results section, lines 476-491:

To confirm that our model captures the optogenetically driven regime also in adult animals, we replicated the combined visual and optogenetic stimulation experiment of Chernov et al. (33), who recorded intrinsic imaging responses in adult macaques. Following Chernov et al., we optogenetically stimulated the horizontal orientation preference column in a circle of $100\mu\text{m}$ radius, while simultaneously visually presenting full-field drifting square gratings either iso- or ortho-oriented relative to the stimulated column (Figure 4J, see Methods). We also presented the same gratings without optogenetic stimulation, as the baseline condition. We then identified cortical positions significantly (two-tailed paired t-test for dependent samples, $n = 10$, $P<0.05$) preferring either the orientation of the optogenetically targeted column (iso-regions) or the orthogonal orientation (ortho-regions) (Figure 4K). We observed that the response in iso-regions increased significantly (two-sided Wilcoxon signed-rank test, $p<0.006$, $n = 10$) during optogenetic stimulation compared to the visual-only condition (Figure 4L, left), while the response of ortho-regions decreased ($p<0.004$, $n = 10$). Chernov et al. (33) observed an analogous iso-activity increase and ortho-decrease of the intrinsic imaging signal when stimulating the vertical orientation column (Figure 4L, right).

259: In ref (14) the pattern of activity during full-field optogenetic stimulation changes from trial-to-trial (Fig 2e). Does the current model replicate these results?

Yes, see the plots with optogenetic events (defined as the frame at opto-stimulus offset in (14)) and correlation analysis results below, now added as Supplementary Figure S9, which we now refer to on lines 434-437 of the updated manuscript:

Consistent with these results, the model full-field optogenetic responses were variable (Supplementary Figure S9) and the derived CMs closely matched spontaneous CMs (Figure 4C), irrespective of seed position (Figure 4D).

Figure S9: Model responses to fullfield optogenetic stimulation are variable between trials. (A) Optogenetically evoked events by fullfield stimulation (defined as the frame of opto-stimulus offset) are distinct (B) The sorted spatial correlations between optogenetically evoked events show considerable trial-to-trial variability (inter-stimulus interval = 60s).

71: “removing the orientation bias of long-range connections from the model decreased the similarity to chance level, demonstrating that the functional biases in later-connectivity mediate the link between orientation maps and optogenetically evoked activity”. Following my remarks for line 214, could the model in ref 10 without long-range connections also reproduce these results with full-field optogenetic stimulation?

Due to its level of abstraction, the model in ref 10 does not have a notion of orientation maps, as it does not model feed-forward processing in any way. Note that because of this, in ref 10 authors do not (and can not) reproduce in the model the similarity between orientation maps and cortical activity, be it spontaneous or optogenetically evoked. They can (and do) show only that their model has long-range correlations in the simulated spontaneous activity. This further highlights the range of conceptual advancement that our model brings in comparison to this earlier modeling.

275: Once again, the study in ref 12 is a developmental model of ferret, postnatal day 25-31, prior to eye opening. Again, the long-range connections can be expected to be weakly developed or even absent.

We already addressed this issue in response to L259 and earlier in the response to reviewer’s general comments.

279: I can find no mention of “surrogate stimulation pattern” in Mulholland et al (ref 12). The description pertaining for Fig 5 in ref 12 does not mention “surrogate stimulation patterns” nor any relationship to orientation maps. If this is the case, then the manuscript does not cite

any experimental work connecting optogenetic stimulation and orientation maps. This is problematic, because the next two subsections of the Results focus on the relationship between orientation maps and optogenetic stimulation. I appreciate that these two subsections can be considered to be predictions of the model. However, these predictions are then made without any data supporting link between optogenetic stimulation and orientation maps.

We sincerely thank the reviewer for catching this mistake. We inadvertently cited the wrong reference and regret this oversight. We intended to cite reference (13), a poster presentation presented at the 2024 COSYNE conference. The full abstract is available on the COSYNE website, as part of the program book, page 278, link: https://static1.squarespace.com/static/6102ca347474c263c40150cd/t/65e1abdbf843e41837fc9c0d/1709288389623/Cosyne2024_program_book.pdf. A very similar poster has also been presented in 2021 at the Bernstein conference.

We specifically refer to the following part of the abstract: *“We found that optogenetic stimuli derived from endogenous patterns of spontaneous activity evoked stronger average responses in targeted areas than artificial random patterns made from bandpass white noise. In addition, endogenous stimuli evoked responses that were more reliable from trial-to-trial and were more similar to their input pattern compared to artificial stimuli that had similar spatial frequencies.”*. As we did not have all the details of the bandpass white noise process, we used the endogenous stimuli rotated by 90° as surrogates, as they had much lower (0.1) similarity to the orientation preference map than endogenous stimuli (0.54).

803: Explain the 400um spacing of the LFP grid.

For the General Phase analysis, we originally computed our LFP in a 100 x 100 grid with 50 μm of spacing. This means that prior to the convolution of the signal with the gaussian kernel, each neuron contributed only to the 50 μm x 50 μm cell within the grid of the LFP signal corresponding to its position. For the stLFP analysis, to be consistent with Nauhaus et al. 2009 (ref 8) which recorded with LFPs with an Utah array, we only considered $\sim 1/8$ th of the channels of the grid, such the channels used for this specific analysis resulted in a 13 x 13 subgrid with 50 μm x 50 μm cells but with 400 μm of spacing. To make this clearer, we have now updated Method 2.2 so that it explicitly mentions that the spacing of the original grid was 50 μm (line 1117) and we modified the following sentence in Methods 2.4 (lines 1168-1170):

For each reference neuron, we then computed the stLFP as the spike-triggered average of the LFP signal over a 13 x 13 sub-grid of the original LFP grid, keeping a cell size of 50 μm x 50 μm but with now 400 μm of spacing.

807: Were the z-scores calculated in this analysis as per ref 8?

As indicated in Methods 2.2, the z-scores were computed using the mean and standard deviations across the whole array as in Davis et al. (2021). This is indeed unlike Nauhaus et al. (2009) who normalized their LFP signal per channel. We used the same way of z-scoring for both Generalized Phased and stLFP analyses for the sake of simplicity. As shown below z-scoring our LFP signal per channel doesn't impact the result of our stLFP residual analysis:

Fig 3C: give similarity index in caption or text.

We added the relevant similarity values into the caption for C and D.

Fig 3H: it is not clear what is plotted in Fig 3H either in the caption or the text. A more precise definition in the caption is required.

The reviewer is correct, we have added a clarifying statement for the meaning of axis labels to the caption.

309: Could you clarify how “activation” is being measured here.

We rephrased the relevant section for clarity, lines (451-455):

Model responses to stimulation with endogenous patterns matched the maximum-normalised population response time course of young ferret (12), both within and outside the stimulation ROIs (Figure 4H, left). This result suggests that our model can account for the spatio-temporal dynamics of V1 responses to patterned optogenetic stimulation.

317: A “winner take-all mechanism in layer 2/3” is mentioned. What is this mechanisms in the model? How is implemented or how does it emerge?

The reviewer is right in that the proposed soft winner-take-all effect of the recurrent amplification through long-range functionally specific connectivity is rather discussed later in the manuscript (lines 465-471) and not well described yet at this point. We rephrased this section to provide a smoother introduction to our hypothesis:

We hypothesised that this is because aligned stimulation amplifies network responses via functionally specific recurrent connectivity, while unaligned stimulation fails to effectively engage this circuitry. This recurrent amplification thus acts as a ‘soft winner-take-all’ mechanism across the orientation domain, amplifying the dominant orientations while inhibiting the rest, leading to stronger, more correlated responses.

352: The method use in the paragraph is not very clear, mainly because it is technical without explaining the basic idea.

We thank the reviewer for their comment, we found it indeed challenging to write this part succinctly yet clearly. We have now rewritten the DAOD and DAOD entropy sections such that it better introduces the idea behind the metric (lines 564-588):

We have shown that model responses to endogenous (OP map correlated) optogenetic stimuli are stronger than to equivalent surrogate stimuli. We hypothesise this is due to

orientation-biassed recurrent connectivity selectively boosting the activity of the dominantly active orientation, resulting in a sharper orientation tuning profile than would be expected from the underlying orientation preference map. To quantify how the population activity is distributed across the orientation domain during patterned stimulation, we introduced the measure of Distribution of Activity across the Orientation Domain (DAOD). For each frame of activity, we binned cortical positions by their orientation preference, summed the activity within each orientation bin, and normalised the resulting distribution to a unit-integral density function. For each of the 8 endogenous and 8 surrogate stimulation patterns, we calculated DAOD within the optogenetically stimulated regions and averaged it across all frames of the response and 5 trials (see Figure 5D for example patterns, Supplementary Figure S10 for all patterns).

To quantify the sharpening of DAODs over the distribution of orientation preferences in stimulated regions, we compared the Shannon entropies of these two distributions (Figure 5E), with lower entropies indicating sharper tuning in the orientation domain. For all of the 16 tested stimulation patterns (8 endogenous, 8 surrogate), DAOD entropy was lower than that of the distribution of stimulated orientations. The entropy difference between the DAOD and stimulated orientation distribution was significant compared to spatially shuffled controls ($p < 10^{-17}$, two-tailed Z-test, $n = 1000$; for all stimulation patterns, see Methods). This result supports our hypothesis that the network amplifies the response of the dominant orientation preference subpopulation, resulting in a more sharply oriented response than the orientation distribution of the stimulation pattern, suggesting soft-winner-takes-all dynamics often hypothesised in cortical networks (35).

354: What is mean by the Orientation Domain here?

We have rewritten the DAOD and DAOD entropy sections such that it better introduces the idea behind the metric (see above).

356: Should make it clear that the normalisation of the DOAD makes it a prob density function.

We added a clarification that the normalisation results in a unit-integral density function (lines 572-573).

359: What is the ROI being referred to here? The Orientation Domain?

Thank you for spotting this. The term "ROI" referred to the stimulated area (defined earlier in the manuscript). The revised version of the section should now better convey that all analyses are restricted to the stimulated area (see the rewrite of the DAOD section above).

361: While the entropy analysis is nice in some ways, the values are not intuitive to interpret. Giving the entropy values for Fig 5C would help, but additional supplementary graphs like Fig 5C and their entropy values would help more.

We added Supplementary Figure S10 (see below) with corresponding plots for all stimulation patterns, and referenced it in the Figure 5 caption

Figure S10: Distribution of Activity across the Orientation Domain (DAOD) compared to the distribution of stimulated orientations, all stimulation patterns. (A) Individual endogenous stimulation patterns (B) For all patterns, the Distribution of Activity across the Orientation Domain (DAOD) under the stimulated area is more sharply oriented than the distribution of orientation preferences (DOP) for the same area, suggesting the selective amplification of the dominant orientation by orientation-biased connectivity. (C,D) Same as (A,B) for surrogate stimulation patterns (E) For all patterns, Shannon entropy values of DAODs are lower than corresponding (DOP) entropies, signifying sharper histograms

Fig 6C: It is odd to measure orientation in radians in previous figures and then degrees here. It was indeed inconsistent, we thank the reviewer for spotting this. We have updated Fig 6C to use radians as well.

Fig 6E is not referred to in the text.

It was referred to in Discussion. We now added a mention of it to the results as well (lines 640-642):

Yet, the activity and the overall orientation selectivity of the excitatory response in the surround is non-monotonic (Figure 6D), in contrast with the monotonic increase of inhibitory activity (Figure 6E).

584: The ratio of LGN to L4 V1 neurons is much lower than biology, even for area centralis.

The reviewer is correct, this arises from the fact that we increased the cortical density of the model in respect to Antolík et al. 2024, but without further modifying the LGN density. Our simulation time is quite sensitive to the number of LGN cells and this was therefore to avoid our simulations being too computationally inefficient. However, please note that each layer 4 cells receive the same amount of feedforward connections as in Antolík et al. 2024, within a range centered on 140. This number was chosen based on cat data from Da Costa & Martin 2011. We therefore consider that the fact that the LGN/L4 ratio is low isn't an issue as what matters the most is that L4 V1 neurons receive input from a biologically plausible number of LGN cells. From previous empirical tests, we generally find very little impact of LGN density on model properties as long as it is not very low (well below the numbers used here).

631: In Eq.2, I take it that x and y are the positions of LGN neurons? How should the function g be interpreted here? It is not really a distribution, as it takes negative values and is not normalised. Is it a modulation of the synaptic conductance from LGN to L4? How does the position of the L4 neurons affect g ? – this is missing. Presumably this should be via determining the center of the Gaussian factor in Eq. 2. A pre-computed map for the orientation, θ , is mentioned, but no details are given. Please provide this.

Indeed x and y represent the **retinotopic** positions of LGN neurons but relative to the **retinotopic** position of L4 neurons. The reviewer is right that this was not clearly stated, so we have reformulated this in the following way (lines 937-942):

In all sheets neurons are populated at random physical positions that correspond to retinotopic positions assuming a magnification factor of 1 between LGN and cortex, corresponding to retinotopic eccentricity parafoveally of roughly 3° . The coordinates x and y represent the retinotopic position of the LGN neurons relative to the retinotopic center of the layer 4 cell. The phase preference ψ is generated randomly.

It is correct that this equation can result in negative values, we have now expanded the Methodology to fully describe how the Gabor template is used to generate the afferent connections (lines 945-948):

To generate connections from LGN cells to a given layer 4 neuron, we take the absolute value of the positive (ON cells) or negative (OFF cells) part of the template, renormalize it to form distribution, and sample from it the respective connections to ON or OFF cells.

The pre-computed orientation map is shown in Figure 1B. We took your comment into account and now refer to this figure in this paragraph. We provide here a more detailed illustration of the orientation map for the reviewer:

656: Eq(4) describes the connectivity pdf in terms of x , which is the “distance between two neurons”. In ref 23 the pdf is described in terms of x , y and z where x is cortical depth of presynaptic neuron, y is cortical depth of postsynaptic neurons and z is lateral (radial) displacement. So in Eq.4 in the current manuscript, is x the lateral displacement, or the combination of lateral and vertical (between layers) displacement.

We understand why the ubiquitous usage of ‘ x ’ in section 6.1.4 of ref 23 (Antolik et al. 2024) was confusing. We have now replaced ‘ x ’ by ‘ d ’ in both equations 4 and 7 so that it is clearer that it represents a distance in physical space and not a coordinate, and we modified line 982 so that it is explicitly stated that ‘ d ’ refers to the lateral distance between neurons.

784: It is not explicitly stated how the phase is calculated from the analytical signal. Nor how the Hilbert transform was calculated.

We have now modified lines 1147-1149 to indicate that the Hilbert transform calculation was based on fast Fourier transform. We have also added the following sentence to precise that the phase was calculated as the argument of the analytical signal (lines 1149-1150):

The phase ϕ of the signal was computed as the argument of the analytic signal.

796: Using a ‘*’ for multiplication is not generally accepted mathematical notation and not consistent with the rest of the manuscript.

We have removed the ‘*’ symbol from equations at lines 1021 and 1163.

874: I initially thought phase referred to spatial phase and direction referred to drift direction of a Gabor. Perhaps something can be said here to make it clear that phase and direction refer to an ambiguity of orientation. I.e. the self-organizing map can be shifted or reversed.

Thank you for pointing out the unclear phrasing, we adjusted it so that it can be more clear what phase and direction refer to here.

876: “between the estimated orientations and the orientation map” – can you clarify that the latter means the true orientation map

We updated the text to specify that we are referring to the true orientation map.

981: Can you provide a reference for “the light intensity falloff from the surface is exponential”?

We added Supplementary Figure S20 showing the exponential decrease of light intensity with depth (also see below):

Figure S20: Exponential light intensity falloff with increasing cortical depth, calculated using the simulations of the ‘Human Brain Grey Matter’ model implemented in the LightTools software, assuming emitted light at 590 nm.

Please note the exponential decrease with depth is a direct consequence of simulating light absorption and dispersion in a medium with optical parameters matched to cortical tissue.

Generally the mathematical notation in the manuscript should not use abbreviations like ‘wl’ for ‘wavelength’ as this is confusable with ‘w’ multiplied by ‘l’. There are many other instances of this type of problematic notation (e.g. LFP, ISI, CV etc.)

We now use the notation λ_{LFP} for the wavelengths calculated from the LFP signal with the Generalized Phase analysis. For the other instances of abbreviations, we believe that the issue arose because they were formatted in italic. We have now corrected that which should prevent any confusion.

Refs 6, 13 are incomplete

Thank you for noting this, the references in question have been fixed.

Reviewer #1 (Remarks on code availability):

The code is extensive and would require considerable time to review.

Reviewer #2

Reviewer #2 (Remarks to the Author):

Summary:

R{`o}zsa et al. investigate the relationship between recurrent connectivity and brain dynamics in cat primary visual cortex using computer simulations of a spatially organized spiking network model.

Structurally, the model focuses on layer 23 and layer 4, includes a retino-thalamic, orientation specific long-range connectivity in layer 23, and push-pull connectivity in layer 4. Dynamically, the model reproduces a broad range of experimentally observed phenomena. By varying the structure and other key parameters of the network model, the authors relate network structure to the exhibited activity in a mechanistic manner, and are able to generate predictions to be tested in future electrophysiological experiments.

Review:

Introduction:

L36 - L40

The authors establish a distinction between spontaneous traveling waves and module spontaneous activity.

Regarding the former they write

"[...] with propagation biases aligned with cortical functional organisation (8)" while for the latter they assert that

"emerging patterning in V1 are correlated with functional maps".

Later in the manuscript (L173ff) they write regarding reference (8):

"Crucially, they also observed significantly higher amplitudes of stLFP (spike triggered LFP, reviewer) residuals at recording sites with orientation preference [...]."

To my understanding this implies that also the traveling waves are related to the functional map of V1.

The authors should expand on the difference between spontaneous traveling waves and modular spontaneous activity. Otherwise, why not group the former as a part of the latter?

We appreciate this insightful question! We likewise find the question about the relationship between spontaneous travelling waves and spontaneous patterned activity exciting. To the best of our knowledge it has not been addressed yet, but our model offers the opportunity to do so. We have in fact already attempted to examine the question prior to submission, however, we did not find a short decisive answer that we could easily fit within the scope of the manuscript.

To address the question we analysed the wave properties (amplitude, wavelength and propagation speed) at different time points of the spontaneous activity as a function of the similarity of the given spontaneous activity snapshot to the orientation maps. Our hypothesis was that, if there is a link between the spontaneous waves and patterned activity we would expect to find at time points when spontaneous activity showed greater similarity to orientation maps:

- Increased wave amplitude
- Wavelengths approaching the spatial period of the orientation preference map
- Wave speeds approaching the propagation speed of long-range lateral connections

The results from our model are shown on the figure below, for a bin size of 10ms: Panel A shows a considerable positive correlation between firing rate and similarity to the orientation map, in line with results from Figure 3L and Figure 5F (previously 5E). This positive correlation is also shared by wave amplitude (panel B), although the effect there is considerably weaker. Panel C shows a slight decrease of wavelength with increasing similarity, while in panel D there does not seem to be a strong trend for wave speed. These trends seem to gradually weaken as bin size increases (panels E,F,G,H).

The results above partially support the hypothesis that travelling waves and structured spontaneous activity are linked:

- Wave amplitude in B (F) increases with similarity, as predicted. Firing rate in A (E) increases analogously, in line with the results of Tsodyks et al. 1999 shown on Figure 3L, and our modelling results from Figure 5F (previously 5E).
- Wavelengths show a decreasing trend with increasing similarity in the direction of the spatial period of the orientation preference maps

However, we ran into the following problems when interpreting these results:

- For C (G), the observed wavelengths are larger (min. wavelength~5mm) and never reach the spatial period of orientation maps (~1mm). This makes suggesting a mechanistic connection between the two phenomena problematic.
- For D (H), there does not seem to be any strong trend for wave speed with increasing orientation map similarity

It is important to point out that these results do not rule out the interaction of cortical travelling waves and structured spontaneous activity, and clearly they do coexist both in vivo and in our model. However, we could not identify a valid mechanistic explanation for the results above during this initial analysis. We were also concerned with the marked change of these trends for different bin sizes, pointing to the complexity of the relationship between these metrics, and the need to find more robust markers to compare the two phenomena. Given that the interpretation of our analysis cannot be made firmly enough, we are reluctant to publish it as a model prediction. Clearly substantially deeper analysis, preferably coupled with analogous analysis of experimental data would need to be done to satisfactorily address this question. This is beyond the scope of this manuscript, which already as it is greatly exceeds the length limits recommended by Nature Communications. We hope to work on this exciting question in the future in a dedicated manuscript.

L45 - L47

The authors write:

"Ultimately, visual spatial integration across these cortical regimes (spontaneous, visually driven, and optogenetically driven) relies on a shared neural substrate, especially the functionally specific long-range lateral connectivity in layer 2/3."

I do not fully understand what the authors want to communicate here. What is visual spatial integration during spontaneous and optogenetically driven activity? To my understanding, there is no visual spatial integration during these regimes by definition since there is no visual input. Do the authors want to refer to cortical dynamics here? The authors should expand on this.

The reviewer is absolutely right, while there is a connection between visual spatial integration and spontaneous/optogenetically driven cortical regimes via the common substrate, the way the sentence was formulated was misleading. It should read, as reviewer suggests: '*Ultimately, cortical dynamics across these different driving regimes (spontaneous, visually driven, and optogenetically driven) rely on a shared neural substrate.*' It was updated in the manuscript (lines 45-47).

Additionally, the authors should expand why

"especially the functionally specific long-range lateral connectivity in layer 2/3"

is relevant for "visual spatial integration" (i.e. why it is most relevant). What about for example patchy connectivity in deeper layers, see e.g. Gabbott et al. 1987.

The reviewer is right. While we do find in our model that the various dynamic phenomena examined in the manuscript are particularly sensitive to the presence of the functional biases

in the long-range layer 2/3 connections, this is the finding of our study not a previously established fact and thus this part of the sentence was misleading. We have hence removed that part of the sentence from the introduction. Additionally we want to point out that we do not make any claim that this is the only element important, in the end of the manuscript we show that other model parameters also have an impact, but we rather highlight that the functional bias of long-range lateral connections is particularly impactful. This of course does not imply that other aspects of cortical circuits, especially those not considered in our study such as connectivity of the sub-granular layers (but also feedback) might also not be of importance. Note that we have acknowledged this limitation explicitly in the last discussion sentence already in the first manuscript version: '*The present model also does not consider layers 5 and 6, which, with their extensive lateral arborisation, likely also contribute to the spatial integration of information in V1.*' (lines 851-853).

Results

L127 - 130

The authors write:

"To determine if the model's spontaneous activity exhibits wave-like patterns we generated a synthetic LFP signal from the neurons' input current (27), which we then convolved with a 2D Gaussian ($SD=250\ \mu\text{m}$) to account for the spatial pooling of LFPs in electrode recordings.

Could the authors comment on the choice of SD for the Gaussian kernel? $250\ \mu\text{m}$ appears rather large. Additionally, in light of e.g. Linden et al., Neuron, 2011: Why choose a Gaussian kernel at all? Linden et al. show that for large enough distances the contribution of a single cell to the LFP goes like $d^{-\gamma}$ (see their Figure 2 and eq 8). Wouldn't this be a more appropriate model for the pooling undertaken by the authors?

We used $250\ \mu\text{m}$ for the gaussian SD because Davis et al., 2021 used the same. This seemed a reasonable value as in the literature the spatial reach of LFP is assumed to be from several hundreds of micrometers (Liu & Newsome 2006, Berens et al. 2008, Katzner et al. 2009) to few millimeters (Kreiman et al. 2006, Kajikawa et al. 2011). However, as you rightly pointed out, a modeling study done by Linden et al., 2011 showed that the spatial reach of LFPs (measured as the radius of the area which contribute to 95% of LFP amplitude) generated by neurons with uncorrelated inputs is about $200\ \mu\text{m}$, which would correspond roughly to a kernel twice smaller than we used, as in our case this spatial reach corresponds to two standards deviations i.e. $500\ \mu\text{m}$. A great strength of this study was to show that this spatial range is dependent on the level of correlations in the input of the neurons, explaining the discrepancies between values estimated in experimental studies.

Note that although correlations of spiking activity between neurons in our network is low as it is in a state of asynchronous spontaneous activity, activity of excitatory neurons in layer 2/3 still exhibit some correlation at the level of 0.013 as shown in Figure S1C. Interestingly in Łęski et al. 2013 the same authors include a figure (Figure 1) summarizing the extent of the spatial reach of LFPs in relation to the signal frequency and the level of input correlation. For

frequencies of 30-60 Hz and input correlations around 0.01, it seems that this spatial range matches well the spatial range of 500 μm induced by our gaussian kernel. However it is true that the results we presented in Figure 2 are sensitive to the SD of the gaussian kernel. As expected and shown below, a half smaller kernel yields lower wavelengths and propagation speeds, and higher absolute magnitude of stLFP residuals. We now present these findings in the newly added Figure S18

Figure S18: Dependence of the LFP signal properties on the standard deviation used in the gaussian convolution. (A) The average wavelengths for model layer 2/3 for standard deviations of 500 μm (brown red), 250 μm (red) and 125 μm (pale red), and for the experimental data of Davis et al. for marmoset MT (blue) (Davis et al. 2021), as computed with the Generalized phase analysis. (B) Same as (A) for the average propagation speeds. (C) The average of the spike-triggered LFP residuals for model layer 2/3 for standard deviations of 500 μm (brown red), 250 μm (red) and 125 μm (pale red) across reference neurons and recording sites, binned according to their orientation preference differences. Error bars: 95% confidence interval.

Again, the choice of the gaussian kernel was principally motivated to make the analysis directly comparable to Davis et al. 2021. The advantage of a gaussian kernel is that it is easy to compute from its standard deviation the size of the area corresponding to 95% of the signal amplitude, which is generally the measure used to determine the spatial reach of LFPs. Moreover, it has only one parameter, compared to two in the case of the kernel in Linden et al. (the power and the cutoff distance). In our case both parameters would need to be defined somewhat arbitrarily, not necessarily resulting in a better estimation to the spatial reach of the LFPs.

We therefore added the following paragraph in Methods:

To simulate the spatial reach of the LFP, we convolved the LFP signal with a 2D Gaussian with a standard deviation of 250 μm based on the previous modeling study of Davis et al. (2021). A modeling study from Linden et al. has estimated that without any input correlations the spatial reach of the LFPs is around 200 μm , corresponding to roughly half the gaussian kernel size used in this study, but found that this value increases with the level of input correlations (Lindén et al. 2011). For the low level of pairwise correlations in our model of 0.013 (Figure S1C), Lindén et al. find that the spatial reach of LFP starts increasing by a few hundreds of micrometers. Therefore the choice of 250 μm as the standard deviation of the gaussian kernel is a plausible estimate in our context.

Ultimately, the current understanding allows only for rough choice of the kernel width. We therefore verified the impact of the kernel width on the key metrics of the spontaneous waves in V1 for which the synthetic LFP is used, that we present in Figure S18. We find that halving the kernel width moderately but significantly reduces the estimated wavelength and speed of the spontaneous waves, while the magnitude of the iso-orientation propagation bias increases. Doubling the kernel width has the opposite effect on all metrics.

L130 - 131

The author write:

"As shown in Figure 2A, the synthetic LFP signal propagates across the model cortical surface as a STW (spontaneous traveling wave, reviewer)"

To what extent can the STW already be observed in the spiking activity? Would the wavefront be "sharper" (in the sense or more clearly discernible)? Can one make a prediction of how the STW on the spiking level would look like if experimenters could record from neurons with a higher density?

This is indeed an interesting question. A recent study from Orsher et al. 2024 investigated how spiking activity looked like during travelling waves in turtle cortex. They find that spiking activity is better described by a sequential discrete activation of separate cortical populations rather than as a clear traveling wave. It's true that the model is useful to study this phenomenon given that we're able to record spikes at higher density than experiments.

We make similar observations as Orsher et al., with spiking activity propagating sequentially within segregated clusters. Spikes at one cortical region appear with a small delay of a few milliseconds after the appearance of LFP activity in the same region, consistent with the fact that our LFP signal is generated from input currents. Below is an illustration of the spatial propagation of spiking activity in the model, binned temporarily at 1ms and spatially within $50 \mu\text{m} \times 50 \mu\text{m}$, without any spatial smoothing with a gaussian kernel. The time at the top of each frame is relative to the 0 ms time point shown in Figure 2A. Note that in both cases, there is a clear propagation of activity upwards, but the LFP propagation in Figure 2A is more continuous. The scale is logarithmic for better visibility of cortical sites with moderate spiking activity.

While we find this a rather interesting preliminary finding and exciting match to Orsher et al. 2024 results, and thank the reviewer for pointing us in this direction, we cannot find a way how to fit this new result into the current manuscript as it is already 28% above the recommended length limits of Nature Communications, and this topic would deserve at least a full dedicated figure to be rigorously presented. We thus propose to leave these findings for future studies, where it can be examined in greater depth, that we certainly intend to continue to undertake with our model.

Figure 1:

It is confusing that the authors first refer to Figure 1 panel B and only after that to Figure 1 panel A while it seems that the panels also could be swapped without breaking the design of the figure.

The reviewer is right, we swapped the order of the two panels.

Figure 2:

They authors write:

"(B) Synthetic LFP signal (black) and corresponding generalised phase analysis, with the colour indicating the instantaneous phase."

It is not clear what the authors plot here. To my understanding they compute a spatially resolved LFP signal (shown in Figure 2 panel A). How do they get to a single time series in panel B? Is this the mean LFP? Is this for one "channel"? Also: To my understanding the authors overly the LFP signal with the instantaneous amplitude and instantaneous phase (color-coded). Is this correct?

The authors should make this more clear.

The reviewer is indeed correct, this data is from one recording site. The colored line refers to the output of the generalized phase analysis, color coded with the instantaneous phase. We have therefore modified the caption of Figure 2F (previously Figure 2B) to make it more clear:

Synthetic LFP signal (black) and the corresponding output of the generalised phase analysis (color-coded with the instantaneous phase), for a single recording site.

Panel C, D, G: The label of the vertical axes reads "Normalized density". The normalized is redundant - a density is by definition normalized.

We understand that this seems redundant to the reviewer, however, we would like to keep it this way as it is similar to the labels used by Davis et al. 2021 in Figure 6. We believe that it would facilitate the comparison across the two articles.

Panel F: The label of the vertical axis reads "Cumulative distributor". Probably "Cumulative distribution" is meant, correct?

We thank the reviewer for pointing that out. The text of the label was a bit too long so the end of it got cropped. We are sorry for the oversight, it has now been corrected.

Panel I, J: I could not find references to these panels in the text. Please elaborate on the interpretation of the results.

We are sorry for the oversight, we have now added the following sentences explaining the panels (lines 151-154)

As in the experimental data, we observe that the delay of the peak of stLFP increases with distance between recording sites (Figure 2B), and that its amplitude decreases exponentially based on distance (Figure 2C), confirming the validity of the stLFP method implementation in our model.

Note that Figure 2B and 2C in the revised version of the manuscript refer to the panels which were labelled as Figure 2I and 2J in the previous version.

L176 - L178:

The authors write:

"Crucially, they also observed significantly higher amplitudes of stLFP residuals at recording sites with orientation preference [...]."

It would be good to briefly explain what they mean by "residuals". I think it is necessary to make this clear not only in the methods but also here!

Thank you for spotting this omission. A clear description of this metric in the main text of the article was indeed missing. We have now rewritten our description of the results of Nauhaus et al. so that the stLFP residual is more clearly explained (lines 120-161):

We next wished to study the relationship between spontaneous wave propagation and the intrinsic functional maps identified by Nauhaus et al. in cats and macaques (Nauhaus et al. 2009). They used spike-triggered LFP (stLFP), which enables investigating how spiking activity at one reference recording location affects the LFP activity across all other recording sites.

[...]

For each reference electrode, Nauhaus et al. used an exponential function to fit the relationship between the stLFP amplitude and the distance to the reference electrode, and then computed the stLFP residuals by subtracting the value predicted by the exponential fit from the stLFP amplitude at each electrode.

L197 - L200:

The authors write:

"Interestingly, we also find that propagation speeds and wavelenghts in model layer 4 are lower then in layer 2/3 (Figure 2EH), which we attribute to shorter-range connectivity in layer 4 which requires activity to make multiple synaptic jumps to propagate to longer distances"

An alternative explanation could be that the additional synaptic jump comes the time it takes for activity to propagate from L2/3 to L4. Can the authors exclude this? Why do the authors favor their explanation?

Thank you for pointing out this second option. To address this concern we have run an ablation experiment to assess the influence of the feedback connections. As shown below, removing the feedback connectivity in the model does not significantly modify the average wavelenghts and only marginally increases the propagation speeds in layer 4 confirming our original hypothesis:

We have added this figure as Figure S15DE. We refer to this figure in lines 969-970 in methods when arguing that the presence of feedback connectivity has little impact on the phenomena we present in this study:

Removing this feedback projection doesn't significantly impact the results presented in this study (Figure S15).

It is worth pointing out that the impact of polysynaptic transmission due to shorter lateral connections in layer 4 on the propagation speed has been hypothesized earlier by Sato et al.. We have now added a reference to Sato et al. to support our claim at lines 276-280:

We attribute this to shorter-range connectivity in layer 4 which requires activity to make multiple synaptic jumps to propagate across the same distance. Polysynaptic transmission of activity has previously been hypothesized to slow down traveling waves propagation (Sato et al. 2012).

Figure 3:

Panel F: The color bar for the similarity attains values between -1 and 1. It is impossible to tell from the chosen color bar whether negative values are attained (my hunch would be yes). To my understanding the definition of the similarity in the appendix implies that the quantity is always non-negative.

Please resolve this.

We agree with the reviewer that the ranges of various similarities between panels E and F are not matching. However, the ranges are both correct, as the similarity of a correlation map to the orientation map and the similarity of two distinct correlation maps are two quite different metrics (see Methods 2.6 2-4. paragraphs vs. last paragraph), with respective ranges of (0,1) and (-1,1). Note that these metrics were developed by Smith et al. and Mulholland et al., not us.

We acknowledge that using the same color scheme for two different metrics was misleading to the reader. We chose them to ease comparison with experimental data, where this exact combination of color schemes was used - Smith et al. 2018 (Supplementary Figures 4f,g 5e, 6d, 7e) , Mulholland et al. 2021 (Figure 5), Mulholland et al. 2024 (Supplementary Figure S10).

We have now changed the color scheme used for similarity to the orientation map, and updated the similarity map labels to more clearly reflect the displayed metric. Additionally, we rewrote the relevant Results section to now more clearly reflect the difference between the two metrics (lines 297-304):

The model also replicates the finding that excitatory CMs match inhibitory CMs (Figure 3D). When we quantify their similarity (see Methods) it is consistently high regardless of seed position (Figure 3E, mean similarity= 0.457 ± 0.002 , range (-1,1)) (20). Additionally, excitatory CMs exhibit high similarity to the orientation preference map, regardless of seed position (Figure 3F (left), mean similarity= 0.744 ± 0.001 , range (0,1)). Note that this similarity metric is different from the aforementioned CM to CM similarity due to the periodicity of orientation preference (10).

We considered additionally renaming one of the similarity measures to further decrease ambiguity but ultimately decided against it, as it would create too large of a disconnect from the cited studies

L227 - L231

The authors write:

"Finally, the model accurately matches several key metrics of spontaneous ferret V1 activity: [...] correlation wavelength [...] excitatory dimensionality [...] local correlation eccentricity"

Why do the authors only here refer to ferrets but refrain from doing so for the findings regarding the correlations maps (Smith et al.)? I think it would overall help if the authors would more consistently highlight from which animal model the findings they are replicating are coming from. Surely, I agree with the authors that many of these findings are shared between highly visual mammals. But this would highlight even better the generality of the derived model.

We thank the reviewer for their suggestion. We have revised the manuscript and made sure that every time a study is introduced, we mention the species, modifying the following lines: lines 288-289:

... we have replicated the analysis of Smith et al. (10), who recorded from ferret V1: ...
line 309:

Next, replicating the findings of Kenet et al. (30) in cat V1, ...

Also: What do the authors refer to when they write "correlation wavelength"? I could not find a definition in the manuscript. Same with "excitatory dimensionality" (probably referring to dimensionality of excitatory neurons, but it's still unclear how the dimensionality was assessed). . Same with "correlation eccentricity".

The definition of these metrics was indeed missing, we thank the reviewer for pointing this out. We added the definitions for the missing metrics (dimensionality of spontaneous activity, correlation wavelength, local correlation eccentricity) to Methods 2.8 (lines 1250-1278).

Figure 4:

Panel D: Similarity between -1 and 1.

See our response to the comments on Figure 3.

Figure 5:

The authors write:

"(C) The mean Distribution of Activity across the Orientation Domain (DAOD) over the stimulation period and trials is more sharply oriented for both endogenous (orange) and surrogate (purple) stimuli, than the distribution of orientation preference (OD) in the stimulated area. [...]."

While I agree that this can be concluded for the graph showing the "Endog. DAOD" I do not see how the claim follows from the graph for the "Surr. DAOD." Could the author please expand on this.

The reviewer is right that the difference between OD and DAOD distributions is not clearly visible for the surrogate case. However, the sharpening is statistically significant ($p < 10^{-17}$), and visible, albeit weakly - the activation over the majority orientation (towards $\pi / 2$) is stronger than what would be expected by the OD distribution, and the activation over the minority orientation (towards $-\pi / 2$) is weaker.

We chose to present this pair of endogenous-surrogate stimuli, as they illustrated most clearly the sharpening in the endogenous case. We have now swapped it out for another pair where endogenous sharpening is statistically slightly less strong, but where the sharpening is clearly visible for both endogenous and surrogate stimuli. We have additionally added Supplementary Figure S10 containing all DAOD histograms:

Figure S10: Distribution of Activity across the Orientation Domain (DAOD) compared to the distribution of stimulated orientations, all stimulation patterns. (A) Individual endogenous stimulation patterns (B) For all patterns, the Distribution of Activity across the Orientation Domain (DAOD) under the stimulated area is more sharply oriented than the

distribution of orientation preferences (DOP) for the same area, suggesting the selective amplification of the dominant orientation by orientation-biased connectivity. (C,D) Same as (A,B) for surrogate stimulation patterns (E) For all patterns, Shannon entropy values of DAODs are lower than corresponding (DOP) entropies, signifying sharper histograms.

L361 - L363

The authors write:

"To test the existence of orientation tuning sharpening, we compared the entropies of the distribution of orientations under the stimulated regions and DAODs with lower entropies indicating broader tuning in the orientation domain."

Could the authors please expand on why a lower entropy indicates broader tuning? To my understanding, the authors are considering distributions on the domain of orientations $[-\pi/2, \pi/2]$. On this domain, the distribution with the highest entropy is the uniform distribution. I would argue, that the uniform distribution is the least broadly tuned, and that lower entropies are indicative of (but not proving) a sharper tuning.

From the displayed data in Fig 5 Panel C I would conclude that the endogenous stimulation has a lower entropy (less uniform than the surrogate stimulation), which is then also shown in panel D vertical axis.

Could the authors please comment on this issue?

We thank the reviewer for notifying us of this mistake, it was a typo - we meant to write sharper tuning there. Endogenous stimulation is indeed more sharply tuned (less uniform) than surrogate stimulation. We have additionally rewritten the entire section introducing DAODs, so that their purpose and implications are clearer (upon a request from another reviewer).

Figure 6

Panel E Vertical axis tick labels are both 3.4. Please fix.

We added an additional decimal place to the axes, after which the ranges are now shown correctly.

L389 - L392

The authors write:

"To understand these network phenomena more mechanistically, we performed simulations focusing on how lateral centre-surround interactions emanating from a single column shape the response to optogenetic stimulation."

Subsequently, the authors stimulate a circular region in their model. Do the authors observe a traveling wave after that stimulation, akin to the traveling waves after thalamic stimulation observed in Senk et al. Cerebral Cortex, 2024?

Thank you for notifying us about this publication, upon inspection we realized we should have referenced it. We have now added a reference to it in the Introduction.

The traveling wave created by Senk et al. was in response to the coherent activation of all thalamic neurons inside a circle with a radius of 400 μm . To best match this experiment, we stimulated Layer 2/3 of the model cortex with a short optogenetic pulse lasting 50ms. You can see the time course of the initial activation below. Although the effect observed is not a dense wave as Senk et al. have observed, it seems to be a somewhat concentric sparse travelling wave, mostly spreading through iso-oriented cortical columns (see time indices $t=60$, 65 ms).

As earlier, we find this line of investigation very exciting and thank the reviewer for pointing us towards it, but we believe a considerably deeper simulation and analysis would have to be done for this to be rigorous enough to publish, adding at least another altogether new section to the manuscript. Given that the current manuscript is already very long (28% above the recommended length) and explores many different phenomena, and that these experiments are not necessary to support the main claims of our study, we argue this experimental line is beyond the scope of the present manuscript. We hope to return to these questions in our future studies.

L418 - L419

The authors write:

"In line with iso-orientation biased long-range lateral connectivity, we found that neurons in the surround of directly simulated cortex are biased towards [...]"

"biased" -> "biased"

We aim to use British English throughout the article, which spells biased with a double s. Do the authors mean here "simulated" or do they mean "stimulated"?

We meant stimulated here; we simulate the entire 5x5 mm² cortical model in every experiment; here we optogenetically stimulate a circular area in the centre.

Discussion

General

Pre-computed Orientation Map

Does the pre-computed orientation map satisfy the pinwheel density derived by Kaschube et al. 2010?

In order to give an answer to the reviewer, we have computed both the hypercolumns and pinwheels density from our orientation map. We found 1.2 hypercolumns per mm² and 3.4 pinwheels per mm² (see figure below for an illustration of the pinwheels detection), yielding a density of 2.83 pinwheels per hypercolumn. It is worth pointing out that given the small section of V1 map included in our model, and performing the analysis on essentially a single sample, this estimate has to be considered with some error margin. Kaschube et al. found mean pinwheels density per hypercolumn of 3.12 in tree shrew, 3.15 in galago and 3.18 in ferret (Kaschube et al. 2010). Given that Kaschube et al. found densities lower than 2.8 in a substantial number of animals (see their Figure S4), the pinwheel density of our orientation map lies within the range outlined by experimental data.

Downscaling of model

Even though in comparison to the previous version of the model the density of neurons has increased, the model is not full scale (the authors write that they increase from a certain density of neurons to a higher density of neurons "to reduce downsampling of the number of neurons and synapses" implying that the network is still downsampled).

This is a potential problem since a substantial number of findings in this paper are based on correlations which can be strongly effected in models that are downsampled (van Albada et al. Plos CB 2017). Yet, the downscaling is only insufficiently discussed (in the main text only in

relation to low wavelengths). The authors should not en passant mention this but rather discuss this fact and the potential limitations that follow from the downscaling.

We couldn't find the 2017 article and instead assume that the reviewer is referring to the well known article (van Albada et al. Plos CB 2015). In any case, the reviewer is correct that although we increased the cortical density in respect to Antolik et al. (2024), the model is still downscaled compared to reality. The level of downscaling of neuronal density in the present V1 model is about one fifth (5 000 cells/mm² VS 26 500 cells/mm² reported by Beaulieu & Colomnier in area 17 in cat). We originally aimed at having a paragraph describing how the neuronal density affects the results of the model, but we deleted it to meet the word limit of Nature Communications. We have now been informed this is rather a recommendation, not a strict limit. The neuronal density in the model can indeed severely affect some of the model properties such as the distribution of wavelengths and propagation speeds, the level of similarity between the correlation maps of activity and the orientation map and the spatial scale of correlations:

Figure S17: Effect of cortical density on the model results. (A) The distribution of firing rates for six parametrizations varying in cortical density (see Methods). Red font in legends indicates the parameter values used in the base model. (B) Same as (A), for the cumulative distribution of wavelengths. (C) Same as (A), for the distribution of propagation speeds. (D) stLFP residuals difference between recording sites with orientation preference congruent to the reference electrode and sites with orthogonal preference for six parametrizations varying in cortical density. (E) Similarity between the excitatory CMs and the orientation map (blue), and between the excitatory and inhibitory CMs (orange) for six parametrizations varying in cortical density. (F) Similarity between the excitatory CMs and the uniform full-field optogenetically evoked excitatory CMs (blue), and between the full-field optogenetically evoked excitatory CMs and the orientation map (orange) for four parametrizations varying in cortical density. We couldn't run these experimental protocols on high density networks due to the large computational cost. The red circles in (D-F) indicate the parameter values used in the base model.

However, as shown in the newly added Figure S17, all of these metrics saturate at about 5000 cells/mm² density, which is what we use in this manuscript. Unfortunately, we cannot run optogenetic simulations with density larger than 5000 cells/mm² as their simulation time is extremely dependent on the number of neurons, making such high density simulations intractable on hardware available to us. We therefore chose this value of 5000 cells/mm² as the best compromise in terms of quality of results and computational feasibility.

We agree with the reviewer that the dependence of the results on the level of density should be discussed in the manuscript and have thus added the above figure as Figure S17 as well as the following paragraph in Methods (lines 1055-1063):

This density roughly represents a one fifth downscaling relative to the neuronal density in cat V1 (Beaulieu & Colonnier 1989), which was necessary for our computational simulations to run in a tractable amount of time and memory. Previous studies have shown that both properties of waves (Davis et al. 2021) and correlation of activity (van Albada et al. 2015) are negatively affected by downscaling the neuronal density. We find that downscaling indeed affects the properties we investigated, but that this effect saturates for a downscaling by a factor 5 or less, showing that the density used in this study represents a good compromise between fidelity and computational efficiency (Figure S17).

We also discuss this in more detail in the caption of Figure S17:

Previous studies have shown that both properties of waves (Davis et al. 2021) and correlation of activity (van Albada et al. 2015) are negatively affected by downscaling the neuronal density. The model instance we present here contains 5000 cells per square millimeter which corresponds to a downscaling of roughly one fifth in respect to the real neuronal density in V1 (Beaulieu & Colonnier 1989). This downscaling is necessary for running our computational simulations within a practical amount of time and memory. Several of the metrics we report here are substantially affected by the model neuronal density. When decreasing the density of the model, both distribution of wavelengths and propagation speeds are increased. Moreover, the similarity between spontaneous excitatory correlation maps and the orientation map also decrease as the density of the model is reduced. However, the effect of density considerably saturates once reaching the value of 5000 cells per square millimeter used in this paper. These results therefore indicate that a model needs to have a high enough density to give a quantitatively accurate account of the phenomena that are the focus of this study, but that a downsampling by a factor five or less doesn't significantly affect the results and represents a good compromise between fidelity and computational efficiency.

We also modified methods 2.14 and table 2 to account for the parameterizations used for the simulations shown in the newly added Figure S17.

However, these findings refute our hypothesis that the over-representation of low-wavelengths in the LFP signal could be caused by this downscaling. We have therefore removed this from the sentence in lines 240-242 which now reads as:

However, we observe a greater proportion of low wavelengths, likely due to the limited simulated cortical area and the absence of extra-areal input which increases global network synchronisation.

Spike-triggered LFP

In the methods the authors describe how they calculate the stLFP residual. According to their description, first an expected stLFP amplitude is determined via an exponential fit, then the actual stLFP is subtracted. The authors show in Figure 2K that if the biased connectivity is removed the residuals are small for L2/3 neurons (as is the case in L4). It would be helpful to see how expected stLFPs look like to convince the reader that an exponential fit is indeed a good model.

The sentence in the methods describing the exponential fit referred incorrectly to Figure 2E whereas it was meant to refer to Figure 2C (previously Figure 2J). We have now modified the line 1172 accordingly. Figure 2C shows how, for a single reference recording site, the amplitude of the peak of the stLFP is linked to the spatial distance between recording sites. It is clear there that the exponential (blue line) fits the data point distribution well. For this analysis, we have only considered data from reference neurons for which the exponential fit had an explained variance larger than 0.5. Nauhaus et al. 2009 also used exponential fits when performing the same analysis.

Complex nonlinear interaction across the cortical space underlies population responses to optogenetic stimulation in V1 The authors show that when increasing an optogenetic stimulus that especially the response of excitatory neurons show a highly non-linear behavior: first increasing the firing rate, then decreasing (Fig6D). I was wondering whether this can be related to extra-classical receptive field effects like end-inhibition (Bolz and Gilbert, Nature 1986). Famously, predictive coding was suggested as a means of explaining these effects in an hierarchical network (Rao and Ballard, Nature Neuroscience 1999). However, to my knowledge it is still not resolved whether feedback is necessary to explain end-stopping in the brain. It would be interesting to see whether your network can explain this effect without additional areas just via the local recurrent connectivity.

In this study we measure the responses of the neurons in the surround of the disk of light that stimulates the cortex, not in its center - thus the analysis of the external optogenetic stimulation we present does not directly correspond to the standard visual size tuning stimulation protocols, where grating disk is gradually increased but the cells measured have RFs centered on the gratings stimulus. That said, the answer to your question is yes: it has been shown that size tuning can be explained by only lateral connections (e.g. Somers et al. 1998), although feedback can also explain size tuning (e.g. Ichida et al 2007). However, to the best of our knowledge, a direct experimental evidence decisively deciding if contextual modulation in V1 is purely due to intra-areal lateral interactions, or purely due to feedback, or combination of both is currently missing.

Minor

In the equations many of the labels like "dist" and "func" are typeset as variables using an italics font. This is not correct, they should be typeset in an upright font using a command like $\mathrm{\{}}$ or similar to distinguish labels from variables or indices.

We thank the reviewer for pointing that out. We have now corrected that in all equations.

Reviewer #2 (Remarks on code availability):

I have not reviewed the model code but the fact that the authors use the established simulation codes Mozaik and NEST in recent versions substantially increases the probability that the results can be reproduced. This is how computational studies should be done today. In terms of reproducibility the work is well ahead of a typical study in computational neuroscience today.

Thank you for appreciating our efforts on the reproducibility front!

Reviewer #3

Reviewer #3 (Remarks to the Author):

This manuscript by Dr. Antolik et al. presents a neural network model of V1 layer 4 receiving LGN input, and its primary downstream target, layer 2/3. Based on anatomical connectivity studies, the model connects neurons in a biologically realistic manner, incorporating distance-dependent strength and synaptic delay, patchiness given by an orientation map, a push-pull scheme in layer 4, and iso-orientation biased long-range cortical connectivity in layer 2/3. The model successfully captures numerous aspects of experimental observations from the early visual system. Various parameters have been explored, and overall, the study is carefully and comprehensively implemented, effectively focusing on the role of iso-orientation biased long-range cortical connectivity in V1 layer 2/3 across a series of experimental results. The study provides substantial computational insights and intuition, aiding understanding.

Major comments:

1. As referenced in the manuscript, Z. Davis (2024) presented a model with a highly similar recurrent structure and demonstrated functionally biased spontaneous activity. The authors claim to extend this by linking it to visually or optogenetically driven processing. However, for the results investigating how the functional specificity of cortical connections dictates the correlation structure of spontaneous activity, it is unclear how the biased long-range horizontal connections give rise to the similarity between CMs and the orientation map differently compared to the previous model (Fitzpatrick, M. Kaschube et al, 2018) with heterogeneous Mexican hat connectivity giving rise to modular activity pattern. Or say, how are these two models essentially different?

First we just want to make sure that it is clear that our model is not an extension of Z. Davis but a rather slight modification of our previous model of V1 (Antolik et al. 2021, Taylor et al. 2021, Antolik et al. 2024) as stated in the manuscript, although both models have a number of similarities. The key commonality between our model and Davis et al. is that both models have distance dependent cortical connectivity with functionally biased long-range lateral

projections. Beyond that, our model simulates layer 4 and layer 2/3, with both having different functional rules of connectivity, and takes into consideration both retino-thalamo-cortical processing and synaptic depression, whereas Davis et al. does not contain any of these features. Moreover, although both models demonstrate functionally biased spontaneous activity, functional bias in the propagation of traveling waves was not investigated by Davis et al.

The model by Fitzpatrick, M. Kaschube et al, 2018, does not have a notion of orientation maps, as it does not model feed-forward processing in any way. Note that because of this, authors do not reproduce in this model the similarity between orientation maps and spontaneous activity. They only show that the model has long-range correlations in the simulated spontaneous activity. In terms of recurrent cortical connectivity, the main difference is that the Fitzpatrick model does not assume long-range excitatory lateral connectivity that we do include in layer 2/3, and does not assume functional bias of connectivity, because they model young animals before eye-opening. The Fitzpatrick model also assumes longer inhibitory connections than excitatory connections which is contrary to experimental evidence in adult animals, while we set up lateral connectivity in the model based on anatomical studies in adult animals (see Methods). We have now extended the discussion to more thoroughly explain the relationship of our model to previous modeling work (lines 793-833). Also in the summary to all reviewers we explain in greater detail and justify the relationship of our model to the studies in young animals by Kashube & Fitzpatrick

2. As a modeling study, the strength of the work lies in the solid analysis proving the ability of the model to capture a number of important experimental results in the field. The weakness is the testable predictions generated by the model. I am skeptical about the significance of the predictions listed in the manuscript. The authors predict slower wave propagation lacking orientation bias in the layer 4. How this could influence behavior given the established finding of the characteristics of traveling wave in its downstream target layer 2/3 remains to be addressed. Also, the prediction of the non-monotonicity of optogenetically driven cortical response to increasingly stimulated area is not surprising since it can be explained by the extensively studied inhibition-stabilized network (ISN) regime, as the authors mentioned in the discussion. This also pertains to my first major comment.

We agree that the nature of wave propagation in layer 4 is unlikely to predictively contribute to the final behavior beyond what we already know about the propagation in a downstream processing stage - layer 2/3. Nonetheless this still constitutes a prediction through which the current model can be falsified. Furthermore, it is important to explain how the computations leading to visual perception and eventually behavior are implemented in neural substrates at all stages of processing, and hence dynamics of layer 4 circuits are still important. If we accept the notion that to explain behavior one just needs to make a strong link between as late stage of processing as possible and the behavior, we might as well explain behavior only as a function of the final output motor signals the brain generates.

Indeed, in-general, the non-monotonicity of optogenetically driven stimulation can be explained by the ISN regime, as we already prominently acknowledged in the reviewed manuscript version (lines 761-765). However, to the best of our knowledge, the ISN regime was not studied when embedded in highly heterogeneous and biologically detailed circuits

as those examined in this study. The dynamical consequences of the ISN regime when embedded in such highly heterogeneous cortical circuits, and depending on the functional congruence of the stimulated vs recorded sites, is far from trivial or well understood. Thus studying the dynamics of biologically detailed models and generating predictions is of high importance to generate insights on the dynamics and computations performed in V1, and the current ISN body of work hasn't addressed this fully.

3. Since the paper covers a broad range of experimental results (traveling wave, distal correlated activities, optogenetically driven cortical responses), a brief introduction on each metric being investigated in the result section may provide useful intuition and help communicating the concept for readers with diverse research interests. For example, fig. 3H is abruptly presented with little introduction. The same comment applies to the concept of stLFP residual, entropy of DAOD distributions.

We acknowledge that this was lacking in the manuscript. It now includes more extensive descriptions of these metrics:

Fig 3H: We now clarified that these metrics quantify the spatial properties of spontaneous activity. We found that a detailed description of each metric would make the section too long and tedious to read which is why we refer reader to Methods for the detailed description. The revised text reads (lines 315-328):

Finally, we quantitatively assess how well the model captures the spatial properties of in-vivo spontaneous activity using several key metrics of spontaneous V1 activity (10, 20) (Figure 3H), overall finding a good match with young ferret data (see Methods for a detailed description of all metrics). The model reproduces the degree of similarity of excitatory and inhibitory CMs, long-range correlations and correlation wavelength for both excitatory and inhibitory activity, while the inhibitory dimensionality and local correlation eccentricity lie only slightly out of experimental data ranges. Our model does exhibit somewhat higher similarity between spontaneous activity and orientation preference maps than observed in young ferrets, which could be due to the absence of measurement noise in the model. Alternatively, this discrepancy could be explained by the young age of the imaged ferrets, whereby the orientation representation becomes more refined as the animal matures (see figure 5C in (10)), reflecting the stronger alignment between orientation maps and spontaneous activity we observe in the model of adult V1.

stLFP residual (lines 120-161):

We next wished to study the relationship between spontaneous wave propagation and the intrinsic functional maps identified by Nauhaus et al. in cats and macaques (Nauhaus et al. 2009). They used spike-triggered LFP (stLFP), which enables investigating how spiking activity at one reference recording location affects the LFP activity across all other recording sites.

[...]

For each reference electrode, Nauhaus et al. used an exponential function to fit the relationship between the stLFP amplitude and the distance to the reference electrode, and then computed the stLFP residuals by subtracting the value predicted by the exponential fit from the stLFP amplitude at each electrode.

DAOD and DAOD entropy (lines 564-580):

We have shown that model responses to endogenous (OP map correlated) optogenetic stimuli are stronger than to equivalent surrogate stimuli. We hypothesise this is due to orientation-biased recurrent connectivity selectively boosting the activity of the dominantly active orientation, resulting in a sharper orientation tuning profile than would be expected from the underlying orientation preference map. To quantify how the population activity is distributed across the orientation domain during patterned stimulation, we introduced the measure of Distribution of Activity across the Orientation Domain (DAOD). For each frame of activity, we binned cortical positions by their orientation preference, summed the activity within each orientation bin, and normalised the resulting distribution to a unit-integral density function. For each of the 8 endogenous and 8 surrogate stimulation patterns, we calculated DAOD within the optogenetically stimulated regions and averaged it across all frames of the response and 5 trials (see Figure 5D for example patterns, Supplementary Figure S10 for all patterns).

To quantify the sharpening of DAODs over the distribution of orientation preferences in stimulated regions, we compared the Shannon entropies of these two distributions (Figure 5E), with lower entropies indicating sharper tuning in the orientation domain.

Minor comments:

Figure 1A inset illustrating the lateral connectivity properties: incorrect x-axis label in the right panel.

The reviewer is correct, thank you for spotting it. We modified the axis label to more clearly convey the nature of functional connectivity bias.

Figure 2K: x-axis label should be '.... in orientation preference'.

We modified the label accordingly.

Figure 6B, right: incorrect x axis label.

We corrected the label to orientation preference.

as the model is claimed to be applicable to primate V1, a discussion of eccentricity-dependent cortical organization should be included.

The present cortical model corresponds to 5x5 mm of V1 at para-foveal eccentricity. Since within such relatively small cortical space the eccentricity dependent changes to RFs parameters are minimal we chose to ignore them for the sake of simplicity. We have added into the Materials & Methods section discussing the LGN model following text: '*Due to the relatively small region of visual space our model covers, we do not model the systematic changes in RF parameters with foveal eccentricity and thus assume that all ON and OFF LGN neurons have identical parameters.*'

Let us just add that this was clearly explained in the previous model publications this study builds upon and we have omitted this justification in the present paper for the sake of conciseness, but acknowledge this was a mistake.

the model includes feedback pathway from the layer 4 to the layer 2/3. This is a bit confusing since the output superficial layers don't project significantly to the input layers to my knowledge. Is this necessary to any of the key results?

The reviewer is indeed correct in saying in V1 there is no significant input from superficial layers to layer 4. However, supragranular layers project to infragranular layers which in turn project to layer 4. We found it important to respect this feedback loop in the model, but because it does not explicitly model the infragranular layers we have added this 'artificial' projection. We have justified this addition in the previous publication of the model (Antolik et al. 2024) but omitted it from the present manuscript for conciseness, which we agree wasn't ideal. We have now more clearly communicated this in Material & Methods:

Although there is little evidence of such direct connections from supragranular layers to layer 4, an important feedback projection from supragranular layers reaches layer 4 via infragranular layers. Because we believe that it is important to close this cortico-cortical loop and as infragranular layers are absent in the model, we have therefore decided to create this direct projection from layer 2/3 to layer 4 in the model.

We have now verified how does the inclusion of the feedback impact the results in this study, finding that no key conclusion is affected by the presence/absence of this feedback projection:

Figure S15: Effect of the absence of the feedback connectivity on model results. (A) The average wavelengths in layer 2/3 for the original instance of the model (red, full) and a version where the feedback connectivity is removed (red, hollow), and for the experimental data of Davis et al. for marmoset MT (blue) (Davis et al. 2021), as computed with the Generalized phase analysis. (B) Same as (A) for the average propagation speeds. (C) The average of the spike-triggered LFP residuals in layer 2/3 for the original instance of the model (red, solid) and a version where the feedback connectivity is removed (red, dashed) across reference neurons and recording sites, binned according to their orientation preference differences. Error bars: 95% confidence interval. (D) The average wavelengths in layer 4 for the original instance of the model (pink, full) and a version where the feedback connectivity is removed (pink, hollow), as computed with the Generalized phase analysis. (E) Same as (D) for the average propagation speeds. (F) The average of the spike-triggered LFP residuals in layer 4 for the original instance of the model (pink, solid) and a version where the feedback connectivity is removed (pink, dashed) across reference neurons and recording sites, binned according to their orientation preference differences. Error bars: 95% confidence interval. (G) Comparison of the layer 2/3 model results with (red, full) and without feedback (red, hollow) for a wide range of metrics also computed in the ferret cortex (gray, with mean represented in black) (Smith et al. 2018, Mulholland et al. 2021). Experimental data scraped from Smith et al. (2018) Supplementary Figure 5g, Mulholland et al. (2021) Figure 4c,e,g,j, 5f.

We have added this figure as Figure S15, and refer to it in Methods at lines 969-970:
Removing this feedback projection doesn't significantly impact the results presented in this study (Figure S15).

Reviewer #4

Reviewer #4 (Remarks to the Author):

In this work Rózsa et al. use a spiking recurrent network model to simulate activity patterns in the visual cortex under a variety of stimulus conditions. They show the model can describe data on spontaneous traveling waves and the spatial patterns that arise from spontaneous activity or optogenetic stimulation. This model is a combination of two previously models developed by these authors: 1) a multi-layer model of feedforward visual stimuli (Antolík et al., 2024), and 2) a brain-machine interface model to simulate optogenetic stimulation of cortical networks (Antolík et al., 2021).

Overall, this work is an extension of previous work studying spatially biased functional connectivity within recurrent networks. As the authors note in the introduction, biased topographic connectivity is required for orientation preference biased spontaneous traveling waves (Davis et al., 2021), spontaneous activity patterns with long-range correlations (Smith et al., 2018), and optogenetically evoked activity patterns (Mulholland et al., 2024). The main advance here is repeating this in a spiking network where temporal dynamics are better constrained by synaptic time constants, delays, and biophysical mechanisms. The authors use this model to make several testable predictions: 1) recurrent network amplification of input patterns that align with the iso-orientation-biased functional connectivity, 2) and the nonlinear effects of local recurrent connections on surround activity when provided a localized input.

We would like to emphasize that the contribution of this paper is not just 'repeating what previous models did in spiking networks' but also demonstrating that the same neural substrate can explain all these different phenomena across the visually driven, spontaneous and externally driven regimes. Our model is for example very different from that used in (Smith et al., 2018) and (Mulholland et al., 2024), which is necessary to satisfy the much greater set of biological phenomena that we seek to explain. We cannot emphasize enough that this is not trivial at all and cannot be concluded from any of the mentioned previous models. Furthermore several phenomena were actually modeled for the first time, which we now emphasize better in the paper (see lines 729-734)

The primary result here is that moderate-strength iso-oriented long-range connectivity improves model description of the data. My enthusiasm is limited by several factors. The level of bias is not systematically varied and at times the effects of adding bias can be weak (Fig. 4I).

The bias was systematically varied, as presented in Figures 7, S12 (previously S10) and S13 (previously S11) already in the first submitted paper version. Additionally, we now also investigate the effect of cortical density across six parametrizations, with parameters shown in Table 2 in methods and results presented in Figure S17. We do not explore all model parameters. In a biologically detailed model like this, it is simply impossible given the

complexity of the system under study and the combinatorial explosion. But we would argue that this is a considerably more thorough model parameter exploration than what is typically done in biologically realistic network models like the one we present here. We would also like to point out that the sizable yet far from exhaustive set of parameters we chose to explore was selected based on our extensive empirical experience that they are parameters with particularly strong impact on the cortical dynamics.

Regarding the effect being 'weak', it depends on the phenomenon but it would be misleading to mistake magnitude of phenomena for their importance - small phenomena can have major implications especially in complex non-linear systems, and even if not, can be important indicators of the characteristics of the system, i.e. can facilitate the understanding of the system nonetheless. Please note that a large proportion of phenomena in the experimental papers this paper is targeting are themselves small in magnitude (e.g. the stLFP residual differences between co- and ortho- oriented domains in Nauhaus et al, or the absolute magnitude of the spontaneous structured correlations in the Fitzpatrick/Smith series of papers), yet nobody is questioning their importance.

Specifically in Fig 4I the phenomenon - i.e. the difference between congruent vs incongruent stimulation with respect to the OR maps is small in absolute magnitude but statistically highly significant ($p_{\text{endo-surr}} = 2 \times 10^{-18}$, two sample T-test). Crucially, the removal of the orientation bias of the layer 2/3 lateral connectivity eliminates 59% of this relative difference (mean correlation $\mu_{\text{endo}}=0.800\pm0.004$, $\mu_{\text{surr}}=0.710\pm0.006$, $\mu_{\text{endo_no_bias}}=0.747\pm0.003$), i.e. in relative terms (which is what matters) the orientation bias of lateral connectivity has a major impact on the phenomena.

This connects to a larger point: the manuscript does not deeply explore the circuit mechanisms that allow the data to be described. This is true especially for the bias parameter but also holds for other parameters. There is some of this in Fig. 7, but the parameter optimization is relatively limited and in some cases (Fig. 7D) parameters have little influence. I would rather have seen this manuscript focus on the parameter regimes — the network mechanisms — that describe the data. Why is this model the model of greatest interest, and what are the essential mechanisms that make it so?

The main goal of this paper was to demonstrate that a single unifying substrate can explain a range of phenomena across three driving regimes (visually driven, spontaneous, and externally driven). We want to emphasize that the transition from multiple (simpler) models explaining one such regime and few such phenomena at a time to a single unifying model is highly non-trivial, requires a major body of work, and constitutes a major achievement on its own. We also stress that it is necessary to establish such a base unifying model if we ever hope to gain a holistic understanding of visual processing, and for example deeply link findings from optogenetic stimulation experiments to visually driven processing. In other words, the present manuscript is a necessary **prerequisite** to gaining the circuit-level understanding of visual processing the reviewer asks for, and stands as a major achievement on its own.

That said, we appreciate your interest in studying the circuit mechanisms and fully share your enthusiasm. We would like to point out that the last two result sections already go in

that direction, but also fully acknowledge that there is a lot more work that can be done. But all that work simply cannot be squeezed into a single manuscript, nor do we find it fair to expect such a range of achievements within a single study. The last paragraph of the discussion section fully acknowledges this, and has now been further expanded to be even more encompassing and specific (lines 834-870). We have several ongoing projects that will build upon the presented model, and go in the direction the reviewer is pointing to.

Regarding studying the influence of specifically the orientation bias of the lateral connectivity, as we already explained in previous answers, it is explored rather extensively and thoroughly in (Figures 7, S12 (previously S10) and S13 (previously S11)) for a model of this complexity, and the general parameter explorations have now been further extended with exploration of the impact of neural density. The motivation for the design of Figure 7 is to only highlight the most interesting findings of the parameter search, leaving the rest for supplementary figures to maintain the readability of the already very long and dense manuscript that now surpasses the NC length recommendations by 28%.

That said, I believe the paper in the present form will be a contribution to the field. I have some comments below but none are serious obstacles to their results.

Major:

I would have liked to see the authors take advantage of the multiple layers of their model to study the interaction between feedforward stimuli (visual input) and local recurrent network activity (spontaneous or optogenetically-evoked), which is not examined here.

We fully appreciate the reviewers' interest in this question. We agree this is an important and exciting question, and our model represents an ideal framework for its investigation. As such, we extended Figure 4 with three panels (J,K,L), showing an instance of specifically this type of analysis, which serves as additional validation of our optogenetic stimulation framework.

We simulated the joint optogenetic-visual experiments of Chernov et al. 2018, who recorded in adult macaque. They combined optogenetic stimulation of a cortical column with visual stimulation either iso-oriented or orthogonal to the orientation preference column, and observed stronger response compared to visual-only stimulation for the iso- condition, and weakened response in the ortho- condition. We observe an analogous iso-facilitation and ortho-suppression in our model (Figure 4L).

Unfortunately, we were limited to a qualitative comparison to the experiment results, because converting model spiking firing rates to the intrinsic signal involves multiple unknowns such as local vasculature, species-specific hemodynamics, dura thickness and optical scattering. Thus, we could not quantitatively interpret the difference in the dynamic range of the observed effect (model firing rate range of 14-17 sp/s vs. experiment relative intrinsic luminance 0-0.15 dR/R).

We would like to point out that this use case further highlights the significance of the model we are presenting here, as it opens avenues towards studying a number of key open

questions in visual processing that could not be addressed in previous models that focused on visual stimulation, external (e.g. optogenetic) stimulation or spontaneous activity.

We share the reviewer's enthusiasm for this question and are already undertaking another separate project investigating precisely such combined visual and optogenetic stimulation, particularly focusing on how this is influenced by the spatial and functional properties of the excitatory vs. inhibitory interactions. We do so together with experimental collaborators that collect analogous data. However, it is clear from our preliminary results that this question is complex enough to deserve a full dedicated study and not just a minor addition to this paper.

In Figure 3H the authors argue that the model results replicate in vivo ferret data, but for several of the metrics the model falls outside the in vivo ranges. Specifically, there are discrepancies in the similarity between excitatory correlation maps and orientation preference maps, inhibitory event dimensionality, and inhibitory eccentricity. How do you account for these differences, is there a mechanistic explanation? The data taken from Smith et al., 2018 and Mulholland et al., 2021 was measured in young, postnatal ferrets—could this be due to differences in the developing brain? Overall, the authors should modify their statement to highlight where the model differs from the in vivo data, and offer a discussion of the possible reasons why.

In this manuscript we present a generic model of columnar V1 and compare it to data from 11 experimental studies obtained under three different driving regimes. We have to rely on data recorded in 4 different species - cat, ferret, macaque, marmoset, as the investigated conditions were not all explored in any single species. Given this, our primary goal is to demonstrate the qualitative presence of all the key phenomena in the model, but some quantitative variations from the individual species data are hence inevitable. That said, we do wish to be transparent and present the readers the quantitative comparisons to show that we do not deviate from the data fundamentally. We would argue that overall, across all the many points of reference the model gives surprisingly good quantitative match (given the multi-species comparison) to the data nevertheless. We are now also more diligently emphasizing throughout the manuscript the specific species different data points come from. We have now more explicitly discussed this issue in the discussion (line 835-843):

With the increasing range of phenomena we wish to explain within one model we face the challenge that data from no animal preparation offer full coverage of phenomena of interest. Thus our model comparisons involve data from multiple species. While this approach encourages a focus on principles common across mammals with columnar V1, rather than species-specific details, it inevitably limits precise quantitative fits across all measures due to inter-species differences. Nonetheless, our quantitative comparisons reveal consistently good matches across diverse measures, with only occasional moderate deviations, confirming the model operates within a biologically plausible regime.

Specifically, regarding the comparisons the reviewer directly mentions, we deviate only rather slightly from the data on inhibitory event dimensionality and inhibitory eccentricity, but a bit more substantially on the similarity of excitatory correlation maps and orientation preference maps. We think the latter is an interesting question. One simple explanation could be the greater level of inherent system noise in the real biological V1 or more likely measurement noise that is present in real recordings but completely absent in our in-silico

twin. Such extra noise in biological data could be systematically driving the correlations down in real data explaining the discrepancy. But we do agree that the recordings in young animals as opposed to our adult model is another potential explanation. It is well described that the functional representations (i.e. OR maps) becomes more refined as the animal matures, and it can hence be also expected to be better aligned with the spontaneous activity with progressing age as is indicated directly in Figure 5C of Smith et al. 2018. This could explain the stronger average alignment between orientation maps and spontaneous activity we observe in the model of adult V1 vs the data obtained from the young animals. As suggested, we have now more explicitly highlighted the points of disagreement in the manuscript and offered the possible explanations:

Finally, we quantitatively assess how well the model captures the spatial properties of in-vivo spontaneous activity using several key metrics of spontaneous V1 activity (10, 20) (Figure 3H), overall finding a good match with young ferret data (see Methods for a detailed description of all metrics). The model reproduces the degree of similarity of excitatory and inhibitory CMs, long-range correlations and correlation wavelength for both excitatory and inhibitory activity, while the inhibitory dimensionality and local correlation eccentricity lie only slightly out of experimental data ranges. Our model does exhibit somewhat higher similarity between spontaneous activity and orientation preference maps than observed in young ferrets, which could be due to the absence of measurement noise in the model. Alternatively, this discrepancy could be explained by the young age of the imaged ferrets, whereby the orientation representation becomes more refined as the animal matures (see figure 5C in (10)), reflecting the stronger alignment between orientation maps and spontaneous activity we observe in the model of adult V1.

This paper would benefit from a more detailed explanation of the Distribution Across Orientation Domains (DAODs) and distribution of orientation preferences within the stimulated ROIs (OD) (Fig 5D), which was non-intuitive and initially difficult to evaluate how it relates to functional network connectivity. What specifically is the difference between these two measures and their entropies? I presume that DAODs are derived from the activity during modeled optogenetic stimulation and ODs are derived from the activity of modeled spontaneous activity within stimulated ROIs (based on the black labels in Figure 5E), but this was not clearly stated in the legend or text.

Furthermore, interpreting this was occasionally made more difficult by apparent contradictions in the text. For example, in lines 362-363, it states: “with lower entropies indicating broader tuning in the orientation domain”. However, Figure 5C suggests that endogenous patterns –which in Figure 5D and E have lower entropies—have sharper tuning. The fact that patterns with lower entropies correlates with higher firing rates is consistent with the finding that endogenous patterns are more strongly amplified than surrogate, which suggests to me that the statement in lines 362-363 is a typo.

Overall, a more explicit, simplified explanation of these terms in section will help readers evaluate and interpret the results in figures 5&6.

We agree with the reviewer that the section introducing DAOD was convoluted, as other reviewers pointed out also. We have now rewritten the relevant section to more clearly communicate what the measure is for, and how it compares to the distribution of orientation preferences in the stimulated area (pooled from the orientation preference map). We have

additionally changed the color of the spontaneous activity DAOD/firing rate points in Figure 5F (previously 5E) to indicate their lack of relationship with the orientation preference histograms of Figure 5D (previously 5C).

The reviewer is also correct about the contradiction in the text. It indeed was a typo - we meant to write sharper tuning there. We apologize for this oversight, and thank you for spotting this important mistake! It was now corrected.

How does the location of the local stimulus affect the results of Figure 6? If the stimulus was not centered on an iso-orientation domain, but instead overlapping over two domains or a pinwheel center, how would that affect the firing rates for excitatory and inhibitory cells? Would the difference in firing rates between the 'iso' and the 'ortho' surround disappear, or would there be a winner-take-all effect?

Thank you for the insightful question. When we don't stimulate in the center of a cortical column, but on an edge or pinwheel, we found it not straightforward to define what counts as a "close" or "far" orientation - the majority stimulated orientation might change from radius to radius. We will thus define the reference orientation with respect to which the other orientations are close or far as the mean orientation preference under the smallest radius, 50um.

In general, the Distribution of Activity across the Orientation Domain (DAOD) in the surround seems proportional to the activated orientations under the stimulation area, for both excitatory and inhibitory neurons.

In the case of stimulating a center of a cortical column, unsurprisingly co-oriented neurons are the most activated in the surround, and ortho-oriented ones the least activated.

In the case of stimulating an edge, at smaller radii the edge is not yet prevalent in the distribution of orientation preferences, and thus the majority orientation is activated (though noticeably less than when targeting a column). As the stimulation radius increases, orientation preferences under the stimulated area start to cluster into two separate local maxima. Interestingly, the response does not follow the increasingly bimodal distribution of orientation preferences within the stimulated area - we observe a single, broad peak of activity in the orientation domain, centered between the stimulated orientations.

As for the case of stimulating the pinwheel, both the histogram of stimulated orientations and DAODs are fairly uniform. Overall there seems to be a light bias towards two orientations that are slightly overrepresented in the local vicinity ($r < 300\mu\text{m}$) of the specific pinwheel we tested, even for radii where there seems to be no obvious bias towards these orientations. We hypothesize that this is the result of the stimulated center activating its local vicinity, where a soft winner-take-all mechanism decides the majority activated orientation in that local vicinity, which, in turn, differentially activates the surround.

While we genuinely find these questions interesting and their are worthwhile avenues for future investigation, and we wished to offer reviewer at least an initial insight, a full systematic analysis across different pinwheels and edges would require a large number of

simulations along with the extensive accompanying analyses, which in our opinion is firmly beyond the scope of this manuscript, which is already 28% over the recommended length. It is important to point out that, while such analysis would be an interesting addition, it is not essential for supporting any claims made in the manuscript, and which already presents a major body of work. We hope to pursue similar experiments in a future project, ideally in collaboration with experimental groups, to explore these effects in a broader context and with data available for validating the findings.

Minor comments:

In lines 137-139, you state “Prior to convolution, the wavelengths of our synthetic LFP signal are significantly higher than in the spatially shuffled signal, demonstrating that the signal is indeed spatially structured”. I assume you mean that the wavelengths are significantly smaller (higher frequency), as shown in Figure 2F. Also, here and in Figures 2E,F it was not immediately clear in the text or the legend whether you were measuring the spatial wavelength of the traveling wave or the temporal wavelength calculated from the Generalized Phase analysis.

No, we indeed meant 'higher' in this sentence. Figure 2J (previously figure 2F) shows that the distribution of wavelengths in the LFP signal prior to convolution with a spatial kernel (orange trace) contains larger values compared to the spatially shuffled signal (yellow trace).

We acknowledge that the caption of Figure 2I (previously Figure 2E) should state that these wavelengths were computed using the Generalized Phase analysis. We have now modified it to state it explicitly. We believe that the changes made in the main text now also express clearly that the wavelengths we are discussing refer to the ones computed with the Generalized Phase analysis:

We applied the Generalized Phase analysis to compute the instantaneous phases of the wideband filtered signal (Figure 2F), yielding the distribution of wavelengths and propagation speeds of the synthetic LFP (see Methods) which we compared to Davis et al. marmoset data. Prior to convolution, the wavelengths of our synthetic LFP signal are significantly higher than in the spatially shuffled signal, demonstrating that the signal is indeed spatially structured (Figure 2J).

Also the caption of Figure 2J (previously 2F) now reads as:

The average wavelengths for the model layers 4 (pink) and 2/3 (red) and for the experimental data of Davis et al. (Davis et al. 2021) (blue), as computed with the Generalized phase analysis.

You predict that Layer 4 would have slower spontaneous traveling wave propagation than Layer 2/3. Given that Davis et al., 2020 shows that the phase of spontaneous activity affects the sensitivity to visual stimuli, could you speculate what effect, if any, this slight temporal de-synchrony could have on visual perception?

Thank you for the interesting question. We believe that as conscious visual perception is likely a consequence of the activity in higher visual areas (and beyond), and as only output cortical layers (i.e. layer 2/3) project upwards the visual hierarchy, this difference in wave propagations across input and output layers might not have significant effect on visual perception itself. We however view this to be too speculative, and hence do not feel comfortable adding it to the paper directly. It is worth pointing out that even if such layer-dependent differences in propagation might not directly impact perception, this finding still remains a meaningful piece in the mosaic of understanding of how visual processing is implemented in the neural substrate.

In the legend of Figure 5C, it says "The mean Distribution of Activity across the Orientation Domain (DAOD) (see Methods) over the stimulation period and trials is more sharply oriented for both endogenous (orange) and surrogate (purple) stimuli, than the distribution of orientation preferences (OD) in the stimulated area." However in Fig 5C, the surrogate DAOD values (purple line) do not look different from the stimulated area (black line). Is there a significant difference here, or is this an error? I assume so based on Fig 5E, but this was not clear from the legend or text.

The difference in entropies between the distribution of orientation preferences and DAOD is statistically significant for the example surrogate stimulus provided ($p=0$ for float64 precision). We agree with the reviewer that visually the difference is very difficult to discern, due to the size of the figure and overlapping curves. However, notice that the activation over the majority orientation (towards $\pi / 2$) is stronger than what would be expected by the OD distribution, and the activation over the minority orientation (towards $-\pi / 2$) is weaker.

Nevertheless we agree that this makes the paper confusing to read. In that specific panel we chose to present a specific pair of endogenous-surrogate stimuli to illustrate the relationship, as they showed most clearly the sharpening in the endogenous case. We have now swapped it out for another pair where endogenous sharpening is less strong, but where the sharpening is easier to discern for both endogenous and surrogate, hence better serving the illustrative purpose. Finally, note that a proper entropy quantification across all pairs is presented in panel E (OD vs. DAOD entropy, previously panel D). Furthermore, we added Supplementary Figure S10, showing all individual DAOD curves.

Other:

What is the scale bar on the traveling waves in Fig 2A?

This was indeed missing, we have now added both a colorbar and an indication of the spatial scale of the activity shown in Figure 2A.

Fig 2 C, D and G do not have y-axes.

We on purpose did not add any tick on the y-axes of these panels. These y-axes refer to normalized distribution densities, the exact values are hence irrelevant and would provide no information to the reader. We reproduced the results of Fig. 6 Davis et al. 2021 which also don't provide any numerical values nor ticks for these axes.

In Fig 4C please add labels for which map was made from spontaneous activity and optogenetic stimulation.

We thank the reviewer for spotting this omission, the labels to Fig 4C and D have been added.

In Fig 4I, is the 'Endogenous No Bias' significantly different from 'Surrogate' or 'Endogenous'? Please include this analysis and add errorbars / variance metric to the panel.

All three populations of correlations have significantly different means (endogenous: 0.7996 ± 0.004 , surrogate: 0.7097 ± 0.006 , endogenous no bias: 0.7465 ± 0.003 ; $p_{\text{endo-surr}}=2 \times 10^{-18}$, $p_{\text{endo-endo_no_bias}}=5 \times 10^{-16}$, $p_{\text{surr-endo_no_bias}}=3 \times 10^{-6}$) - we have added this detail to the figure caption. The confidence intervals for all three populations are already present in the panel.

In Figure 6 C-F, to evaluate how changing different model parameters affects the different metrics, it would be helpful if you visually indicated the biological range of the in vivo data and/or the model parameters used in the previous figures.

We thank the reviewer for those suggestions. We believe that the reviewer intended to refer to Figure 7 instead of Figure 6 here. We have now updated Figure 7 such that the

parameters used in the base version of the model are highlighted either with red font (Figure 7AB) or with red circles (Figure 7C-F). We applied the same changes to Figures S12 (previously S10) and S13 (previously S11), as well as to the newly added Figure S17.

However, we decided to not follow the recommendation concerning the indication of the biological range of the in vivo data. For some of the measures (the similarity between optogenetically evoked correlation maps and orientation map) this data is not available, while for others only the mean across animals is available, not allowing indication of meaningful inter-individual range (stLFP residual difference between iso- and ortho-oriented sites). Finally, while we have to resort to such multi-species comparisons in this paper due to lack of data in single species, we feel placing it in this way within a single figure would be particularly misleading as it would insinuate these ranges are one single 'achievable' reference, where in fact they are multiple (even if highly related and overlapping) references. We have now included a discussion of the limitations caused by these multi-species comparisons in the manuscript's discussion (lines 835-843).

Reviewer #5

Reviewer #5 (Remarks to the Author):

Reviewer #6

Reviewer #6 (Remarks to the Author):

Other modifications

Statistical testing of the iso-orientation bias of the stLFP residuals (section Spontaneous travelling waves exhibit iso-oriented bias of propagation that depends on functionally biased connectivity in layer 2/3) is now made using two-sided Wilcoxon signed-rank tests instead of simple t-tests, as the former avoids making any assumption of normality in the distribution of the data. We now report there the means, the standard error of the means, the p-value, the sample size and the type of test applied. In the same result subsection, we have now also added statistical testing for the configuration of the model without any iso-orientation bias in long-range horizontal connectivity (lines 166-168, 175-177 and 273-274). Consecutive to the changes in statistical reporting, we have added the following sentence in methods (line 1095):

Quantitative values in this article are expressed as mean \pm SEM

In the legends of Figure 7B and in the corresponding caption, we replaced 'Propagation delay' with 'conduction delay' so that it remains consistent with the rest of the article. We have also now added the adjective 'distance-dependent' when referring to those conduction delays in the same caption and in lines 694 and 1371. This was to emphasize explicitly that here we refer to these distance-dependent conduction delays and not to the constant synaptic transmission delays.

Similarity values in Figure 7C-F and Figure S13 (previously S11) were computed without prior bandpass filtering the calcium signal (see Methods 2.5), unlike the results shown in Figure 3H and described in section 'Functional specificity of cortical connections dictate correlation structure of spontaneous activity'. We have now corrected those figures so that the level of similarity corresponds to the results shown in figure 3H.

Line 1289: We had wrongly stated that s_n in the prediction interval equation is the sample variance whereas it actually represents the sample standard deviation. We have now corrected that.

Dimensionality for Figure 3H has previously been computed with added screening for large-scale events, while other metrics were not. We rectified this inconsistency, with Figure 3H now showing dimensionality values consistent with the other metrics. See comparison of screening and non-screening results on added Supplementary Figure S7.

We discovered that in our calculation of DAOD, histogram bins were set automatically to the data range instead of correctly spanning the full range of $[0, \pi]$. In some endogenous patterns, this resulted in incorrectly high DAOD entropy values. We corrected this issue by explicitly setting the bin range to $[0, \pi]$. All conclusions remain unchanged.

There was an error in this Figure S5: The results shown in the previous version corresponded to the model presented in Antolík et al. 2024 and not to the new iteration of the model which we present in this paper. We apologize for this unfortunate oversight. The results are however very similar across both model versions as shown now in the updated version of Figure S5. Panels A and B have also now been swapped to correspond to the order of presentation of Antolík et al. 2024, and the caption of the figure has been modified accordingly.

Finally, as a part of the revision process, we have made various minor changes to the manuscript text to improve readability.

Our responses to reviewers' comments are in **red**. Changed text in the manuscript cited in our responses is in *italics*, and **line numbers refer to the version of the manuscript with highlighted changes**.

Reviewer #1 (Remarks to the Author):

The authors have done a commendable job at addressing the issues I raised in my previous review. A few are worth noting.

First, on the issue of young (pre-eye opening) vs adult animals and the absence vs presence of long-range patch connections: I accept the point that the model of Smith et al., for young animals, required very strong recurrent connections relative to afferent connections, which is not appropriate for the adult. So the phenomena seems to require two models for a complete explanation: the Smith model for young animals and the model in the current study for adult animal. While this is not parsimonious, it is not unreasonable. This is acknowledged and discussed in the revised manuscript. Meanwhile, there are many other contributions of the paper, so issue should not be an impediment to publication.

Second, the explanation about the different observed time scales for spontaneous activity in V1, as a measurement artifact, in terms of the effect of the dynamics of calcium imaging was interesting.

Third, it is also reasonable to leave to further work a study of the functional/computational insights available from the model, and instead focus on firmly establishing the model's explanatory power in relation to diverse experimental observations.

With respect to the explanation about the emergence of complex cells in L2/3: while it is reasonable to point out that L2/3 neurons pool of the responses of L4 neurons with different spatial phase tuning, this is not sufficient - the form of the pooling matters. E.g. linear or threshold-linear pooling will not result in phase invariance (e.g. the threshold-linear model in Carandini 2006, Fig. 1 won't work, with Gabor receptive fields). So some explanation of how the correct nonlinear pooling arises would be helpful.

We are somewhat confused about this comment. Pooling of many spatially-coregistered and co-oriented gabor filters varying in spatial phase, each followed by a threshold-linear function (prior pooling), will indeed lead to a complex-cell like behavior. For example, the cell will respond to every phase of an optimally oriented grating - as for every phase it will have a highly responding pre-synaptic neuron - as per the construction we have outlined above. This is exactly the mechanism through which complex cells arise in our model - the L4 simple cells being the 'gabor filters with threshold-linear transfer function' and L2/3 cells pooling from many such L4 co-oriented cells that only vary in phase preference. Of course this simplified description ignores all the recurrent interactions that further reshape the neural responses in our model (and cortex), but the basic simple/complex properties already arise in the model just via the feed-forward pathway alone. This also is not a new mechanism,

having been used in previous computational studies including some from our group (Sakai et al., 2000; Antolík et al., 2011).

Regarding the Carandini 2006 figure 1, we do not understand why the reviewer states that 'Fig 1 won't work'. The Carandini 2006 itself is making a case that the models in Figure 1 are a good approximation of simple/complex cells. Carandini 2006 figure 1 shows 'sketches' of gabor filters (quantised to only three intensity levels) and it would indeed work with fully detailed Gabor filters. That particular figure is a sketch and to get a truly 'smoothly phase invariant' complex cell one would have to have a greater variety of phases in the gabor bank (not just 4 as in the illustrative figure), but it is a good first approximation, and in our model there is indeed large L4 presynaptic pool for every L2/3 neuron.

Is the reviewer concerned about some other secondary properties of the complex cells, other than the canonical phase invariance which is certainly reproduced in the Carandini 2006 model and in ours? If so we would need to know specifics to be able to react. Likewise, If the reviewer disagrees for some reason with the conclusions of Carandini 2006 we would need to hear the specific counterargument so we can react.

Reviewer #2 (Remarks to the Author):

Review of revised manuscript "Iso-orientation bias of layer 2/3 connections: the unifying mechanism of spontaneous, visually and optogenetically driven V1 dynamics"

(Line numbering of document including changes in the manuscript.)

The reviewer thanks the authors very comprehensively addressing the raised questions (including insightful additional analyses) and correcting the identified minor mistakes.

The additional work greatly improves the strength of the manuscripts and makes it more convincing by the unique opportunities to alter key parameters unique to simulation studies.

The following comments are only minor and meant to further improve the clarity of the manuscript.

Review:

Introduction

L33 f.

The authors write

"Several phenomena highlight the interconnectedness of visually and non-visually driven dynamics: ..."

One of the later mentioned phenomena is grouped under the heading: "Optogenetically evoked patterns of activity".

In my opinion, this is in stark contrast to the other mentioned phenomena, "Spontaneous travelling waves", "Modular spontaneous activity".

Do the authors agree that there seems to be somewhat of a difference? Or could they explain why there is none?

The explanatory text for the "Optogenetically evoked patterns of activity" does not make this clearer, since no concrete "phenomenon" is mentioned, but rather what optogenetic stimulations allows to study.

The reviewer is correct that we didn't make the 'interconnectedness' sufficiently clear. We expanded the explanatory text of the relevant section with an example of optogenetic stimulation patterns to clarify it:

For example optogenetic stimulation aligned with an orientation map is more effective at eliciting a response than un-aligned stimulation.

L70 ff.

The authors write:

"By comparing to data from multiple species, we demonstrate that the model replicates (i) distributions of wavelength and propagation speed in spontaneous waves observed in marmoset, along with biases toward iso-oriented propagation seen in macaque and cat V1, (ii) modular spontaneous activity correlated with orientation maps in both excitatory and inhibitory neurons in ferret V1, (iii) impulse-response properties to both full-field homogeneous and patterned optogenetic stimulation in ferret V1, and (iv) range of visually driven phenomena."

Do they mean "a range of visually driven phenomena" (as in "a number") or the "spatial range" of such phenomena? Please clarify.

Thank you for your suggestion, it is the second interpretation. We modified the sentence to make it clear (lines 80-81):

... and (iv) a broad range of different visually driven phenomena.

L79 ff.

The authors write:

"Parametric analysis of the model reveals ..."

I think what the authors mean is that changing certain parameters leads to activity that does not match to the experimentally observed one. This is of course a strong case for their model. I stumbled, however, over the formulation of "Parametric analysis" which for me immediately evoked "Analysis via parametric statistics". Maybe the authors could reformulate this slightly for the sake of clarity.

Thank you for the notice, we changed the sentence (line 84) to:

Analysis of the model's parameters reveals ...

Results

L180 f.

The authors write:

"Next, Davis et al. offered a detailed characterization of the spontaneous wave properties in marmoset MT using the Generalized Phase analysis (7, 15)."

I couldn't find the place in the manuscript where the authors resolve the abbreviation MT. If it is not in the manuscript, please add.

It was indeed not in the manuscript, we added it to the mentioned sentence (lines 160-161):

Next, Davis et al. offered a detailed characterization of the spontaneous wave properties in marmoset middle temporal area (MT) using ...

L285 ff.

The authors write:

"Activity of both inhibitory and excitatory neurons is correlated over multiple millimetre distances, with excitatory activity resembling orientation preference maps in young and adult carnivore/primate ferret V1 (10, 20, 31, 32)."

To my understanding of the cited literature, the correlation patterns of neural activity (and thus predominantly excitatory activity) resemble orientation preference maps and not the "excitatory activity" itself, as stated by the authors. Please comment on this.

Thank you for pointing out this imprecision, the reviewer is correct. We modified the relevant sentence (lines 194-196) to:

Activity of both inhibitory and excitatory neurons is correlated over multiple millimetre distances, with correlation patterns of excitatory activity resembling orientation preference maps ...

L564 f.

The authors write:

"We have shown that model responses to endogenous (OP map correlated) optogenetic stimuli are stronger than to equivalent surrogate stimuli."

I couldn't find what OP might stand for? Does the other mean OT for orientation?

Indeed, we never introduced this abbreviation - it stands for “orientation preference”. We have replaced “OP” with “orientation preference” here, as the abbreviation is not used anywhere else in the text.

L656 ff.

The authors write:

"The precise location of this inflection point is the result of a complex interplay between the spatial scale of the orientation map, the spatial extents of long-range excitatory and short-range inhibitory connections, and balance of excitation and inhibition in the network. Nevertheless, the existence of the inflection point at around 100 μ m radius is an important prediction of the model."

Some (if not all) of the quantities mentioned to underlie the point of inflection depend on the species (e.g. the iso-orientation domain size between cat and macaque differs on average). It is great that you make a specific prediction from your computational model. For which animal model would you expect it to fit best?

We agree that this prediction lacked clarity given that this value would be dependent on the species. Given that the model presented here is derived from a model constrained anatomically on mostly cat data Antolík et al. (2024), we'd expect this prediction to fit cat V1 the best.

That said we agree with the reviewer that the spatial frequency is likely to be the main determining factor, and so in the manuscript we now chose to report a value in terms of hypercolumn spacing rather than on absolute distance, to make the prediction species independent (lines 424-426):

Nevertheless, the existence of the inflection point at around 100 μ m radius, corresponding to ~ 0.11 of the model hypercolumn distance, is an important prediction of the model."

Methods

L904

The threshold slope factor is written as ΔT instead of Δ_{T} (the letter version is used in equation 1).

Thank you, this is now corrected.

L942

The authors write:

"The phase preference Ψ , is generated randomly."

I assume the authors mean generated randomly following a uniform distribution with on an appropriate interval. Could the authors make this more explicit?

Indeed, we have now corrected this sentence to make it more explicit (lines 693-394):
The phase preference Ψ is generated randomly from an uniform distribution with support $[0, 2\pi)$.

L950 ff.

The authors write:

"Individual layer 4 excitatory and inhibitory cells receive connections from the LGN whose number is drawn from uniform distributions of respective boundaries of [90, 190] and [112, 168]."

I had to read the sentence multiple times to understand what the author want to communicate with the term "boundaries". I think the technical term the authors are looking form is the "support" of a uniform distribution.

Thank you, we have now modified this sentence accordingly.

L978 ff.

Could the authors add the units for θ and α ?

We have now added the units for both theta (μm) and alpha (μm^{-1}).

L1250 ff.

The dimensionality metric employed by the authors is known as participation ration (e.g. Dahmen et al. 2019). Since there many other way to assess the dimensionality of neural activity (for example based on the number of principal components needed to reach a certain value of explained variance) the authors should mention the name of this dimensionality metric and cite the parts of the relevant literature (Abbot, Rajan, Sompolinsky, 2011, "Interactions between intrinsic and stimulus evoked activity in recurrent neural networks", Mazzucato et al. 2016, "Stimuli reduce the dimensionality of cortical activity")

Thank you for the notice - as Smith et al. and Mulholland et al. did not detail the background of this metric in their Methods, we only referred to them when describing it. We have now changed the relevant Methods section to include this detail and reference relevant literature (lines 995-997):

We calculated all 3 metrics following Mulholland et al. (20). We estimated the dimensionality (also known as participation ratio (63–65)) of spontaneous activity patterns as follows: ...

L1323 ff.

I might have missed it here but I do not understand how the authors estimate the temporal scale of spontaneous activity. I do get that they calculate the temporal autocorrelation function. But what then? Do they fit an exponential to extract something. Do they use a

method like the one suggested by Zeerati et al 2022. Or do you just visually compare the autocorrelation function of simulated and recorded activity?

For comparing the temporal scale of calcium imaging and raw signal we did not perform further quantitative fits of the autocorrelations functions as the difference between the two conditions was so large that we found additional analysis superfluous (Figure 5C). For the comparison involving deconvolved time courses, we quantified the timescale as the position of the first maximum of the autocorrelation function. We chose this measure because the source data contained only a short (4 s) segment of deconvolved ferret activity, which made the tails of the autocorrelation function less reliable. We added this detail to Methods (lines 1077-1079):

As the available deconvolved time course spanned only 4s, the autocorrelation features on longer timescales (>0.5s) might be less reliable. As such, when comparing the timescales of simulated and ferret spontaneous activity, we used the position of the first autocorrelation peak as our metric.

Reviewer #3 (Remarks to the Author):

The authors have satisfied all my concerns from the from first round of reviews. I appreciated their diligent responses to all four reviewers, the manuscript is now much improved in my opinion. I wish the authors the best of luck in managing the current overwhelming length of the paper moving forward.

Reviewer #4 (Remarks to the Author):

Thank you for the work and the new revisions. The authors have made a significant effort in revision to assess reviewer concerns. The edits made to the manuscript have greatly improved its interpretability and transparency. The authors have addressed my concerns. I have one comment on the revisions, which does not require followup.

Regarding the local stimulus location experiments: Thank you for taking the time to investigate this question. By moving the stimulation target around, you have demonstrated that where you stimulate can have variable effects on the response of the network, with column centers providing the strongest amplification, while edges and pinwheels are less effective. This is another piece of evidence supporting a “soft winner-take-all mechanism”, which has been demonstrated in other ways throughout the manuscript. For this reason, it may be worthwhile to include these figures in the supplement (both are illuminating, but the first one alone may be enough to make this point without having to do extensive simulations). However, I agree an in-depth study on this topic is outside the scope of this paper, and therefore leave this to the decision of the authors.

We thank the reviewer for their interest in this analysis. We found these results quite interesting, especially how the various ratios of orientations in the stimulated areas influenced the recruitment of oriented subpopulations in the surround. We think there is more

interesting nuance to explore in these analyses, which a short reference to the Supplementary Figures would not do justice to. Therefore, we opt to not include this analysis in this paper and hope to expand on it in the future work.

Reviewer #5 (Remarks to the Author):

Reviewer #6 (Remarks to the Author):

Other changes:

Equation (3) was incorrectly showing that the functional and spatial components of the connectivity probability distributions are summed together whereas they are multiplied to one another. This was now corrected (line 729). Following this logic, as inhibitory connections in layer 2/3 are purely distance-based, their functional component is set to 1 and not to 0 as previously indicated. Line 751 has now been modified accordingly.